# Variance Reduced Distributed Nonconvex Optimization Using Matrix Stepsizes

## Abstract

Matrix-step-size gradient descent algorithms have demonstrated superior performance in solving non-convex optimization problems compared to their scalar step-size counterparts. The det-CGD algorithm, as introduced by Li et al. (2024), leverages matrix stepsizes to perform compressed gradient descent for non-convex objectives and matrix-smooth problems in a federated manner. The authors establish the algorithm's convergence to a neighborhood of a weighted stationarity point under a convex condition for the symmetric and positive-definite matrix stepsize. In this paper, we propose two variance-reduced versions of the det-CGD algorithm, incorporating MARINA and DASHA methods. Notably, we establish theoretically and empirically, that det-MARINA and det-DASHA outperform MARINA, DASHA and the distributed det-CGD algorithms in terms of iteration and communication complexities.

## 1 Introduction

We focus on optimizing the finite sum non-convex objective

$$\min_{x \in \mathbb{R}^d} \left\{ f(x) := \frac{1}{n} \sum_{i=1}^{n} f_i(x) \right\}. \tag{1}$$

In this context, each function $f_i : \mathbb{R}^d \to \mathbb{R}$ is differentiable and bounded from below. This type of objective function finds extensive application in various practical machine learning algorithms, which increase not only in terms of the data size but also in the model size and overall complexity as well. As a result, most neural network architectures result in highly non-convex empirical losses, which need to be minimized. In addition, it becomes computationally infeasible to train these models on one device, often excessively large, and one needs to redistribute them amongst different devices/clients. This redistribution results in a high communication overhead, which often becomes the bottleneck in this framework.

In other words, we have the following setting. The data is partitioned into $n$ clients, where the $i$-th client has access to the component function $f_i$ and its derivatives. The clients are connected to each other through a central device, called the server. In this work, we are going to study iterative gradient descent-based algorithms that operate as follows. The clients compute the local gradients in parallel. Then they compress these gradients to reduce the communication cost and send them to the server in parallel. The server then aggregates these vectors and broadcasts the iterate update back to the clients. This meta-algorithm is called federated learning. We refer the readers to Konečný et al. (2016); McMahan et al. (2017); Kairouz et al. (2021) for a more thorough introduction.

### 1.1 Contributions

In this paper, we introduce two novel federated learning algorithms named det-MARINA and det-DASHA. These algorithms extend a recent method called det-CGD (Li et al., 2024), which aims to solve problem (1) using matrix stepsized gradient descent. Under the matrix smoothness assumption, the authors demonstrate that the matrix stepsized version of the distributed compressed gradient gescent (Khirirat et al., 2018) algorithm enhances communication complexity compared to its scalar counterpart. However, in their analysis, Li et al. (2024) show stationarity only within

a certain neighborhood due to stochastic compressors. The neighborhood influences the solution's accuracy, leading to a smaller step size and, consequently, convergence when aiming for a specified accuracy. Our algorithms address this issue by adapting two variance reduction schemes, namely, MARINA (Gorbunov et al., 2021) and DASHA (Tyurin & Richtárik, 2024), incorporating variance reduction into matrix stepsizes. We establish theoretically and empirically, that both algorithms outperform their scalar alternatives, as well as the distributed det-CGD algorithms. In addition, we describe specific matrix stepsize choices, for which our algorithms beat MARINA, DASHA and distributed det-CGD both in theory and in practice. The various numerical evidence obtained from the extensive experiments further corroborates our findings.

## 2 BACKGROUND

For a given $\varepsilon > 0$, finding an approximately global optimum, that is $x_\varepsilon$ such that $f(x_\varepsilon) - \min_x f(x) < \varepsilon$, is known to be NP-hard (Jain et al., 2017; Danilova et al., 2022). However, gradient descent based methods are still useful in this case. When these methods are applied to non-convex objectives, they treat the function $f$ as locally convex and aim to converge to a local minimum. Despite this simplification, such methods have gained popularity in practice due to their superior performance compared to other approaches for non-convex optimization, such as convex relaxation-based methods (Tibshirani, 1996; Cai et al., 2010).

### 2.1 STOCHASTIC GRADIENT DESCENT

Arguably, one of the most prominent meta-methods for tackling non-convex optimization problems is stochastic gradient descent (SGD). The formulation of SGD is presented as the following iterative algorithm: $x^{k+1} = x^k - \gamma g^k$. Here, $g^k \in \mathbb{R}^d$ serves as a stochastic estimator of the gradient $\nabla f(x^k)$. SGD essentially mimics the classical gradient descent algorithm, and recovers it when $g^k = \nabla f(x^k)$. In this scenario, the method approximates the objective function $f$ using a linear function and takes a step of size $\gamma$ in the direction that maximally reduces this approximation. When the stepsize is sufficiently small, and the function $f$ is suitably smooth, it can be demonstrated that the function value decreases, as discussed by Bubeck et al. (2015); Gower et al. (2019).

However, computing the full gradient can often be computationally expensive. In such cases, stochastic approximations of the gradient come into play. Stochastic estimators of the gradient can be employed for various purposes, leading to the development of different methods. These include stochastic batch gradient descent (Nemirovski et al., 2009; Johnson & Zhang, 2013; Defazio et al., 2014), randomized coordinate descent (Nesterov, 2012; Wright, 2015), and compressed gradient descent (Alistarh et al., 2017; Khirirat et al., 2018; Mishchenko et al., 2019). The latter, compressed gradient descent, holds particular relevance to this paper, and we will delve into a more detailed discussion of it in subsequent sections.

### 2.2 SECOND ORDER METHODS

The stochastic gradient descent is considered as a first-order method as it uses only the first order derivative information. Although being immensely popular, the first order methods are not always optimal. Not surprisingly, using higher order derivatives in deciding update direction can yield to faster algorithms. A simple instance of such algorithms is the Newton Star algorithm (Islamov et al., 2021):

$$x^{k+1} = x^k - \left(\nabla^2 f(x^\star)\right)^{-1} \nabla f(x^k), \tag{NS}$$

where $x^\star$ is the minimum point of the objective function. The authors establish that under specific conditions, the algorithm's convergence to the unique solution $x^\star$ in the convex scenario occurs at a local quadratic rate. Nonetheless, its practicality is limited since we do not have prior knowledge of the Hessian matrix at the optimal point. Despite being proposed recently, the Newton-Star algorithm gives a deeper insight on the generic Newton method (Gragg & Tapia, 1974; Miel, 1980; Yamamoto, 1987):

$$x^{k+1} = x^k - \gamma \left(\nabla^2 f(x^k)\right)^{-1} \nabla f(x^k). \tag{NM}$$

Here, the unknown Hessian of the Newton-Star algorithm, is estimated progressively along the iterations. The latter causes elevated computational costs, as the inverting a large square matrix is

expensive. As an alternative, quasi-Newton methods replace the inverse of the Hessian at the iterate with a computationally cheaper estimate (Broyden, 1965; Dennis & Moré, 1977; Al-Baali & Khalfan, 2007; Al-Baali et al., 2014).

## 2.3 Fixed matrix stepsizes

The det-CGD algorithm falls into this framework of the second order methods as well. Proposed by Li et al. (2024)[1], the algorithm suggests using a uniform "upper bound" on the inverse Hessian matrix. Assuming matrix smoothness of the objective (Safaryan et al., 2021), they replace the scalar stepsize with a positive definite matrix $\boldsymbol{D}$. The algorithm is given as follows:

$$x^{k+1} = x^k - \boldsymbol{D}\boldsymbol{S}^k\nabla f(x^k). \tag{det-CGD}$$

**Matrix $\boldsymbol{D}$.** Here, $\boldsymbol{D}$ plays the role of the stepsize. Essentially, it uniformly lower bounds the inverse Hessian. The standard SGD is a particular case of this method, as the scalar stepsize $\gamma$ can be seen as a matrix $\gamma\boldsymbol{I}_d$, where $\boldsymbol{I}_d$ is the $d$-dimensional identity matrix. An advantage of using a matrix stepsize is more evident if we take the perspective of the second order methods. Indeed, the scalar stepsize $\gamma\boldsymbol{I}_d$ uniformly estimates the largest eigenvalue of the Hessian matrix, while $\boldsymbol{D}$ can capture the Hessian more accurately. The authors show both theoretical and empirical improvement that comes with matrix stepsizes.

**Matrix $\boldsymbol{S}^k$.** We assume that $\boldsymbol{S}^k$ is a positive semi-definite, stochastic sketch matrix. Furthermore, it is unbiased: $\mathbb{E}[\boldsymbol{S}^k] = \boldsymbol{I}_d$. We notice that det-CGD can be seen as a matrix stepsize instance of SGD, with $g^k = \boldsymbol{S}^k\nabla f(x^k)$. The sketch matrix can be seen as a linear compressing operator, hence the name of the algorithm: Compressed Gradient Descent (CGD) (Alistarh et al., 2017; Khirirat et al., 2018). A commonly used example of such a compressor is the Rand-$\tau$ compressor. This compressor randomly selects $\tau$ entries from its input and scales them using a scalar multiplier to ensure an unbiased estimation. By adopting this approach, instead of using all $d$ coordinates of the gradient, only a subset of size $\tau$ is communicated. Formally, Rand-$\tau$ is defined as $\boldsymbol{S} = \frac{d}{\tau}\sum_{j=1}^{\tau} e_{i_j}e_{i_j}^\top$, where $e_{i_j}$ denotes the $i_j$-th standard basis vector in $\mathbb{R}^d$. For a more comprehensive understanding of compression techniques, we refer to Safaryan et al. (2022b).

## 2.4 The neighborhood of the distributed Det-CGD

The distributed version of det-CGD follows the standard federated learning paradigm (McMahan et al., 2017). The pseudocode of the method, as well as the convergence result of Li et al. (2024), can be found in Appendix I. Informally, their convergence result can be written as

$$\min_{k=1,\dots,K} \mathbb{E}\left[\|\nabla f(x^k)\|_{\boldsymbol{D}}^2\right] \le \mathcal{O}\left(\frac{(1+\alpha)^K}{K}\right) + \mathcal{O}(\alpha),$$

where $\alpha > 0$ is a constant that can be controlled. The crucial insight from this result is that the error bound does not diminish as the number of iterations increases. In fact, by controlling $\alpha$ and considering a large $K$, it is impossible to make the second term smaller than $\varepsilon$. This implies that the algorithm converges to a certain neighborhood surrounding the (local) optimum. Ultimately, the model we obtain suffers from lower accuracy and performance due to the inaccuracies introduced by this neighborhood. This phenomenon is a common occurrence in SGD and is primarily attributable to the variance associated with the stochastic gradient estimator. In the case of det-CGD the stochasticity comes from the sketch $\boldsymbol{S}^k$.

## 2.5 Variance reduction

To eliminate this neighborhood, various techniques for reducing variance are employed. One of the simplest techniques applicable to CGD is gradient shifting. By replacing $\boldsymbol{S}^k\nabla f(x^k)$ with $\boldsymbol{S}^k(\nabla f(x^k) - \nabla f(x^\star)) + \nabla f(x^\star)$, the neighborhood effect is removed from the general CGD.

---

[1]In the original paper, the algorithm is referred to as det-CGD, as there is a variant of the same algorithm named det-CGD2. Since we are going to use only the first one and our framework is applicable to both, we will remove the number in the end for the sake of brevity.

This algorithm is an instance of a more commonly known method called SGD$_\star$ (Gower et al., 2020). However, since the exact optimum $x^\star$ is typically unknown, this technique encounters similar challenges as the Newton-Star algorithm mentioned earlier. Fortunately, akin to quasi-Newton methods, one can employ methods that iteratively learn the optimal shift (Shulgin & Richtárik, 2022). A line of research focuses on variance reduction for CGD based algorithms on this insight.

To eliminate the neighborhood in the distributed version of CGD, denoted as det-CGD1, we apply a technique called MARINA (Gorbunov et al., 2021). MARINA cleverly combines the general shifting (Shulgin & Richtárik, 2022) technique with loopless variance reduction techniques (Qian et al., 2021). This approach introduces an alternative gradient estimator specifically designed for the federated learning setting. Thanks to its structure, it allows to establish an upper bound on the stationarity error that diminishes significantly with a large number of iterations. In this paper, we construct the analog of the this algorithm called det-MARINA, using matrix stepsizes and sketch gradient compressors. For this new method, we prove a convergence guarantee similar to the results of Li et al. (2024) without a neighborhood term.

Furthermore, we also propose det-DASHA, which is the extension of DASHA in the matrix stepsize setting. The latter was proposed by Tyurin & Richtárik (2024) and it combines MARINA with momentum variance reduction techniques (Cutkosky & Orabona, 2019). DASHA offers better practicality compared to MARINA, as it always sends compressed gradients and does not need to synchronize among all the nodes.

## 2.6 ORGANIZATION OF THE PAPER

The rest of the paper is organized as follows. Section 3 discusses the general mathematical framework. Section 4 and Section 5 present the det-MARINA and det-DASHA algorithms, respectively. We show the superior theoretical performance of our algorithms compared to the relevant existing algorithms, that is MARINA, DASHA and det-CGD in Section 6. The experimental results validating our theoretical findings are presented in Section 7, with additional details and setups available in the Appendix.

## 3 MATHEMATICAL FRAMEWORK

In this section we present the assumptions that we further require in the analysis.

**Assumption 3.1.** (Lower Boundedness) There exists $f^\star \in \mathbb{R}$ such that, $f(x) \geq f^\star$ for all $x \in \mathbb{R}^d$.

This is a standard assumption in optimization, as otherwise the problem of minimizing the objective would not be correct mathematically. We then introduce a matrix version of Lipschitz continuity for the gradient.

**Definition 3.2.** Matrix Smoothness  Assume that $f : \mathbb{R}^d \to \mathbb{R}$ is a continuously differentiable function and matrix $\boldsymbol{L} \in \mathbb{S}_{++}^d$. We say the gradient of $f$ is $\boldsymbol{L}$-Lipschitz if for all $x, y \in \mathbb{R}^d$

$$\|\nabla f(x) - \nabla f(y)\|_{\boldsymbol{L}^{-1}} \leq \|x - y\|_{\boldsymbol{L}} . \tag{2}$$

**Assumption 3.3.** Each function $f_i$ is $\boldsymbol{L}_i$-gradient Lipschitz, while $f$ is $\boldsymbol{L}$-gradient Lipschitz.

In fact, the second half of the assumption is a consequence of the first one. Below, we formalize this claim.

**Lemma 3.4.** *If $f_i$ is $\boldsymbol{L}_i$-gradient Lipschitz for every $i = 1, \ldots, n$, then function $f$ has $\boldsymbol{L}$-Lipschitz gradient with $\boldsymbol{L} \in \mathbb{S}_{++}^d$ satisfying*

$$\frac{1}{n} \sum_{i=1}^{n} \lambda_{\max}\left(\boldsymbol{L}^{-1}\right) \cdot \lambda_{\max}\left(\boldsymbol{L}_i\right) \cdot \lambda_{\max}\left(\boldsymbol{L}_i \boldsymbol{L}^{-1}\right) = 1 .$$

*Remark* 3.5. In the scalar case, where $\boldsymbol{L} = L\boldsymbol{I}_d$, $\boldsymbol{L}_i = L_i \boldsymbol{I}_d$, the relation becomes $L^2 = \frac{1}{n} \sum_{i=1}^{n} L_i^2$. This corresponds to the statement in Assumption 1.2 in (Gorbunov et al., 2021).

Nevertheless, the matrix $\boldsymbol{L}$ found according to Lemma 3.4 is only an estimate. In principle, there might exist a better $\boldsymbol{L}_f \preceq \boldsymbol{L}$ such that $f$ has $\boldsymbol{L}_f$-Lipschitz gradient.

More generally, this condition can be interpreted as follows. The gradient of $f$ naturally belongs to the dual space of $\mathbb{R}^d$, as it is defined as a linear functional on $\mathbb{R}^d$. In the scalar case, $\ell_2$-norm is self-dual, thus (2) reduces to the standard Lipschitz continuity of the gradient. However, with the matrix smoothness assumption, we are using the $\boldsymbol{L}$-norm for the iterates, which naturally induces the $\boldsymbol{L}^{-1}$-matrix norm for the gradients in the dual space. This insight, which is originally presented by Nemirovski & Yudin (1983), plays a key role in our analysis. See Appendix F for a more thorough discussion on the properties of Assumption 3.3, as well as its connection to matrix smoothness (Safaryan et al., 2021).

# 4 MARINA-BASED VARIANCE REDUCTION

In this section, we present det-MARINA with its convergence result. We construct a sequence of vectors $g^k$ which are stochastic estimators of $\nabla f(x^k)$. At each iteration, the server samples a Bernoulli random variable (coin flip) $c_k$ and broadcasts it in parallel to the clients, along with the current gradient estimate $g^k$. Each client, then, does a det-CGD-type update with the stepsize $\boldsymbol{D}$ and a gradient estimate $g^k$. The next gradient estimate $g^{k+1}$ is then computed. With a low probability, that is when $c_k = 1$, we take the $g^{k+1}$ to be the full gradient $\nabla f(x^{k+1})$. Otherwise, we update it using the compressed gradient differences at each client. See Algorithm 1 for the pseudocode of det-MARINA.

---

**Algorithm 1** det-MARINA

1: **Input:** starting point $x^0$, stepsize matrix $\boldsymbol{D}$, probability $p \in (0, 1]$, number of iterations $K$
2: Initialize $g^0 = \nabla f(x^0)$
3: **for** $k = 0, 1, \ldots, K - 1$ **do**
4:     Sample $c_k \sim \text{Be}(p)$
5:     Broadcast $g^k$ to all workers
6:     **for** $i = 1, 2, \ldots$ in parallel **do**
7:         $x^{k+1} = x^k - \boldsymbol{D} \cdot g^k$
8:         **if** $c_k = 1$ **then**
9:             $g_i^{k+1} = \nabla f_i(x^{k+1})$
10:        **else**
11:           $g_i^{k+1} = g^k + \boldsymbol{S}_i^k \left( \nabla f_i(x^{k+1}) - \nabla f_i(x^k) \right)$
12:        **end if**
13:     **end for**
14:     $g^{k+1} = \frac{1}{n} \sum_{i=1}^n g_i^{k+1}$
15: **end for**
16: **Return:** $\tilde{x}^K$ uniformly sampled from $\{x^k\}_{k=0}^{K-1}$

---

## 4.1 CONVERGENCE GUARANTEES

In the following theorem, we formulate one of the main results of this paper, which guarantees the convergence of Algorithm 1 under the above-mentioned assumptions.

**Theorem 4.1.** *Assume that Assumptions 3.1 and 3.3 hold, and the following condition on stepsize matrix $\boldsymbol{D} \in \mathbb{S}_{++}^d$ holds,*

$$\boldsymbol{D}^{-1} \succeq \left( \frac{(1-p) \cdot R(\boldsymbol{D}, \mathcal{S})}{np} + 1 \right) \boldsymbol{L}, \tag{3}$$

*where $R(\boldsymbol{D}, \mathcal{S}) := \frac{1}{n} \sum_{i=1}^n \lambda_{\max}(\boldsymbol{L}_i) \lambda_{\max}(\boldsymbol{L}^{-\frac{1}{2}} \boldsymbol{L}_i \boldsymbol{L}^{-\frac{1}{2}}) \times \lambda_{\max}\left( \mathbb{E}\left[ \boldsymbol{S}_i^k \boldsymbol{D} \boldsymbol{S}_i^k \right] - \boldsymbol{D} \right)$. Then, after $K$ iterations of det-MARINA, we have*

$$\mathbb{E}\left[ \left\| \nabla f(\tilde{x}^K) \right\|_{\frac{\boldsymbol{D}}{\det(\boldsymbol{D})^{1/d}}}^2 \right] \leq \frac{2 \left( f(x^0) - f^\star \right)}{\det(\boldsymbol{D})^{1/d} \cdot K}. \tag{4}$$

*Here, $\tilde{x}^K$ is chosen uniformly randomly from the first $K$ iterates of the algorithm.*

*Remark* 4.2. The criterion $\|\cdot\|_{\boldsymbol{D}/\det(\boldsymbol{D})^{1/d}}^2$ is the same as that used in Li et al. (2024), known as determinant normalization. The weight matrix of the matrix norm has determinant 1 after normalization, which makes it comparable to the standard Euclidean norm.

*Remark* 4.3. We notice that the right-hand side of the algorithm vanishes with the number of iterations, thus solving the neighborhood issue of the distributed det-CGD. Therefore, det-MARINA is indeed the variance reduced version of det-CGD in the distributed setting and has better convergence guarantees.

*Remark* 4.4. Theorem 4.1 implies the following iteration complexity for the algorithm. In order to get an $\varepsilon^2$ stationarity error[2], the algorithm requires $K$ iterations, with

$$K \geq \frac{2(f(x^0) - f^\star)}{\det(\boldsymbol{D})^{1/d} \cdot \varepsilon^2}.$$

*Remark* 4.5. In the case where no compression is applied, that is we have $\boldsymbol{S}_i^k = \boldsymbol{I}_d$, condition (3) reduces to $\boldsymbol{D} \preceq \boldsymbol{L}^{-1}$. The latter is due to $\mathbb{E}\left[\boldsymbol{S}_i^k \boldsymbol{D} \boldsymbol{S}_i^k\right] = \boldsymbol{D}$, which results in $R(\boldsymbol{D}, \mathcal{S}) = 0$. This is expected, since in the deterministic case det-MARINA reduces to GD with matrix stepsize.

The convergence condition and rate of matrix stepsize GD can be found in (Li et al., 2024). Below we do a sanity check to verify that the convergence condition for scalar MARINA can be obtained.

*Remark* 4.6. Let us consider the scalar case. In this case, we have $\boldsymbol{L}_i = L_i \boldsymbol{I}_d, \boldsymbol{L} = L \boldsymbol{I}_d, \boldsymbol{D} = \gamma \boldsymbol{I}_d$ and $\omega = \lambda_{\max}\left(\mathbb{E}\left[\left(\boldsymbol{S}_i^k\right)^\top \boldsymbol{S}_i^k\right]\right) - 1$ Then, condition (3) reduces to

$$\gamma \leq \left[L\left(1 + \sqrt{\frac{(1-p)\omega}{pn}}\right)\right]^{-1}.$$

The latter coincides with the stepsize condition of the convergence result of scalar MARINA.

## 4.2 Optimizing the Matrix Stepsize

As previously noted in Remark 4.2, the norm on the left-hand side of (4) is comparable to the standard Euclidean norm. To optimize the matrix stepsize, our focus will be directed toward the right-hand side of (4). We notice that it decreases in terms of the determinant of the stepsize matrix. Therefore, one needs to solve the following optimization problem to find the optimal stepsize:

$$\begin{aligned} &\text{minimize} &&\log\det(\boldsymbol{D}^{-1}) \\ &\text{subject to} &&\boldsymbol{D} \text{ satisfying (3)}. \end{aligned}$$

The solution of this constrained minimization problem on $\mathbb{S}_{++}^d$ is not explicit. In theory, one may show that the constraint (3) is convex and attempt to solve the problem numerically. However, as stressed by Li et al. (2024), the similar stepsize condition for det-CGD is not easily computed using solvers like CVXPY (Diamond & Boyd, 2016). Instead, we may relax the problem to certain linear subspaces of $\mathbb{S}_{++}^d$. In particular, we fix a matrix $\boldsymbol{W} \in \mathbb{S}_{++}^d$, and define $\boldsymbol{D} := \gamma \boldsymbol{W}$. Then, the condition on the matrix $\boldsymbol{D}$ becomes a condition for the scalar $\gamma$, which is given in the following corollary.

**Corollary 4.7.** *Let* $\boldsymbol{W} \in \mathbb{S}_{++}^d$, *defining* $\boldsymbol{D} := \gamma \cdot \boldsymbol{W}$, *where* $\gamma \in \mathbb{R}_+$. *then the condition in* (3) *reduces to the following condition on* $\gamma$

$$\gamma \leq \frac{2\lambda_{\boldsymbol{W}}}{1 + \sqrt{1 + 4\alpha\beta \cdot \Lambda_{\boldsymbol{W},\mathcal{S}}\lambda_{\boldsymbol{W}}}}, \tag{5}$$

*where*

$$\Lambda_{\boldsymbol{W},\mathcal{S}} := \lambda_{\max}\left(\mathbb{E}\left[\boldsymbol{S}_i^k \boldsymbol{W} \boldsymbol{S}_i^k\right] - \boldsymbol{W}\right),$$

$$\lambda_{\boldsymbol{W}} := \lambda_{\max}^{-1}\left(\boldsymbol{W}^{\frac{1}{2}} \boldsymbol{L} \boldsymbol{W}^{\frac{1}{2}}\right), \quad \alpha := \frac{1-p}{np},$$

$$\beta := \frac{1}{n}\sum_{i=1}^n \lambda_{\max}\left(\boldsymbol{L}_i\right) \cdot \lambda_{\max}\left(\boldsymbol{L}^{-1}\boldsymbol{L}_i\right).$$

---

[2]We say a (possibly random) vector $x \in \mathbb{R}^d$ is an $\varepsilon$-stationary point of a possibly non-convex function $f : \mathbb{R}^d \mapsto \mathbb{R}$, if $\mathbb{E}\left[\|\nabla f(x)\|^2\right] \leq \varepsilon^2$. The expectation is over the randomness of the algorithm

This means that for every fixed $\boldsymbol{W}$, we can find the optimal scaling coefficient $\gamma$. In section Section 6, we will use this corollary to prove that a suboptimal matrix step size, determined in this efficient way, is already better than the optimal scalar step size.

**Further Extension.** A variant of det-CGD was also proposed by Li et al. (2024). This algorithm, has the same structure as det-CGD with the sketch and stepsize interchanged. It was shown, that this algorithm has explicit stepsize condition in the single node setting. In Appendix J, we propose the variance reduced extension of the this algorithm following the MARINA scheme.

## 5 DASHA-BASED VARIANCE REDUCTION

In this section, we present our second algorithm based on DASHA. The latter utilizes a different type of variance reduction based on momentum. Compared to MARINA, DASHA makes simpler optimization steps and does not require periodic synchronization with all the nodes.

---

**Algorithm 2** det-DASHA

---

1: **Input:** starting point $x^0 \in \mathbb{R}^d$, stepsize matrix $\boldsymbol{D} \in \mathbb{S}_{++}^d$, momentum $a \in (0, 1]$, number of iterations $K$
2: Initialize $g_i^0, h_i^0 \in \mathbb{R}^d$ on the nodes and $g^0 = \frac{1}{n} \sum_{i=1}^n g_i^0$ on the server
3: **for** $k = 0, 1, \ldots, K-1$ **do**
4:    $x^{k+1} = x^k - \boldsymbol{D} \cdot g^k$
5:    Broadcast $x^{k+1}$ to all nodes
6:    **for** $i = 1, 2, \ldots n$ in parallel **do**
7:       $h_i^{k+1} = \nabla f_i(x^{k+1})$
8:       $m_i^{k+1} = \boldsymbol{S}_i^k \left( h_i^{k+1} - h_i^k - a \left( g_i^k - h_i^k \right) \right)$
9:       $g_i^{k+1} = g_i^k + m_i^{k+1}$
10:      Send $m_i^{k+1}$ to the server.
11:    **end for**
12:    $g^{k+1} = g^k + \frac{1}{n} \sum_{i=1}^n m_i^{k+1}$
13: **end for**
14: **Return:** $\tilde{x}^K$ uniformly sampled from $\{x^k\}_{k=0}^{K-1}$

---

### 5.1 THEORETICAL GUARANTEES

**Theorem 5.1.** *Suppose that Assumptions 3.1 and 3.3 hold. Let us initialize $g_i^0 = h_i^0 = \nabla f_i(x^0)$ for all $i \in [n]$ in Algorithm 2, and define $\omega_{\boldsymbol{D}} := \lambda_{\max}\left(\boldsymbol{D}^{-1}\right) \cdot \Lambda_{\boldsymbol{D}, \mathcal{S}}$. If $a = \frac{1}{2\omega_{\boldsymbol{D}}+1}$, and the following condition on stepsize $\boldsymbol{D} \in \mathbb{S}_{++}^d$ is satisfied*

$$\boldsymbol{D}^{-1} \succeq \boldsymbol{L} - \frac{4\lambda_{\max}\left(\boldsymbol{D}\right)\omega_{\boldsymbol{D}}\left(4\omega_{\boldsymbol{D}}+1\right)}{n^2} \sum_{i=1}^n \lambda_{\max}\left(\boldsymbol{L}_i\right)\boldsymbol{L}_i,$$

*then the following inequality holds for the iterates of Algorithm 2*

$$\mathbb{E}\left[\left\|\nabla f(\tilde{x}^K)\right\|_{\boldsymbol{D}/(\det(\boldsymbol{D}))^{1/d}}^2\right] \leq \frac{2(f(x^0) - f^\star)}{\det(\boldsymbol{D})^{1/d} \cdot K}.$$

*Here $\tilde{x}^K$ is chosen uniformly randomly from the first K iterates of the algorithm.*

*Remark* 5.2. The term $\Lambda_{\boldsymbol{D}, \mathcal{S}}$ can be viewed as the matrix version of $\gamma \cdot \omega$, where $\omega$ is associated with the sketch, and $\gamma$ is the scalar stepsize. On the other hand, the $\omega_{\boldsymbol{D}}$ is the extension of $\omega$ in matrix norm. Similar to Remark 4.6, plugging in scalar arguments in the algorithm, we recover the result from Tyurin & Richtárik (2024).

Following the same scheme as in Section 4, we choose $\boldsymbol{D} = \gamma_{\boldsymbol{W}} \cdot \boldsymbol{W}$, where $\boldsymbol{W} \in \mathbb{S}_{++}^d$. Thus, for a fixed $\boldsymbol{W}$, we relax the problem of finding the optimal stepsize to the problem of finding the optimal scaling factor $\gamma_{\boldsymbol{W}} > 0$.

**Corollary 5.3.** *For a fixed $\boldsymbol{W} \in \mathbb{S}_{++}^d$, the optimal scaling factor $\gamma_{\boldsymbol{W}} \in \mathbb{R}_+$ is given by*

$$\gamma_{\boldsymbol{W}} = \frac{2\lambda_{\boldsymbol{W}}}{1 + \sqrt{1 + 16 C_{\boldsymbol{W}} \lambda_{\min}(\boldsymbol{L}) \cdot \lambda_{\boldsymbol{W}}}},$$

*where $C_{\boldsymbol{W}} := \lambda_{\max}(\boldsymbol{W}) \cdot \omega_{\boldsymbol{W}} (4\omega_{\boldsymbol{W}} + 1)/n$ and $\lambda_{\boldsymbol{W}}$ is defined in Corollary 4.7.*

We observe that the structure of the optimal scaling factor for obtained above is similar to the one obtained in Corollary 4.7.

**The availability of $\boldsymbol{L}$:** For both algorithms, in order to determine the matrix stepsize, the knowledge of $\boldsymbol{L}$ is needed, if $\boldsymbol{L}$ is known, better complexities are guaranteed. When $\boldsymbol{L}$ is unknown, a closed-form solution can be obtained for generalized linear models. In more general cases, $\boldsymbol{L}_i$ can be treated as hyperparameters and estimated using first-order information via a gradient-based method (Wang et al., 2022). One can think of this as some type of preprocessing step, after which the matrices are learnt.

# 6 COMPLEXITIES OF THE ALGORITHMS

## 6.1 DET-MARINA

The following corollary formulates the iteration complexity for det-MARINA for $\boldsymbol{W} = \boldsymbol{L}^{-1}$.

**Corollary 6.1.** *If we take $\boldsymbol{W} = \boldsymbol{L}^{-1}$, then the condition (5) on $\gamma$ is given by*

$$\gamma \leq 2 \left( 1 + \sqrt{1 + 4\alpha\beta \cdot \Lambda_{\boldsymbol{L}^{-1}, \mathcal{S}}} \right)^{-1}. \tag{6}$$

*In order to obtain an $\varepsilon$-stationary point, that is, to satisfy $\mathbb{E}\left[ \left\| \nabla f(\tilde{x}^K) \right\|_{\frac{\boldsymbol{D}}{\det(\boldsymbol{D})^{1/d}}}^2 \right] \leq \varepsilon^2$, we require*

$$K \geq \mathcal{O}\left( \frac{\Delta_0 \cdot \det(\boldsymbol{L})^{\frac{1}{d}}}{\varepsilon^2} \cdot \left( 1 + \sqrt{1 + 4\alpha\beta \cdot \Lambda_{\boldsymbol{L}^{-1}, \mathcal{S}}} \right) \right),$$

*where $\Delta_0 := f(x^0) - f(x^\star)$. Moreover, this iteration complexity is always better than the one of MARINA.*

The proof can be found in the Appendix. In fact, we can show that in cases where we fix $\boldsymbol{W} = \boldsymbol{I}_d$ and $\boldsymbol{W} = \mathrm{diag}^{-1}(\boldsymbol{L})$, the same conclusion also holds, relevant details can be found in Appendix G.3. This essentially means that det-MARINA always has a "larger" stepsize compared to MARINA, even if the stepsize is suboptimal for the sake of efficiency, which leads to a better iteration complexity. In addition, since we are using the same compressor for those two algorithms, the communication complexity of det-MARINA is also provably better than that of MARINA.

In order to compute the communication complexity, we borrow the concept of expected density from Gorbunov et al. (2021).

**Definition 6.2.** For a given sketch matrix $\boldsymbol{S} \in \mathbb{S}_+^d$, the expected density is defined as

$$\zeta_{\boldsymbol{S}} = \sup_{x \in \mathbb{R}^d} \mathbb{E}[\|\boldsymbol{S}x\|_0],$$

where $\|x\|_0$ denotes the number of non-zero components of $x \in \mathbb{R}^d$.

In particular, we have $\zeta_{\mathrm{Rand}\text{-}\tau} = \tau$. Below, we state the communication complexity of det-MARINA with $\boldsymbol{W} = \boldsymbol{L}^{-1}$ and the Rand-$\tau$ compressor.

**Corollary 6.3.** *Assume that we are using sketch $\boldsymbol{S} \sim \mathcal{S}$ with expected density $\zeta_{\mathcal{S}}$. Suppose also we are running det-MARINA with probability $p$ and we use the optimal stepsize matrix with respect to $\boldsymbol{W} = \boldsymbol{L}^{-1}$. Then the overall communication complexity of the algorithm is given by $\mathcal{O}\big((Kp+1)d + (1-p)K\zeta_{\mathcal{S}}\big)$. Specifically, if we pick $p = \zeta_{\mathcal{S}}/d$, then the communication complexity is given by*

$$\mathcal{O}\left( d + \frac{\Delta_0 \det(\boldsymbol{L})^{\frac{1}{d}}}{\varepsilon^2} \left( \zeta_{\mathcal{S}} + \sqrt{\frac{\beta}{n} \Lambda_{\boldsymbol{L}^{-1}, \mathcal{S}} \zeta_{\mathcal{S}} (d - \zeta_{\mathcal{S}})} \right) \right).$$

Notice that in case where no compression is applied, the communication complexity reduces to $\mathcal{O}\left(d\Delta_0 \cdot \det(\boldsymbol{L})^{\frac{1}{d}}/\varepsilon^2\right)$. The latter coincides with the rate of matrix stepsize GD (see (Li et al., 2024)). Therefore, the dependence on $\varepsilon$ is not possible to improve further since GD is optimal among first order methods (Carmon et al., 2020).

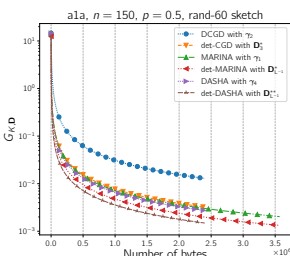 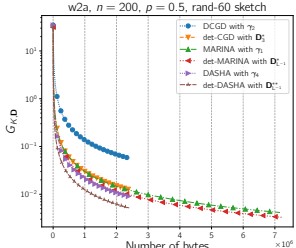 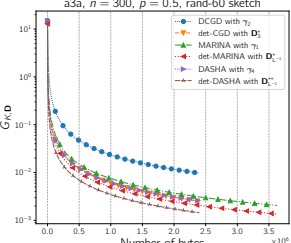

Figure 1: Comparison of DCGD with optimal stepsize, det-CGD with matrix stepsize $\boldsymbol{D}_3^*$, MARINA with optimal stepsize, DASHA with optimal scalar stepsize, det-MARINA with optimal stepsize $\boldsymbol{D}_{L^{-1}}^*$ and det-DASHA with optimal stepsize $\boldsymbol{D}_{L^{-1}}^{**}$. Throughout the experiment, we use Rand-$\tau$ sketch with $\tau = 60$. The $G_{K,\boldsymbol{D}}$ in the y-axis is defined in (51), which is the average squared matrix norm of the gradients.

## 6.2 DET-DASHA

The difference of compression mechanisms, does not allow us to have a direct comparison of the complexities of these algorithms. In particular, det-MARINA compresses the gradient difference with some probability $p$, while det-DASHA compresses the gradient difference with momentum in each iteration.

**Corollary 6.4.** *If we pick $\boldsymbol{D} = \gamma_{\boldsymbol{L}^{-1}} \cdot \boldsymbol{L}^{-1}$, then in order to reach an $\varepsilon^2$ stationary point, det-DASHA needs $K$ iterations with*

$$K \geq \frac{f(x^0) - f^\star}{\det(\boldsymbol{L})^{-\frac{1}{d}} \varepsilon^2}\left(1 + \sqrt{1 + 16C_{\boldsymbol{L}^{-1}}\lambda_{\min}\left(\boldsymbol{L}\right)}\right).$$

The following corollary compares the complexities of DASHA and det-DASHA. For the sake of brevity, we defer the complexities and other details to the proof of this corollary.

**Corollary 6.5.** *Suppose that the conditions in Theorem 5.1 hold, then compared to DASHA, det-DASHA with $\boldsymbol{W} = \boldsymbol{L}^{-1}$ always has a **better** iteration complexity, therefore, communication complexity as well.*

The following corollary suggests that the communication complexity of det-DASHA is better than that of det-MARINA,

**Corollary 6.6.** *The iteration complexity of det-MARINA with $p = 1/(\omega_{\boldsymbol{L}^{-1}}+1)$ and det-DASHA with momentum $1/(2\omega_{\boldsymbol{L}^{-1}}+1)$ is the same, therefore the communication complexity of det-DASHA is **better than** the communication complexity of det-MARINA.*

The resulting rates and communication complexities are summarized in Table 1 and Table 2, which provide a compact comparison of the considered methods under their respective assumptions.

## 7 EXPERIMENTS

We refer the readers to the appendix for more technical details of the experiments. Figure 1 shows that the performance in terms of communication complexity of det-DASHA and det-MARINA is better than their scalar counterpart DASHA and MARINA respectively. This validates the efficiency of using a matrix stepsize over a scalar stepsize. Further, det-DASHA and det-MARINA have better communication complexity in this case, compared to det-CGD. This demonstrates the effectiveness of applying variance reduction. Finally, as expected, det-DASHA has better communication complexity than det-MARINA.

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

## A  LLM USAGE

A language model was employed exclusively for grammar and word-choice refinement at the sentence level. It was not used for content generation, analysis, or any part of the research process.

Table 1: Assumptions and convergence rates. Abbreviations: Sm = smoothness, Interp = interpolation condition, Unb = unbiasedness, MatSm = matrix smoothness. $\zeta$ = expected transmitted coordinates.

| Method | Assumptions | Rate |
|---|---|---|
| DCGD | Sm+Interp+Unb | $\mathcal{O}\left(\frac{L\Delta^0}{K}\right)$ |
| det-CGD | MatSm+Interp+Unb | $\mathcal{O}\left(\frac{\det(\boldsymbol{L})^{1/d}\Delta^0}{K}\right)$ |
| MARINA | Sm+Unb | $\mathcal{O}\left(\frac{L\Delta^0\left(1+\sqrt{\frac{(1-p)\omega}{pn}}\right)}{K}\right)$ |
| DASHA | Sm+Unb | $\mathcal{O}\left(\frac{L\Delta^0\left(1+\frac{\omega}{\sqrt{n}}\right)}{K}\right)$ |
| det-MARINA | MatSm+Unb | $\mathcal{O}\left(\frac{\det(\boldsymbol{L})^{1/d}\Delta^0\left(1+\sqrt{1+4\alpha\beta\Lambda_{\boldsymbol{L}^{-1},S}}\right)}{K}\right)$ |
| det-DASHA | MatSm+Unb | $\mathcal{O}\left(\frac{\det(\boldsymbol{L})^{1/d}\Delta^0\left(1+\sqrt{1+16C_{\boldsymbol{L}^{-1}}\lambda_{\min}(\boldsymbol{L})}\right)}{K}\right)$ |

## B  NOTATIONS

The standard Euclidean norm on $\mathbb{R}^d$ is defined as $\|\cdot\|$. We use $\mathbb{S}^d_{++}$ (resp. $\mathbb{S}^d_+$) to denote the positive definite (resp. semi-definite) cone of dimension $d$. $\mathbb{S}^d$ is used to denote all symmetric matrices of dimension $d$. We use the notation $\boldsymbol{I}_d$ to denote the identity matrix of size $d \times d$, and $\boldsymbol{O}_d$ to denote the zero matrix of size $d \times d$. Given $\boldsymbol{Q} \in \mathbb{S}^d_{++}$ and $x \in \mathbb{R}^d$, $\|x\|_{\boldsymbol{Q}} := \sqrt{x^\top \boldsymbol{Q} x} = \sqrt{\langle x, \boldsymbol{Q} x \rangle}$, where $\langle \cdot, \cdot \rangle$ is the standard Euclidean inner product on $\mathbb{R}^d$. For a matrix $\boldsymbol{A} \in \mathbb{S}^d$, we use $\lambda_{\max}(\boldsymbol{A})$ (resp. $\lambda_{\min}(\boldsymbol{A})$) to denote the largest (resp. smallest) eigenvalue of the matrix $\boldsymbol{A}$. For a function $f : \mathbb{R}^d \mapsto \mathbb{R}$, its gradient and its Hessian at a point $x \in \mathbb{R}^d$ are respectively denoted as $\nabla f(x)$ and $\nabla^2 f(x)$. For the sketch matrices $\boldsymbol{S}_i^k$ used in the algorithm, we use the superscript $k$ to denote the iteration and subscript $i$ to denote the client, the matrix $\boldsymbol{S}_i^k$ is thus sampled for client $i$ in the $k$-th iteration from the same distribution $\mathcal{S}$. For any matrix $\boldsymbol{A} \in \mathbb{S}^d$, we use the notation $\mathrm{diag}(\boldsymbol{A}) \in \mathbb{S}^d$ to denote the diagonal of matrix $\boldsymbol{A}$.

## C  SUMMARY OF COMPLEXITIES

We present two compact tables to summarize the differences among the considered methods. Table 1 shows assumptions and convergence rates, while Table 2 lists communication complexities. The tables clarify the distinctions among CGD, det-CGD, MARINA, DASHA, det-MARINA, and det-DASHA.

## D  ADDITIONAL PRIOR WORK

**Non-convex Optimization.**  Numerous effective convex optimization techniques have been adapted for application in non-convex scenarios. Here's a selection of these techniques, although it's not an exhaustive list: adaptivity (Dvinskikh et al., 2019; Zhang et al., 2020b), variance reduction (J Reddi et al., 2016; Li et al., 2021), and acceleration (Guminov et al., 2019). Of particular relevance to our work is the paper by Khaled & Richtárik (2023), which introduces a unified approach for analyzing stochastic gradient descent for non-convex objectives. A comprehensive overview of non-convex optimization can be found in (Jain et al., 2017; Danilova et al., 2022).

**Matrix Stepsizes.**  An illustrative example of a matrix stepsized method is Newton's method, which has been a long-standing favorite in the optimization community (Gragg & Tapia, 1974; Miel, 1980; Yamamoto, 1987). However, the computational complexity involved in computing the stepsize as the inverse of the Hessian of the current iteration is substantial. An important direction of research that

Table 2: Communication complexities (same abbreviations as Table 1).

| Method | Assumptions | Communication Complexities |
|--------|-------------|---------------------------|
| DCGD | Sm+Interp+Unb | $\mathcal{O}\left(\frac{\zeta L \Delta^0}{\epsilon^2}\right)$ |
| det-CGD | MatSm+Interp+Unb | $\mathcal{O}\left(\frac{\zeta \det(\boldsymbol{L})^{1/d} \Delta^0}{\epsilon^2}\right)$ |
| MARINA | Sm+Unb | $\mathcal{O}\left(\frac{d + \zeta L \Delta^0 \left(1 + \sqrt{\frac{(1-p)\omega}{pn}}\right)}{\epsilon^2}\right)$ |
| DASHA | Sm+Unb | $\mathcal{O}\left(\frac{\zeta L \Delta^0 \left(1 + \frac{\omega}{\sqrt{n}}\right)}{\epsilon^2}\right)$ |
| det-MARINA | MatSm+Unb | $\mathcal{O}\left(\frac{d + \zeta \det(\boldsymbol{L})^{1/d} \Delta^0 \left(1 + \sqrt{\frac{\beta \Lambda_{\boldsymbol{L}^{-1},S}(d-\zeta)}{\zeta}}\right)}{\epsilon^2}\right)$ |
| det-DASHA | MatSm+Unb | $\mathcal{O}\left(\frac{\zeta \det(\boldsymbol{L})^{1/d} \Delta^0 \left(1 + \sqrt{1 + 16 C_{\boldsymbol{L}^{-1}} \lambda_{\min}(\boldsymbol{L})}\right)}{\epsilon^2}\right)$ |

is relevant to our work, studies distributed second order methods. Here is a non-exhaustive list of papers in this area: (Wang et al., 2018; Crane & Roosta, 2019; Zhang et al., 2020a; Islamov et al., 2021; Alimisis et al., 2021; Safaryan et al., 2022a).

**Distributed CGD.** The Distributed Compressed Gradient Descent (DCGD) algorithm, initially proposed by Khirirat et al. (2018), has seen improvements in various aspects, as documented in works such as (Li et al., 2020; Horváth et al., 2022). Its variance reduced version with gradients shifts was studied by Shulgin & Richtárik (2022) in the (strongly) convex setting. Additionally, there exists a substantial body of literature on other federated learning algorithms employing unbiased compressors (Alistarh et al., 2017; Mishchenko et al., 2019; Gorbunov et al., 2021; Mishchenko et al., 2022; Maranjyan et al., 2022; Horváth et al., 2023).

**Variance Reduction.** Variance reduction techniques have gained significant attention in the context of stochastic batch gradient descent that is prevalent in machine learning. Numerous algorithms have been developed in this regard, including well-known ones like SVRG (Johnson & Zhang, 2013), SAG (Schmidt et al., 2017), SDCA(Richtárik & Takáč, 2014), SAGA (Defazio et al., 2014), MISO (Mairal, 2015), and Katyusha (Allen-Zhu, 2017). An overview of more advanced methods can be found in (Gower et al., 2020). Notably, SVRG and Katyusha have been extended with loopless variants, namely L-SVRG and L-Katyusha (Kovalev et al., 2020; Qian et al., 2021). These loopless versions streamline the algorithms by eliminating the outer loop and introducing a biased coin-flip mechanism at each step. This simplification eases both the algorithms' structure and their analyses, while preserving their worst-case complexity bounds. L-SVRG, in particular, offers the advantage of setting the exit probability from the outer loop independently of the condition number, thus, enhancing both robustness and practical efficiency.

This technique of coin flipping allows to obtain variance reduction for the CGD algorithm. A relevant example is the DIANA algorithm proposed by Mishchenko et al. (2019). Its convergence was proved both in the convex and non-convex cases. Later, MARINA (Gorbunov et al., 2021) obtained the optimal convergence rate, improving in communication complexity compared to all previous first order methods. Finally, there is a line of work developing variance reduction in the federated setting using other methods and techniques (Chraibi et al., 2019; Hanzely & Richtárik, 2020; Dinh et al., 2020; Peng et al., 2022).

Another method to obtain variance reduction is based on momentum. It was initially studied by Cutkosky & Orabona (2019), where they propose the STORM algorithm, which is a stochastic gradient descent algorithm with a momentum term for non-convex objectives. They obtain station-

arity guarantees using adaptive stepsizes with optimal convergence rates. However, they require the variance of the stochastic gradient to be bounded by a constant, which is impractical. Using momentum for variance reduction has since been widely studied (Liu et al., 2020; Khanduri et al., 2020; Tran-Dinh et al., 2022; Li et al., 2022).

# E   BASIC FACTS

**Fact E.1.** *For two matrices $A, B \in \mathbb{S}_+^d$, denote the $i$-th largest eigenvalues of $A, B$ as $\lambda_i(A), \lambda_i(B)$. If $A \succeq B$, then $\lambda_i(A) \geq \lambda_i(B)$.*

*Proof.* According to the Courant-Fischer theorem, we have

$$\lambda_i(B) = \max_{S:\dim S=i} \min_{x \in S\setminus\{0\}} \frac{x^\top B x}{x^\top x}.$$

Let $S_{\max}^i$ be a subspace of dimension $i$ where the maximum is attained,

$$\lambda_i(B) = \min_{x \in S_{\max}^i \setminus \{0\}} \frac{x^\top B x}{x^\top x} \leq \min_{x \in S_{\max}^i \setminus \{0\}} \frac{x^\top A x}{x^\top x} \leq \max_{S:\dim S=i} \min_{x \in S\setminus\{0\}} \frac{x^\top A x}{x^\top x} = \lambda_i(A).$$

$\square$

**Fact E.2.** *Given a matrix $M \in \mathbb{S}_{++}^d$, a vector $c \in \mathbb{R}^d$, and a random vector $x \in \mathbb{R}^d$ such that $\mathbb{E}[\|x\|] \leq +\infty$, we have $\mathbb{E}\left[\|x - \mathbb{E}[x]\|_M^2\right] = \mathbb{E}\left[\|x - c\|_M^2\right] - \|\mathbb{E}[x] - c\|_M^2$.*

*Proof.*

$$\mathbb{E}\left[\|x - c\|_M^2\right] - \|\mathbb{E}[x] - c\|_M^2$$
$$= \mathbb{E}[x^\top M x] - 2\mathbb{E}[x]^\top M c + c^\top M c - \mathbb{E}[x]^\top M \mathbb{E}[x] + 2\mathbb{E}[x]^\top M c - c^\top M c$$
$$= \mathbb{E}[x^\top M x] - \mathbb{E}[x]^\top M \mathbb{E}[x]$$
$$= \mathbb{E}[x^\top M x] - 2 \cdot \mathbb{E}[x]^\top M \mathbb{E}[x] + \mathbb{E}[x]^\top M \mathbb{E}[x]$$
$$= \mathbb{E}\left[\|x - \mathbb{E}[x]\|_M^2\right].$$

$\square$

**Fact E.3.** *The mapping $(A, B, X) \mapsto A - X B^{-1} X$ is jointly concave on $\mathbb{S}_+^d \times \mathbb{S}_{++}^d \times \mathbb{S}^d$. It is also monotone increasing in variables $A$ and $B$.*

*Proof.* We refer the reader to Corollary 1.5.3 of Bhatia (2009) for the proof. $\square$

**Fact E.4.** *Suppose $L_i \in \mathbb{S}_{++}^d$, for $i = 1, \ldots, n$. Then, for every matrix $X \in \mathbb{S}_{++}^d$, the following mapping*

$$f(X, L_1, \ldots, L_n) = \frac{1}{n} \sum_{i=1}^n \lambda_{\max}(L_i) \cdot \lambda_{\max}\left(L_i X^{-1}\right) \cdot \lambda_{\max}\left(X^{-1}\right),$$

*is monotone decreasing in $X$.*

*Proof.* Fact E.3 suggests the mapping $X \mapsto X^{-1}$ is monotone decreasing which means that if we have two matrices $X_1, X_2 \in \mathbb{S}_{++}^d$ such that $X_1 \succeq X_2$, then $X_1^{-1} \preceq X_2^{-1}$. This leads to $0 < \lambda_{\max}(X_1^{-1}) \leq \lambda_{\max}(X_2^{-1})$ due to Fact E.1. Since $\lambda_{\max}(L_i X^{-1}) = \lambda_{\max}(L_i^{1/2} X^{-1} L_i^{1/2}) = \lambda_{\max}(X^{-1} L_i)$, and since the mapping $X \mapsto L_i^{1/2} X^{-1} L_i^{1/2}$ is monotone decreasing for every $i \in [n]$, we obtain $0 < \lambda_{\max}\left(L_i X_1^{-1}\right) \leq \lambda_{\max}\left(L_i X_2^{-1}\right)$. Notice that $\lambda_{\max}\left(L_i\right) > 0$, which indicates $f(X_1, L_1, \ldots, L_n) \leq f(X_2, L_1, \ldots, L_n)$. As a result, $f$ is monotone decreasing in $X$. $\square$

**Fact E.5.** *For any two matrices $\boldsymbol{A}, \boldsymbol{B} \in \mathbb{S}_{++}^d$, we have $\lambda_{\max}(\boldsymbol{AB}) \leq \lambda_{\max}(\boldsymbol{A}) \cdot \lambda_{\max}(\boldsymbol{B})$.*

*Proof.* Using the Courant-Fischer theorem, we can write

$$
\begin{aligned}
\lambda_{\max}(\boldsymbol{AB}) = \min_{S:\dim S=d} \max_{x \in S \setminus \{0\}} \frac{x^\top \boldsymbol{AB} x}{x^\top x} &= \max_{x \in \mathbb{R}^d \setminus \{0\}} \frac{x^\top \boldsymbol{AB} x}{x^\top x} \\
&\leq \max_{x \in \mathbb{R}^d \setminus \{0\}} \frac{x^\top \boldsymbol{A} x}{x^\top x} \cdot \max_{x \in \mathbb{R}^d \setminus \{0\}} \frac{x^\top \boldsymbol{B} x}{x^\top x} \\
&= \lambda_{\max}(\boldsymbol{A}) \cdot \lambda_{\max}(\boldsymbol{B}).
\end{aligned}
$$

$\square$

# F  PROPERTIES OF MATRIX SMOOTHNESS

## F.1  THE MATRIX LIPSCHITZ-CONTINUOUS GRADIENT

In this section, we describe the properties of matrix smoothness, matrix gradient Lipschitzness, and their relationships. The following lemma describes a sufficient condition for the matrix Lipschitz-continuity of the gradient.

**Lemma F.1.** *Given twice continuously differentiable function $f : \mathbb{R}^d \mapsto \mathbb{R}$ with uniformly bounded Hessian $\nabla^2 f(x) \preceq \boldsymbol{L}$, where $\boldsymbol{L} \in \mathbb{S}_{++}^d$. Then $f$ satisfies Definition 3.2 (Matrix Lipschitz Gradient) with the matrix $\boldsymbol{L}$.*

The following lemma is a variant of Lemma 3.4, which characterizes the smoothness matrix of the objective function $f$, given the smoothness matrices of the component functions $f_i$.

**Lemma F.2.** *Assume that $f_i$ has $\boldsymbol{L}_i$-Lipschitz continuous gradient for every $i \in [n]$, then function $f$ has $\boldsymbol{L}$-Lipschitz gradient with $\boldsymbol{L} \in \mathbb{S}_{++}^d$ satisfying*

$$
\boldsymbol{L} \cdot \lambda_{\min}(\boldsymbol{L}) = \frac{1}{n} \sum_{i=1}^n \lambda_{\max}(\boldsymbol{L}_i) \cdot \boldsymbol{L}_i. \tag{7}
$$

## F.2  QUADRATICS

**Lemma F.3.** *Consider the quadratic function $f(x) = \frac{1}{2} x^\top \boldsymbol{A} x + b^\top x + c$, where $\boldsymbol{A} \in \mathbb{S}_{++}^d, b \in \mathbb{R}^d, c \in \mathbb{R}$. Then $f$ has $\boldsymbol{A}$ matrix Lipschitz gradient.*

For a more general setting, consider the following $f$:,

$$
f(x) = \sum_{i=1}^s \phi_i(\boldsymbol{M}_i x),
$$

where $\boldsymbol{M}_i \in \mathbb{R}^{q_i \times d}$. Here $f : \mathbb{R}^d \mapsto \mathbb{R}$ is the sum of functions $\phi_i : \mathbb{R}^{q_i} \mapsto \mathbb{R}$. We have the following lemma regarding the matrix gradient Lipschitzness of $f$.

**Lemma F.4.** *Assume that functions $f$ and $\{\phi_i\}_{i=1}^s$ are defined above. If each function $\phi_i$ satisfies Assumption 3.3 (Matrix Lipschitz Gradient) with $\boldsymbol{L}_i$. Then function $f$ has $\boldsymbol{L}$-Lipschitz gradient, if $\sum_{i=1}^s \lambda_{\max}\left(\boldsymbol{L}_i^{\frac{1}{2}} \boldsymbol{M}_i \boldsymbol{L}^{-1} \boldsymbol{M}_i^\top \boldsymbol{L}_i^{\frac{1}{2}}\right) = 1$.*

Note that Lemma F.4 is a generalization of the previous case of quadratics, if we pick $s = 1$, $\boldsymbol{M}_i = \boldsymbol{A}^{\frac{1}{2}}$ and $\phi_1(x) = x^\top \boldsymbol{I}_d x$, the condition becomes $\boldsymbol{L} = \boldsymbol{A}$, which recovers Lemma F.3. In Lemma F.4, we only intend to give a way of finding a matrix $\boldsymbol{L} \in \mathbb{S}_{++}^d$, so that $f$ has $\boldsymbol{L}$-Lipschitz gradient. This does not mean, however, the $\boldsymbol{L}$ here is optimal.

## F.3  RELATION TO MATRIX SMOOTHNESS

Let us recall the definition of matrix smoothness.

**Definition F.5.** (**$L$**-smoothness) Assume that $f : \mathbb{R}^d \to \mathbb{R}$ is a continuously differentiable function and matrix $\boldsymbol{L} \in \mathbb{S}_{++}^d$. We say that $f$ is $\boldsymbol{L}$-smooth if for all $x, y \in \mathbb{R}^d$

$$f(y) \leq f(x) + \langle \nabla f(x), x - y \rangle + \frac{1}{2} \|x - y\|_{\boldsymbol{L}}^2. \tag{8}$$

We provide a lemma that offers an equivalent formulation for stating the $\boldsymbol{L}$-matrix smoothness of a function $f$.

**Lemma F.6.** *Let function $f : \mathbb{R}^d \to \mathbb{R}$ be continuously differentiable. Then the following statements are equivalent: (i) $f$ is $\boldsymbol{L}$-matrix smooth. (ii) $\langle \nabla f(x) - \nabla f(y), x - y \rangle \leq \|x - y\|_{\boldsymbol{L}}^2$ for all $x, y \in \mathbb{R}^d$.*

The two lemmas formulated below illustrate the relationship between matrix smoothness of $f$ and matrix gradient Lipschitzness of $f$.

**Lemma F.7.** *Assume $f : \mathbb{R}^d \mapsto \mathbb{R}$ is a continuously differentiable function, and its gradient is $\boldsymbol{L}$-Lipschitz continuous with $\boldsymbol{L} \in \mathbb{S}_{++}^d$. Then function $f$ is $\boldsymbol{L}$-matrix smooth.*

**Lemma F.8.** *Assume $f : \mathbb{R}^d \to \mathbb{R}$ is a continuously differentiable function. Assume also that $f$ is convex and $\boldsymbol{L}$-matrix smooth. Then $\nabla f$ is $\boldsymbol{L}$-Lipschitz continuous.*

The next proposition shows that standard Lipschitzness of the gradient of a function is an immediate consequence of matrix Lipschitzness.

**Lemma F.9.** *Assume that the gradient of $f$ is $\boldsymbol{L}$-Lipschitz continuous. Then $\nabla f$ is also $L$-Lipschitz with $L = \lambda_{\max}(\boldsymbol{L})$.*

### F.4 PROOF OF LEMMA 3.4

For any $x, y \in \mathbb{R}^d$,

$$\|\nabla f(x) - \nabla f(y)\|_{\boldsymbol{L}^{-1}}^2 = \left\| \frac{1}{n} \sum_{i=1}^n (\nabla f_i(x) - \nabla f_i(y)) \right\|_{\boldsymbol{L}^{-1}}^2 \leq \frac{1}{n} \sum_{i=1}^n \|\nabla f_i(x) - \nabla f_i(y)\|_{\boldsymbol{L}^{-1}}^2,$$

where the last inequality follows from convexity. Rewriting $\boldsymbol{L}^{-1}$ as $\boldsymbol{L}_i^{-1/2} \boldsymbol{L}_i^{1/2} \boldsymbol{L}^{-1} \boldsymbol{L}_i^{1/2} \boldsymbol{L}_i^{-1/2}$,

$$\|\nabla f(x) - \nabla f(y)\|_{\boldsymbol{L}^{-1}}^2$$

$$= \frac{1}{n} \sum_{i=1}^n \left( \boldsymbol{L}_i^{-\frac{1}{2}} (\nabla f_i(x) - \nabla f_i(y)) \right)^\top \boldsymbol{L}_i^{\frac{1}{2}} \boldsymbol{L}^{-1} \boldsymbol{L}_i^{\frac{1}{2}} \left( \boldsymbol{L}_i^{-\frac{1}{2}} (\nabla f_i(x) - \nabla f_i(y)) \right)$$

$$\leq \frac{1}{n} \sum_{i=1}^n \lambda_{\max} \left( \boldsymbol{L}_i^{\frac{1}{2}} \boldsymbol{L}^{-1} \boldsymbol{L}_i^{\frac{1}{2}} \right) \left\| \boldsymbol{L}_i^{-\frac{1}{2}} (\nabla f_i(x) - \nabla f_i(y)) \right\|^2$$

$$= \frac{1}{n} \sum_{i=1}^n \lambda_{\max} \left( \boldsymbol{L}_i^{\frac{1}{2}} \boldsymbol{L}^{-1} \boldsymbol{L}_i^{\frac{1}{2}} \right) \|\nabla f_i(x) - \nabla f_i(y)\|_{\boldsymbol{L}_i^{-1}}^2 \leq \frac{1}{n} \sum_{i=1}^n \lambda_{\max} \left( \boldsymbol{L}_i^{\frac{1}{2}} \boldsymbol{L}^{-1} \boldsymbol{L}_i^{\frac{1}{2}} \right) \|x - y\|_{\boldsymbol{L}_i}^2,$$

where the last inequality follows from Lipschitzness of the gradient of $f_i$. Rewriting $\boldsymbol{L}_i^{-1}$ as $\boldsymbol{L}^{-1/2} \boldsymbol{L}^{1/2} \boldsymbol{L}_i^{-1} \boldsymbol{L}^{1/2} \boldsymbol{L}^{-1/2}$, we obtain

$$\|\nabla f(x) - \nabla f(y)\|_{\boldsymbol{L}^{-1}}^2$$

$$= \frac{1}{n} \sum_{i=1}^n \lambda_{\max} \left( \boldsymbol{L}_i^{\frac{1}{2}} \boldsymbol{L}^{-1} \boldsymbol{L}_i^{\frac{1}{2}} \right) \cdot \left[ (\boldsymbol{L}^{\frac{1}{2}} (x - y))^\top \boldsymbol{L}^{-\frac{1}{2}} \boldsymbol{L}_i \boldsymbol{L}^{-\frac{1}{2}} (\boldsymbol{L}^{\frac{1}{2}} (x - y)) \right]$$

$$\leq \frac{1}{n} \sum_{i=1}^n \lambda_{\max} \left( \boldsymbol{L}_i^{\frac{1}{2}} \boldsymbol{L}^{-1} \boldsymbol{L}_i^{\frac{1}{2}} \right) \cdot \lambda_{\max} \left( \boldsymbol{L}^{-\frac{1}{2}} \boldsymbol{L}_i \boldsymbol{L}^{-\frac{1}{2}} \right) \left\| \boldsymbol{L}^{\frac{1}{2}} (x - y) \right\|^2$$

$$\overset{\text{Fact E.5}}{\leq} \left( \frac{1}{n} \sum_{i=1}^n \lambda_{\max} \left( \boldsymbol{L}^{-1} \right) \cdot \lambda_{\max} \left( \boldsymbol{L}_i \right) \cdot \lambda_{\max} \left( \boldsymbol{L}_i \boldsymbol{L}^{-1} \right) \right) \cdot \|x - y\|_{\boldsymbol{L}}^2 = \|x - y\|_{\boldsymbol{L}}^2.$$

## F.5  PROOF OF LEMMA F.1

For any $x, y \in \mathbb{R}^d$, we have

$$\|\nabla f(x) - \nabla f(y)\|_{\boldsymbol{L}^{-1}}^2$$

$$= \left\| \int_0^1 \nabla^2 f(\theta x + (1-\theta)y)(x-y)\, \mathrm{d}\theta \right\|_{\boldsymbol{L}^{-1}}^2$$

$$= (x-y)^\top \left( \int_0^1 \nabla^2 f(\theta x + (1-\theta)y)\, \mathrm{d}\theta \right)^\top \boldsymbol{L}^{-1} \left( \int_0^1 \nabla^2 f(\theta x + (1-\theta)y)\, \mathrm{d}\theta \right)(x-y).$$

Define $\boldsymbol{F} := \int_0^1 \nabla^2 f(\theta x + (1-\theta)y)\, \mathrm{d}\theta$, notice that $\boldsymbol{F}$ is a symmetric matrix. Then.

$$\|\nabla f(x) - \nabla f(y)\|_{\boldsymbol{L}^{-1}}^2 = (x-y)^\top \boldsymbol{F}^\top \boldsymbol{L}^{-1} \boldsymbol{F} (x-y).$$

Since $\boldsymbol{L}$ is an uniform upper bound of the Hessian, we have $\boldsymbol{F} \preceq \boldsymbol{L}$. which turns out to be equivalent to $\boldsymbol{F}\boldsymbol{L}^{-1}\boldsymbol{F} \preceq \boldsymbol{L}$, as

$$\begin{aligned}
\boldsymbol{F}\boldsymbol{L}^{-1}\boldsymbol{F} \preceq \boldsymbol{L} &\iff \boldsymbol{L}^{-\frac{1}{2}}\boldsymbol{F}\boldsymbol{L}\boldsymbol{F}\boldsymbol{L}^{-\frac{1}{2}} \preceq \boldsymbol{I}_d \\
&\iff \boldsymbol{L}^{-\frac{1}{2}}\boldsymbol{F}\boldsymbol{L}^{-\frac{1}{2}} \cdot \boldsymbol{L}^{-\frac{1}{2}}\boldsymbol{F}\boldsymbol{L}^{-\frac{1}{2}} \preceq \boldsymbol{I}_d \\
&\iff \boldsymbol{L}^{-\frac{1}{2}}\boldsymbol{F}\boldsymbol{L}^{-\frac{1}{2}} \preceq \boldsymbol{I}_d \\
&\iff \boldsymbol{F} \preceq \boldsymbol{L}.
\end{aligned}$$

Thus,

$$\|\nabla f(x) - \nabla f(y)\|_{\boldsymbol{L}^{-1}}^2 \leq (x-y)^\top \boldsymbol{L}(x-y) = \|x-y\|_{\boldsymbol{L}}^2.$$

## F.6  PROOF OF LEMMA F.2

Suppose $\boldsymbol{L}$ is a symmetric positive definite matrix satisfying (7). Let us now show that the function $\nabla f$ is $\boldsymbol{L}$-Lipschitz continuous. Picking any two points $x, y \in \mathbb{R}^d$, we have:

$$\|\nabla f(x) - \nabla f(y)\|_{\boldsymbol{L}^{-1}}^2 = \left\| \frac{1}{n} \sum_{i=1}^n (\nabla f_i(x) - \nabla f_i(y)) \right\|_{\boldsymbol{L}^{-1}}^2 \leq \frac{1}{n} \sum_{i=1}^n \|\nabla f_i(x) - \nabla f_i(y)\|_{\boldsymbol{L}^{-1}}^2.$$

Rewriting $\boldsymbol{L}^{-1}$ as $\boldsymbol{L}_i^{-\frac{1}{2}}\boldsymbol{L}_i^{\frac{1}{2}}\boldsymbol{L}^{-1}\boldsymbol{L}_i^{\frac{1}{2}}\boldsymbol{L}_i^{-\frac{1}{2}}$,

$$\begin{aligned}
\|\nabla f(x) - \nabla f(y)\|_{\boldsymbol{L}^{-1}}^2 &\leq \frac{1}{n} \sum_{i=1}^n (\nabla f_i(x) - \nabla f_i(y))^\top \boldsymbol{L}_i^{-\frac{1}{2}}\boldsymbol{L}_i^{\frac{1}{2}}\boldsymbol{L}^{-1}\boldsymbol{L}_i^{\frac{1}{2}}\boldsymbol{L}_i^{-\frac{1}{2}} (\nabla f_i(x) - \nabla f_i(y)) \\
&\leq \frac{1}{n} \sum_{i=1}^n \lambda_{\max}(\boldsymbol{L}_i) \cdot \lambda_{\max}(\boldsymbol{L}^{-1}) \cdot \|\nabla f_i(x) - \nabla f_i(y)\|_{\boldsymbol{L}_i^{-1}}^2 \\
&\leq \frac{1}{n} \sum_{i=1}^n \lambda_{\max}(\boldsymbol{L}_i) \cdot \lambda_{\max}(\boldsymbol{L}^{-1}) \cdot \|x-y\|_{\boldsymbol{L}_i}^2 \\
&= \|x-y\|_{\lambda_{\max}(\boldsymbol{L}^{-1}) \cdot \frac{1}{n} \sum_{i=1}^n \lambda_{\max}(\boldsymbol{L}_i) \cdot \boldsymbol{L}_i}^2 \overset{(7)}{=} \|x-y\|_{\boldsymbol{L}}^2.
\end{aligned}$$

## F.7  PROOF OF LEMMA F.3

According to Definition 3.2, $\boldsymbol{L}$ must satisfy:

$$\sqrt{(x-y)^\top \boldsymbol{A}\boldsymbol{L}^{-1}\boldsymbol{A}(x-y)} \leq \sqrt{(x-y)^\top \boldsymbol{L}(x-y)},$$

for any $x, y \in \mathbb{R}^d$, which is $\boldsymbol{A}\boldsymbol{L}^{-1}\boldsymbol{A} \preceq \boldsymbol{L}$. Since $\boldsymbol{A} \in \mathbb{S}_{++}^d$, we further simplify the condition to $\boldsymbol{A} \preceq \boldsymbol{L}$. Therefore, the "best" $\boldsymbol{L} \in \mathbb{S}_{++}^d$ that satisfies (2) is $\boldsymbol{L} = \boldsymbol{A}$.

## F.8 PROOF OF LEMMA F.4

For any $x, y \in \mathbb{R}^d$, we have

$$\|\nabla f(x) - \nabla f(y)\|_{\boldsymbol{L}^{-1}} = \left\| \sum_{i=1}^s \boldsymbol{M}_i^\top \nabla \phi_i(\boldsymbol{M}_i x) - \sum_{i=1}^s \boldsymbol{M}_i^\top \nabla \phi_i(\boldsymbol{M}_i y) \right\|_{\boldsymbol{L}^{-1}}$$

$$= s \cdot \left\| \frac{1}{s} \sum_{i=1}^s \boldsymbol{M}_i^\top \left( \nabla \phi_i(\boldsymbol{M}_i x) - \nabla \phi_i\left(\boldsymbol{M}_i y\right) \right) \right\|_{\boldsymbol{L}^{-1}}$$

$$= s \cdot \frac{1}{s} \sum_{i=1}^s \left\| \boldsymbol{M}_i^\top \left( \nabla \phi_i(\boldsymbol{M}_i x) - \nabla \phi_i(\boldsymbol{M}_i y) \right) \right\|_{\boldsymbol{L}^{-1}},$$

where the last inequality follows from the convexity. Thus,

$$\|\nabla f(x) - \nabla f(y)\|_{\boldsymbol{L}^{-1}}$$
$$= \sum_{i=1}^s \sqrt{\left( \nabla \phi_i(\boldsymbol{M}_i x) - \nabla \phi_i(\boldsymbol{M}_i y) \right)^\top \boldsymbol{M}_i \boldsymbol{L}^{-1} \boldsymbol{M}_i^\top \left( \nabla \phi_i(\boldsymbol{M}_i x) - \nabla \phi_i(\boldsymbol{M}_i y) \right)}$$
$$= \sum_{i=1}^s \sqrt{\boldsymbol{B}_i^\top \boldsymbol{L}_i^{\frac{1}{2}} \boldsymbol{M}_i \boldsymbol{L}^{-1} \boldsymbol{M}_i^\top \boldsymbol{L}_i^{\frac{1}{2}} \boldsymbol{B}_i}$$
$$\leq \sum_{i=1}^s \sqrt{\lambda_{\max} \left( \boldsymbol{L}_i^{\frac{1}{2}} \boldsymbol{M}_i \boldsymbol{L}^{-1} \boldsymbol{M}_i^\top \boldsymbol{L}_i^{\frac{1}{2}} \right)} \cdot \|\nabla \phi_i(\boldsymbol{M}_i x) - \nabla \phi_i(\boldsymbol{M}_i y)\|_{\boldsymbol{L}_i^{-1}},$$

where $\boldsymbol{B}_i := \boldsymbol{L}_i^{-\frac{1}{2}} \left( \nabla \phi_i(\boldsymbol{M}_i x) - \nabla \phi_i(\boldsymbol{M}_i y) \right)$. Since $\phi_i$ is $\boldsymbol{L}_i$-Lipschitz, we have

$$\|\nabla f(x) - \nabla f(y)\|_{\boldsymbol{L}^{-1}}$$
$$\leq \sum_{i=1}^s \sqrt{\lambda_{\max} \left( \boldsymbol{L}_i^{\frac{1}{2}} \boldsymbol{M}_i \boldsymbol{L}^{-1} \boldsymbol{M}_i^\top \boldsymbol{L}_i^{\frac{1}{2}} \right)} \cdot \|\boldsymbol{M}_i(x - y)\|_{\boldsymbol{L}_i}$$
$$= \sum_{i=1}^s \sqrt{\lambda_{\max} \left( \boldsymbol{L}_i^{\frac{1}{2}} \boldsymbol{M}_i \boldsymbol{L}^{-1} \boldsymbol{M}_i^\top \boldsymbol{L}_i^{\frac{1}{2}} \right)} \cdot \sqrt{\left[ \boldsymbol{L}^{\frac{1}{2}} (x - y) \right]^\top \boldsymbol{L}^{-\frac{1}{2}} \boldsymbol{M}_i^\top \boldsymbol{L}_i \boldsymbol{M}_i \boldsymbol{L}^{-\frac{1}{2}} \left[ \boldsymbol{L}^{\frac{1}{2}} (x - y) \right]}$$
$$\leq \sum_{i=1}^s \sqrt{\lambda_{\max} \left( \boldsymbol{L}_i^{\frac{1}{2}} \boldsymbol{M}_i \boldsymbol{L}^{-1} \boldsymbol{M}_i^\top \boldsymbol{L}_i^{\frac{1}{2}} \right) \cdot \lambda_{\max} \left( \boldsymbol{L}^{-\frac{1}{2}} \boldsymbol{M}_i^\top \boldsymbol{L}_i \boldsymbol{M}_i \boldsymbol{L}^{-\frac{1}{2}} \right)} \cdot \|x - y\|_{\boldsymbol{L}}$$
$$= \sum_{i=1}^s \lambda_{\max} \left( \boldsymbol{L}_i^{\frac{1}{2}} \boldsymbol{M}_i \boldsymbol{L}^{-1} \boldsymbol{M}_i^\top \boldsymbol{L}_i^{\frac{1}{2}} \right) \cdot \|x - y\|_{\boldsymbol{L}},$$

where the last identity is due to $\lambda_{\max} \left( \boldsymbol{L}_i^{\frac{1}{2}} \boldsymbol{M}_i \boldsymbol{L}^{-1} \boldsymbol{M}_i^\top \boldsymbol{L}_i^{\frac{1}{2}} \right) = \lambda_{\max} \left( \boldsymbol{L}^{-\frac{1}{2}} \boldsymbol{M}_i^\top \boldsymbol{L}_i \boldsymbol{M}_i \boldsymbol{L}^{-\frac{1}{2}} \right)$. Since $\sum_{i=1}^s \lambda_{\max} \left( \boldsymbol{L}_i^{\frac{1}{2}} \boldsymbol{M}_i \boldsymbol{L}^{-1} \boldsymbol{M}_i^\top \boldsymbol{L}_i^{\frac{1}{2}} \right) = 1$, we have $\|\nabla f(x) - \nabla f(y)\|_{\boldsymbol{L}^{-1}} \leq \|x - y\|_{\boldsymbol{L}}$.

## F.9 PROOF OF LEMMA F.6

(i) $\rightarrow$ (ii). If $f$ is $\boldsymbol{L}$-matrix smooth. Then for all $x, y \in \mathbb{R}^d$, we have

$$f(x) \leq f(y) + \langle \nabla f(y), x - y \rangle + \frac{1}{2} \|x - y\|_{\boldsymbol{L}}^2,$$

$$f(y) \leq f(x) + \langle \nabla f(x), y - x \rangle + \frac{1}{2} \|x - y\|_{\boldsymbol{L}}^2.$$

Summing up these two inequalities we get

$$\langle \nabla f(x) - \nabla f(y), x - y \rangle \leq \|x - y\|_{\boldsymbol{L}}^2.$$

(ii) $\to$ (i). Choose any $x, y \in \mathbb{R}^d$ and define $z = x + t(y - x)$, we have,

$$
\begin{aligned}
f(y) &= f(x) + \int_0^1 \langle \nabla f(x + t(y - x)), y - x \rangle \, \mathrm{d}t \\
&= f(x) + \int_0^1 \langle \nabla f(z), y - x \rangle \, \mathrm{d}t \\
&= f(x) + \langle \nabla f(x), y - x \rangle + \int_0^1 \langle \nabla f(z) - \nabla f(x), y - x \rangle \, \mathrm{d}t \\
&= f(x) + \langle \nabla f(x), y - x \rangle + \int_0^1 \langle \nabla f(z) - \nabla f(x), z - x \rangle \cdot \frac{1}{t} \, \mathrm{d}t.
\end{aligned}
$$

For any $x, z \in \mathbb{R}^d$, we have

$$
\langle \nabla f(z) - \nabla f(x), z - x \rangle \leq \|z - x\|_{\boldsymbol{L}}^2 .
$$

As a result,

$$
\begin{aligned}
f(y) &\leq f(x) + \langle \nabla f(x), y - x \rangle + \int_0^1 \|z - x\|_{\boldsymbol{L}}^2 \cdot \frac{1}{t} \, \mathrm{d}t \\
&= f(x) + \langle \nabla f(x), y - x \rangle + \int_0^1 \|y - x\|_{\boldsymbol{L}}^2 \cdot t \, \mathrm{d}t \\
&= f(x) + \langle \nabla f(x), y - x \rangle + \frac{1}{2} \|y - x\|_{\boldsymbol{L}}^2 .
\end{aligned}
$$

### F.10 PROOF OF LEMMA F.7

We start with picking any two points $x, y \in \mathbb{R}^d$. Using Cauchy-Schwarz inequality, we have

$$
\langle \nabla f(x) - \nabla f(y), x - y \rangle \leq \|\nabla f(x) - \nabla f(y)\|_{\boldsymbol{L}^{-1}} \cdot \|x - y\|_{\boldsymbol{L}} \overset{(2)}{\leq} = \|x - y\|_{\boldsymbol{L}}^2 .
$$

According to Lemma F.6, this indicates that function $f$ is $\boldsymbol{L}$-matrix smooth.

### F.11 PROOF OF LEMMA F.8

By Lemma F.6, we have for any $x, y \in \mathbb{R}^d$,

$$
\langle \nabla f(x) - \nabla f(y), x - y \rangle \leq \|x - y\|_{\boldsymbol{L}}^2 . \tag{9}
$$

Now we pick any three points $x, y, z \in \mathbb{R}^d$. With the $\boldsymbol{L}$-smoothness of $f$, we have

$$
f(x + z) \geq f(x) + \langle \nabla f(x), z \rangle + \frac{1}{2} \|z\|_{\boldsymbol{L}}^2 . \tag{10}
$$

Using the convexity of $f$, we have

$$
\langle \nabla f(y), x + z - y \rangle \leq f(x + z) - f(y). \tag{11}
$$

Combining (10) and (11), we obtain

$$
\langle \nabla f(y), x + z - y \rangle \leq f(x) - f(y) + \langle \nabla f(x), z \rangle + \frac{1}{2} \|z\|_{\boldsymbol{L}}^2 .
$$

Rearranging terms we get

$$
\langle \nabla f(y) - \nabla f(x), z \rangle - \frac{1}{2} \|z\|_{\boldsymbol{L}}^2 \leq f(x) - f(y) - \langle \nabla f(y), x - y \rangle .
$$

The inequality holds for any $z$ for fixed $x$ and $y$, and the left hand side is maximized for $z$ when $z = \boldsymbol{L}^{-1} (\nabla f(y) - \nabla f(x))$. Plugging it in, we have

$$
\frac{1}{2} \|\nabla f(x) - \nabla f(y)\|_{\boldsymbol{L}^{-1}}^2 \leq f(x) - f(y) - \langle \nabla f(y), x - y \rangle . \tag{12}
$$

By symmetry, we also have

$$\frac{1}{2} \left\| \nabla f(y) - \nabla f(x) \right\|_{\boldsymbol{L}^{-1}}^2 \leq f(y) - f(x) - \langle \nabla f(x), y - x \rangle .$$

Adding (12) and its counterpart, we obtain

$$\left\| \nabla f(x) - \nabla f(y) \right\|_{\boldsymbol{L}^{-1}}^2 \leq \langle \nabla f(x) - \nabla f(y), x - y \rangle . \tag{13}$$

Combing (13) and (9), it follows

$$\left\| \nabla f(x) - \nabla f(y) \right\|_{\boldsymbol{L}^{-1}}^2 \leq \left\| x - y \right\|_{\boldsymbol{L}}^2 .$$

## F.12    PROOF OF LEMMA F.9

Pick any two points $x, y \in \mathbb{R}^d$. With the matrix $\boldsymbol{L}$-Lipschitzness of the gradient of function $f$, we have

$$\left\| \nabla f(x) - \nabla f(y) \right\|_{\boldsymbol{L}^{-1}}^2 \leq \left\| x - y \right\|_{\boldsymbol{L}}^2 .$$

This implies

$$(x - y)^\top \boldsymbol{L}(x - y) - (\nabla f(x) - \nabla f(y))^\top \boldsymbol{L}^{-1} (\nabla f(x) - \nabla f(y)) \geq 0.$$

Define function $f(\boldsymbol{X}) := a^\top \boldsymbol{X} a - b^\top \boldsymbol{X}^{-1} b$ for $\boldsymbol{X} \in \mathbb{S}_{++}^d$, where $a, b \in \mathbb{R}^d$ are fixed vectors. Then $f$ is monotone increasing in $\boldsymbol{X}$. This can be shown in the following way, picking two matrices $\boldsymbol{X}_1, \boldsymbol{X}_2 \in \mathbb{S}_{++}^d$, where $\boldsymbol{X}_1 \succeq \boldsymbol{X}_2$. We see that $-\boldsymbol{X}_1^{-1} \succeq -\boldsymbol{X}_2^{-1}$, since from Fact E.3 the map $\boldsymbol{X} \mapsto -\boldsymbol{X}^{-1}$ is monotone increasing for $\boldsymbol{X} \in \mathbb{S}_{++}^d$. Thus,

$$\begin{aligned} f(\boldsymbol{X}_1) - f(\boldsymbol{X}_2) &= (x - y)^\top (\boldsymbol{X}_1 - \boldsymbol{X}_2)(x - y) \\ &\quad + (\nabla f(x) - \nabla f(y))^\top \left( -\boldsymbol{X}_1^{-1} - (-\boldsymbol{X}_2^{-1}) \right) (\nabla f(x) - \nabla f(y)) \geq 0. \end{aligned}$$

As a result, $f(\lambda_{\max}(\boldsymbol{L}) \cdot \boldsymbol{I}_d) \geq f(\boldsymbol{L}) \geq 0$, due to the fact that $\lambda_{\max}(\boldsymbol{L}) \cdot \boldsymbol{I}_d \succeq \boldsymbol{L}$. It remains to notice that

$$f(\lambda_{\max}(\boldsymbol{L}) \cdot \boldsymbol{I}_d) = \lambda_{\max}(\boldsymbol{L}) \left\| x - y \right\|^2 - \frac{1}{\lambda_{\max}(\boldsymbol{L})} \left\| \nabla f(x) - \nabla f(y) \right\|^2 \geq 0,$$

which yields

$$\left\| \nabla f(x) - \nabla f(y) \right\|^2 \leq \lambda_{\max}^2(\boldsymbol{L}) \left\| x - y \right\|^2 .$$

# G    ANALYSIS OF DET-MARINA

## G.1    TECHNICAL LEMMAS

We first state some technical lemmas essential for the proof.

**Lemma G.1** (Descent lemma). *Assume function $f$ is $\boldsymbol{L}$ smooth, and $x^{k+1} = x^k - \boldsymbol{D} \cdot g^k$, where $\boldsymbol{D} \in \mathbb{S}_{++}^d$. Then the iterates generated by Algorithm 1 satisfy:*

$$f(x^{k+1}) \leq f(x^k) - \frac{1}{2} \left\| \nabla f(x^k) \right\|_{\boldsymbol{D}}^2 + \frac{1}{2} \left\| g^k - \nabla f(x^k) \right\|_{\boldsymbol{D}}^2 - \frac{1}{2} \left\| x^{k+1} - x^k \right\|_{\boldsymbol{D}^{-1} - \boldsymbol{L}} .$$

The following lemma is obtained for any sketch matrix $\boldsymbol{S} \in \mathbb{S}_+^d$ and any two positive definite matrices $\boldsymbol{D}$ and $\boldsymbol{L}$.

**Lemma G.2** (Property of sketch matrix). *For any sketch matrix $\boldsymbol{S} \in \mathbb{S}_+^d$, a vector $t \in \mathbb{R}^d$, and matrices $\boldsymbol{D}, \boldsymbol{L} \in \mathbb{S}_{++}^d$, we have*

$$\mathbb{E}\left[ \left\| \boldsymbol{S} t - t \right\|_{\boldsymbol{D}}^2 \right] \leq \lambda_{\max} \left( \boldsymbol{L}^{\frac{1}{2}} \left( \mathbb{E}[\boldsymbol{S} \boldsymbol{D} \boldsymbol{S}] - \boldsymbol{D} \right) \boldsymbol{L}^{\frac{1}{2}} \right) \cdot \left\| t \right\|_{\boldsymbol{L}^{-1}}^2 . \tag{14}$$

## G.2 PROOF OF THEOREM 4.1

According to Lemma G.1, we have

$$\mathbb{E}\big[f(x^{k+1})\big] \le \mathbb{E}\big[f(x^k)\big] - \mathbb{E}\left[\frac{1}{2}\left\|\nabla f(x^k)\right\|_{\boldsymbol{D}}^2\right] \tag{15}$$

$$+ \mathbb{E}\left[\frac{1}{2}\left\|g^k - \nabla f(x^k)\right\|_{\boldsymbol{D}}^2\right] - \mathbb{E}\left[\frac{1}{2}\left\|x^{k+1} - x^k\right\|_{\boldsymbol{D}^{-1}-\boldsymbol{L}}^2\right]. \tag{16}$$

Notice that,

$$g^{k+1} = \begin{cases} \nabla f(x^{k+1}) & \text{with probability } p, \\ g^k + \frac{1}{n}\sum_{i=1}^n \boldsymbol{S}_i^k\left(\nabla f_i(x^{k+1}) - \nabla f_i(x^k)\right) & \text{with probability } 1-p. \end{cases}$$

As a result, from the tower property,

$$\mathbb{E}\left[\left\|g^{k+1} - \nabla f(x^{k+1})\right\|_{\boldsymbol{D}}^2 \mid x^{k+1}, x^k\right]$$

$$= \mathbb{E}\left[\mathbb{E}\left[\left\|g^{k+1} - \nabla f(x^{k+1})\right\|_{\boldsymbol{D}}^2 \mid x^{k+1}, x^k, c_k\right]\right]$$

$$= (1-p) \cdot \mathbb{E}\left[\left\|g^k + \frac{1}{n}\sum_{i=1}^n \boldsymbol{S}_i^k(\nabla f_i(x^{k+1}) - \nabla f_i(x^k)) - \nabla f(x^{k+1})\right\|_{\boldsymbol{D}}^2 \mid x^{k+1}, x^k\right].$$

Using Fact E.2, we have

$$\mathbb{E}\left[\left\|g^{k+1} - \nabla f(x^{k+1})\right\|_{\boldsymbol{D}}^2 \mid x^{k+1}, x^k\right]$$

$$= (1-p) \cdot \mathbb{E}\left[\left\|\frac{1}{n}\sum_{i=1}^n \boldsymbol{S}_i^k(\nabla f_i(x^{k+1}) - \nabla f_i(x^k)) - \left(\nabla f(x^{k+1}) - \nabla f(x^k)\right)\right\|_{\boldsymbol{D}}^2 \mid x^{k+1}, x^k\right]$$

$$+ (1-p) \cdot \left\|g^k - \nabla f(x^k)\right\|_{\boldsymbol{D}}^2$$

$$= (1-p) \cdot \mathbb{E}\left[\left\|\frac{1}{n}\sum_{i=1}^n \left(\boldsymbol{S}_i^k(\nabla f_i(x^{k+1}) - \nabla f_i(x^k)) - \left(\nabla f_i(x^{k+1}) - \nabla f_i(x^k)\right)\right)\right\|_{\boldsymbol{D}}^2 \mid x^{k+1}, x^k\right]$$

$$+ (1-p) \cdot \left\|g^k - \nabla f(x^k)\right\|_{\boldsymbol{D}}^2.$$

Notice that the sketch matrix is unbiased, which implies

$$\mathbb{E}\big[\boldsymbol{S}_i^k\left(\nabla f_i(x^{k+1}) - \nabla f_i(x^k)\right) \mid x^{k+1}, x^k\big] = \nabla f_i(x^{k+1}) - \nabla f_i(x^k),$$

Since any two distinct random vectors in the set $\{\boldsymbol{S}_i^k(\nabla f_i(x^{k+1}) - \nabla f_i(x^k))\}_{i=1}^n$ are independent from each other, if $x^{k+1}$ and $x^k$ are fixed, we have

$$\mathbb{E}\left[\left\|g^{k+1} - \nabla f(x^{k+1})\right\|_{\boldsymbol{D}}^2 \mid x^{k+1}, x^k\right]$$

$$= \frac{1-p}{n^2}\sum_{i=1}^n \mathbb{E}\left[\left\|\boldsymbol{S}_i^k(\nabla f_i(x^{k+1}) - \nabla f_i(x^k)) - \left(\nabla f_i(x^{k+1}) - \nabla f_i(x^k)\right)\right\|_{\boldsymbol{D}}^2 \mid x^{k+1}, x^k\right]$$

$$+ (1-p) \cdot \left\|g^k - \nabla f(x^k)\right\|_{\boldsymbol{D}}^2. \tag{17}$$

Applying Lemma G.2, we obtain

$$\mathbb{E}\left[\left\|\boldsymbol{S}_i^k(\nabla f_i(x^{k+1}) - \nabla f_i(x^k)) - \left(\nabla f_i(x^{k+1}) - \nabla f_i(x^k)\right)\right\|_{\boldsymbol{D}}^2 \mid x^{k+1}, x^k\right]$$

$$\le \lambda_{\max}\left(\boldsymbol{L}_i^{\frac{1}{2}}\left(\mathbb{E}\big[\boldsymbol{S}_i^k \boldsymbol{D}\boldsymbol{S}_i^k\big] - \boldsymbol{D}\right)\boldsymbol{L}_i^{\frac{1}{2}}\right)\left\|\nabla f_i(x^{k+1}) - \nabla f_i(x^k)\right\|_{\boldsymbol{L}_i^{-1}}^2.$$

Using the fact that $f_i$ has $\boldsymbol{L}_i$-Lipschitz gradient, we have

$$\mathbb{E}\left[\left\|\boldsymbol{S}_i^k(\nabla f_i(x^{k+1}) - \nabla f_i(x^k)) - \left(\nabla f_i(x^{k+1}) - \nabla f_i(x^k)\right)\right\|_{\boldsymbol{D}}^2 \mid x^{k+1}, x^k\right]$$

$$\le \lambda_{\max}\left(\boldsymbol{L}_i^{\frac{1}{2}}\left(\mathbb{E}\big[\boldsymbol{S}_i^k \boldsymbol{D}\boldsymbol{S}_i^k\big] - \boldsymbol{D}\right)\boldsymbol{L}_i^{\frac{1}{2}}\right)\left\|x^{k+1} - x^k\right\|_{\boldsymbol{L}_i}^2. \tag{18}$$

Plugging (18) into (17), we deduce

$$\mathbb{E}\left[\left\|g^{k+1} - \nabla f(x^{k+1})\right\|_{\boldsymbol{D}}^2 \mid x^{k+1}, x^k\right]$$

$$\leq \frac{1-p}{n^2} \sum_{i=1}^n \lambda_{\max}\left(\boldsymbol{L}_i^{\frac{1}{2}}\left(\mathbb{E}\left[\boldsymbol{S}_i^k \boldsymbol{D} \boldsymbol{S}_i^k\right] - \boldsymbol{D}\right)\boldsymbol{L}_i^{\frac{1}{2}}\right)\left\|x^{k+1} - x^k\right\|_{\boldsymbol{L}_i}^2 + (1-p)\cdot\left\|g^k - \nabla f(x^k)\right\|_{\boldsymbol{D}}^2.$$

Rewriting $\boldsymbol{L}_i^{-1}$ and denote $\lambda_i := \lambda_{\max}\left(\boldsymbol{L}_i^{\frac{1}{2}}\left(\mathbb{E}\left[\boldsymbol{S}_i^k \boldsymbol{D} \boldsymbol{S}_i^k\right] - \boldsymbol{D}\right)\boldsymbol{L}_i^{\frac{1}{2}}\right)$.

$$\mathbb{E}\left[\left\|g^{k+1} - \nabla f(x^{k+1})\right\|_{\boldsymbol{D}}^2 \mid x^{k+1}, x^k\right]$$

$$= \frac{1-p}{n^2} \sum_{i=1}^n \lambda_i \cdot \left(\boldsymbol{L}^{\frac{1}{2}}(x^{k+1} - x^k)\right)^\top \boldsymbol{L}^{-\frac{1}{2}} \boldsymbol{L}_i \boldsymbol{L}^{-\frac{1}{2}} \left(\boldsymbol{L}^{\frac{1}{2}}(x^{k+1} - x^k)\right)$$

$$+ (1-p)\left\|g^k - \nabla f(x^k)\right\|_{\boldsymbol{D}}^2$$

$$\leq \frac{1-p}{n^2} \sum_{i=1}^n \lambda_i \cdot \lambda_{\max}\left(\boldsymbol{L}^{-\frac{1}{2}} \boldsymbol{L}_i \boldsymbol{L}^{-\frac{1}{2}}\right)\left\|x^{k+1} - x^k\right\|_{\boldsymbol{L}}^2 + (1-p)\cdot\left\|g^k - \nabla f(x^k)\right\|_{\boldsymbol{D}}^2.$$

We further use Fact E.5 to upper bound $\lambda_{\max}\left(\boldsymbol{L}_i^{\frac{1}{2}}\left(\mathbb{E}\left[\boldsymbol{S}_i^k \boldsymbol{D} \boldsymbol{S}_i^k\right] - \boldsymbol{D}\right)\boldsymbol{L}_i^{\frac{1}{2}}\right)$ by the product of $\lambda_{\max}(\boldsymbol{L}_i)$ and $\lambda_{\max}\left(\mathbb{E}\left[\boldsymbol{S}_i^k \boldsymbol{D} \boldsymbol{S}_i^k\right] - \boldsymbol{D}\right)$. This allows us to simplify the expression since $\lambda_{\max}\left(\mathbb{E}\left[\boldsymbol{S}_i^k \boldsymbol{D} \boldsymbol{S}_i^k\right] - \boldsymbol{D}\right)$ is independent of the index $i$. Notice that we have already defined:

$$R(\boldsymbol{D}, \mathcal{S}) = \frac{1}{n} \sum_{i=1}^n \lambda_{\max}\left(\mathbb{E}\left[\boldsymbol{S}_i^k \boldsymbol{D} \boldsymbol{S}_i^k\right] - \boldsymbol{D}\right) \cdot \lambda_{\max}(\boldsymbol{L}_i) \cdot \lambda_{\max}\left(\boldsymbol{L}^{-\frac{1}{2}} \boldsymbol{L}_i \boldsymbol{L}^{-\frac{1}{2}}\right).$$

Taking expectation, using tower property and using the definition above, we obtain

$$\mathbb{E}\left[\left\|g^{k+1} - \nabla f(x^{k+1})\right\|_{\boldsymbol{D}}^2\right] \leq \frac{(1-p)\cdot R(\boldsymbol{D}, \mathcal{S})}{n}\mathbb{E}\left[\left\|x^{k+1} - x^k\right\|_{\boldsymbol{L}}^2\right] + (1-p)\mathbb{E}\left[\left\|g^k - \nabla f(x^k)\right\|_{\boldsymbol{D}}^2\right]. \tag{19}$$

Construct the following Lyapunov function $\Phi_k = f(x^k) - f^\star + \frac{1}{2p}\left\|g^k - \nabla f(x^k)\right\|_{\boldsymbol{D}}^2$. Using (15) and (19), we have

$$\mathbb{E}[\Phi_{k+1}] \leq \frac{1}{2p}\left[\frac{(1-p)\cdot R(\boldsymbol{D}, \mathcal{S})}{n}\mathbb{E}\left[\left\|x^{k+1} - x^k\right\|_{\boldsymbol{L}}^2\right] + (1-p)\cdot\mathbb{E}\left[\left\|g^k - \nabla f(x^k)\right\|_{\boldsymbol{D}}^2\right]\right]$$

$$+ \mathbb{E}\left[f(x^k) - f^\star\right] - \frac{1}{2}\mathbb{E}\left[\left\|\nabla f(x^k)\right\|_{\boldsymbol{D}}^2\right]$$

$$+ \frac{1}{2}\mathbb{E}\left[\left\|g^k - \nabla f(x^k)\right\|_{\boldsymbol{D}}^2\right] - \frac{1}{2}\mathbb{E}\left[\left\|x^{k+1} - x^k\right\|_{\boldsymbol{D}^{-1}-\boldsymbol{L}}^2\right]$$

$$= \mathbb{E}[\Phi_k] - \frac{1}{2}\mathbb{E}\left[\left\|\nabla f(x^k)\right\|_{\boldsymbol{D}}^2\right]$$

$$+ \left(\frac{(1-p)\cdot R(\boldsymbol{D}, \mathcal{S})}{2np}\mathbb{E}\left[\left\|x^{k+1} - x^k\right\|_{\boldsymbol{L}}^2\right] - \frac{1}{2}\mathbb{E}\left[\left\|x^{k+1} - x^k\right\|_{\boldsymbol{D}^{-1}-\boldsymbol{L}}^2\right]\right)$$

$$= \mathbb{E}[\Phi_k] - \frac{1}{2}\mathbb{E}\left[\left\|\nabla f(x^k)\right\|_{\boldsymbol{D}}^2\right]$$

$$+ \frac{1}{2}\left(\frac{(1-p)\cdot R(\boldsymbol{D}, \mathcal{S})}{np}\mathbb{E}\left[\left\|x^{k+1} - x^k\right\|_{\boldsymbol{L}}^2\right] - \mathbb{E}\left[\left\|x^{k+1} - x^k\right\|_{\boldsymbol{D}^{-1}-\boldsymbol{L}}^2\right]\right).$$

We rewrite the last term as

$$\mathbb{E}\left[(x^{k+1} - x^k)^\top\left[\frac{(1-p)\cdot R(\boldsymbol{D}, \mathcal{S})}{np}\boldsymbol{L} + \boldsymbol{L} - \boldsymbol{D}^{-1}\right](x^{k+1} - x^k)\right], \tag{20}$$

and we require the matrix in between to be negative semi-definite:

$$\boldsymbol{D}^{-1} \succeq \left(\frac{(1-p)\cdot R(\boldsymbol{D}, \mathcal{S})}{np} + 1\right)\boldsymbol{L}.$$

As a result, (20) is always non-positive and we obtain

$$\mathbb{E}[\Phi_{k+1}] \leq \mathbb{E}[\Phi_k] - \frac{1}{2}\mathbb{E}\left[\left\|\nabla f(x^k)\right\|_{\boldsymbol{D}}^2\right].$$

Unrolling this recurrence, we get

$$\frac{1}{K}\sum_{k=0}^{K-1}\mathbb{E}\left[\left\|\nabla f(x^k)\right\|_{\boldsymbol{D}}^2\right] \leq \frac{2\left(\mathbb{E}[\Phi_0] - \mathbb{E}[\Phi_K]\right)}{K}. \tag{21}$$

The left-hand side can viewed as $\mathbb{E}\left[\left\|\nabla f(\tilde{x}^K)\right\|_{\boldsymbol{D}}^2\right]$, where $\tilde{x}^K$ is sampled uniformly from $\{x_k\}_{k=0}^{K-1}$. Notice that $\Phi_K > 0$, we have

$$\frac{2\left(\mathbb{E}[\Phi_0] - \mathbb{E}[\Phi_K]\right)}{K} \quad \leq \quad \frac{2\Phi_0}{K} = \frac{2\left(f(x^0) - f^\star + \frac{1}{2p}\left\|g^0 - \nabla f(x^0)\right\|_{\boldsymbol{D}}^2\right)}{K} = \frac{2\left(f(x^0) - f^\star\right)}{K}.$$

Plugging in the simplified result into (21), and performing determinant normalization, we obtain

$$\mathbb{E}\left[\left\|\nabla f(\tilde{x}^K)\right\|_{\frac{\boldsymbol{D}}{\det(\boldsymbol{D})^{1/d}}}^2\right] \leq \frac{2\left(f(x^0) - f^\star\right)}{\det(\boldsymbol{D})^{1/d}K}. \tag{22}$$

*Remark* G.3. We can achieve a slightly more refined stepsize condition than (3) for det-MARINA, which is given as follows

$$\boldsymbol{D} \succeq \left(\frac{(1-p)\cdot \tilde{R}(\boldsymbol{D},\mathcal{S})}{np} + 1\right)\boldsymbol{L}, \tag{23}$$

where

$$\tilde{R}(\boldsymbol{D},\mathcal{S}) := \frac{1}{n}\sum_{i=1}^{n}\lambda_{\max}\left(\boldsymbol{L}_i^{\frac{1}{2}}\left(\mathbb{E}\left[\boldsymbol{S}_i^k\boldsymbol{D}\boldsymbol{S}_i^k\right] - \boldsymbol{D}\right)\boldsymbol{L}_i^{\frac{1}{2}}\right)\cdot\lambda_{\max}\left(\boldsymbol{L}^{-\frac{1}{2}}\boldsymbol{L}_i\boldsymbol{L}^{-\frac{1}{2}}\right).$$

This is obtained if we do not use Fact E.5 to upper bound $\lambda_{\max}\left(\boldsymbol{L}_i^{\frac{1}{2}}\left(\mathbb{E}\left[\boldsymbol{S}_i^k\boldsymbol{D}\boldsymbol{S}_i^k\right] - \boldsymbol{D}\right)\boldsymbol{L}_i^{\frac{1}{2}}\right)$ by the product of $\lambda_{\max}\left(\boldsymbol{L}_i\right)$ and $\lambda_{\max}\left(\mathbb{E}\left[\boldsymbol{S}_i^k\boldsymbol{D}\boldsymbol{S}_i^k\right] - \boldsymbol{D}\right)$. However, (23) results in a condition that is much harder to solve even if we assume $\boldsymbol{D} = \gamma\cdot\boldsymbol{W}$. So instead of using the more refined condition (23), we turn to (3). Notice that both of the two conditions (23) and (3) reduce to the stepsize condition for MARINA in the scalar setting.

### G.3 COMPARISON OF DIFFERENT STEPSIZES

In Corollary 6.1, we focus on a special stepsize where we fix $\boldsymbol{W} = \boldsymbol{L}^{-1}$ and demonstrate that, in this case, det-MARINA outperforms MARINA in terms of both iteration and communication complexities. However, other choices for $\boldsymbol{W}$ are also possible. Specifically, we consider the case where $\boldsymbol{W} = \text{diag}^{-1}\left(\boldsymbol{L}\right)$.

#### G.3.1 THE DIAGONAL CASE

We consider $\boldsymbol{W} = \text{diag}^{-1}\left(\boldsymbol{L}\right)$.

**Corollary G.4.** *If we take $\boldsymbol{W} = \text{diag}^{-1}\left(\boldsymbol{L}\right)$ in Corollary 4.7, then the optimal stepsize satisfies*

$$\boldsymbol{D}_{\text{diag}^{-1}(\boldsymbol{L})}^* = \frac{2}{1 + \sqrt{1 + 4\alpha\beta\cdot\Lambda_{\text{diag}^{-1}(\boldsymbol{L}),\mathcal{S}}}}\cdot\text{diag}^{-1}\left(\boldsymbol{L}\right). \tag{24}$$

*This stepsize leads to better iteration complexity for det-MARINA compared to the scalar version of MARINA.*

Since the same sketch is used for MARINA and det-MARINA, the communication complexity is improved as well. However, in general there is no clear relation between the iteration complexity of $\boldsymbol{W} = \boldsymbol{L}^{-1}$ and $\boldsymbol{W} = \text{diag}^{-1}\left(\boldsymbol{L}\right)$. This is also confirmed by one of our experiments, see Figure 6 to see the comparison of det-MARINA using those stepsizes.

### G.4 PROOF OF COROLLARY 4.7

We start with rewriting (3) as

$$\left(\frac{1-p}{np} \cdot R\left(\boldsymbol{D}, \mathcal{S}\right) + 1\right) \boldsymbol{D}^{\frac{1}{2}} \boldsymbol{L} \boldsymbol{D}^{\frac{1}{2}} \preceq \boldsymbol{I}_d.$$

Notice that we have already defined

$$\alpha = \frac{1-p}{np}; \qquad \beta = \frac{1}{n}\sum_{i=1}^{n} \lambda_{\max}\left(\boldsymbol{L}_i\right) \cdot \lambda_{\max}\left(\boldsymbol{L}^{-1}\boldsymbol{L}_i\right);$$

$$\Lambda_{\boldsymbol{W},\mathcal{S}} = \lambda_{\max}\left(\mathbb{E}\left[\boldsymbol{S}_i^k \boldsymbol{W} \boldsymbol{S}_i^k\right] - \boldsymbol{W}\right); \qquad \lambda_{\boldsymbol{W}} = \lambda_{\max}^{-1}\left(\boldsymbol{W}^{\frac{1}{2}} \boldsymbol{L} \boldsymbol{W}^{\frac{1}{2}}\right).$$

Plugging in the definition of $R(\boldsymbol{D}, \mathcal{S})$ and $\boldsymbol{D} = \gamma \boldsymbol{W}$, we obtain

$$\alpha\beta\Lambda_{\boldsymbol{W},\mathcal{S}} \cdot \gamma^2 + \gamma - \lambda_{\boldsymbol{W}} \le 0,$$

which yields the upper bound on $\gamma$

$$\gamma \le \frac{\sqrt{1 + 4\alpha\beta \cdot \Lambda_{\boldsymbol{W},\mathcal{S}}\lambda_{\boldsymbol{W}}} - 1}{2\alpha\beta \cdot \Lambda_{\boldsymbol{W},\mathcal{S}}}.$$

Since $\sqrt{1 + 4\alpha\beta \cdot \Lambda_{\boldsymbol{W},\mathcal{S}}\lambda_{\boldsymbol{W}}} + 1 > 0$, we further simplify the result as

$$\gamma \le \frac{2\lambda_{\boldsymbol{W}}}{1 + \sqrt{1 + 4\alpha\beta \cdot \Lambda_{\boldsymbol{W},\mathcal{S}}\lambda_{\boldsymbol{W}}}}.$$

### G.5 PROOF OF COROLLARY 6.1

It is obvious that (6) directly follows from plugging $\boldsymbol{W} = \boldsymbol{L}^{-1}$ into (5). The iteration complexity of MARINA, according to Gorbunov et al. (2021), is

$$K \ge K_1 = \mathcal{O}\left(\frac{\Delta_0 L}{\varepsilon^2}\left(1 + \sqrt{\frac{(1-p)\omega}{pn}}\right)\right). \tag{25}$$

On the other hand,

$$\det(\boldsymbol{L})^{\frac{1}{d}} \le \lambda_{\max}\left(\boldsymbol{L}\right) = L. \tag{26}$$

In addition, using the inequality

$$\sqrt{1 + 4t} \le 1 + 2\sqrt{t}, \tag{27}$$

which holds for any $t \ge 0$, we obtain the following bound

$$\frac{\left(1 + \sqrt{1 + 4\alpha\beta \cdot \Lambda_{\boldsymbol{L}^{-1},\mathcal{S}}}\right)}{2} \le 1 + \sqrt{\alpha\beta \cdot \Lambda_{\boldsymbol{L}^{-1},\mathcal{S}}}.$$

Next we prove that

$$1 + \sqrt{\alpha\beta \cdot \Lambda_{\boldsymbol{L}^{-1},\mathcal{S}}} \le 1 + \sqrt{\frac{(1-p)}{pn} \cdot \omega}, \tag{28}$$

which is equivalent to

$$\frac{1}{n}\sum_{i=1}^{n} \lambda_{\max}\left(\boldsymbol{L}_i\right) \lambda_{\max}\left(\boldsymbol{L}_i \boldsymbol{L}^{-1}\right) \cdot \lambda_{\max}\left(\mathbb{E}\left[\boldsymbol{S}_i^k \boldsymbol{L}^{-1} \boldsymbol{S}_i^k\right] - \boldsymbol{L}^{-1}\right) \le \omega.$$

The left hand side can be upper bounded by,

$$\frac{1}{n}\sum_{i=1}^{n} \lambda_{\max}\left(\boldsymbol{L}_i\right) \lambda_{\max}\left(\boldsymbol{L}^{-1}\boldsymbol{L}_i\right) \cdot \lambda_{\max}\left(\boldsymbol{L}^{-1}\right) \cdot \frac{\lambda_{\max}\left(\mathbb{E}\left[\boldsymbol{S}_i^k \boldsymbol{L}^{-1} \boldsymbol{S}_i^k\right] - \boldsymbol{L}^{-1}\right)}{\lambda_{\max}\left(\boldsymbol{L}^{-1}\right)}$$

$$\le \frac{\lambda_{\max}\left(\mathbb{E}\left[\boldsymbol{S}_i^k \boldsymbol{L}^{-1} \boldsymbol{S}_i^k\right] - \boldsymbol{L}^{-1}\right)}{\lambda_{\max}\left(\boldsymbol{L}^{-1}\right)},$$

where the inequality is a consequence of Lemma 3.4. We further bound the last term with

$$\frac{\lambda_{\max}\left(\mathbb{E}\left[\boldsymbol{S}_i^k \boldsymbol{L}^{-1} \boldsymbol{S}_i^k\right] - \boldsymbol{L}^{-1}\right)}{\lambda_{\max}\left(\boldsymbol{L}^{-1}\right)} = \lambda_{\max}\left(\mathbb{E}\left[\boldsymbol{S}_i^k \cdot \frac{\boldsymbol{L}^{-1}}{\lambda_{\max}(\boldsymbol{L}^{-1})} \cdot \boldsymbol{S}_i^k\right] - \frac{\boldsymbol{L}^{-1}}{\lambda_{\max}\left(\boldsymbol{L}^{-1}\right)}\right)$$
$$\leq \lambda_{\max}\left(\mathbb{E}\left[\boldsymbol{S}_i^k \boldsymbol{S}_i^k\right] - \boldsymbol{I}_d\right) =: \omega.$$

Here, the last inequality is due to the monotonicity of the mapping $\boldsymbol{X} \mapsto \lambda_{\max}\left(\mathbb{E}\left[\boldsymbol{S}_i^k \boldsymbol{X} \boldsymbol{S}_i^k\right] - \boldsymbol{X}\right)$ with $\boldsymbol{X} \in \mathbb{S}_{++}^d$, which can be shown as follows, let us pick any $\boldsymbol{X}_1, \boldsymbol{X}_2 \in \mathbb{S}_{++}^d$ and $\boldsymbol{X}_1 \preceq \boldsymbol{X}_2$,

$$\left(\mathbb{E}\left[\boldsymbol{S}_i^k \boldsymbol{X}_2 \boldsymbol{S}_i^k\right] - \boldsymbol{X}_2\right) - \left(\mathbb{E}\left[\boldsymbol{S}_i^k \boldsymbol{X}_1 \boldsymbol{S}_i^k\right] - \boldsymbol{X}_1\right) = \mathbb{E}\left[\boldsymbol{S}_i^k\left(\boldsymbol{X}_2 - \boldsymbol{X}_1\right)\boldsymbol{S}_i^k\right] - \left(\boldsymbol{X}_2 - \boldsymbol{X}_1\right) \succeq \boldsymbol{O}_d.$$

The above inequality is due to the convexity of the mapping $\boldsymbol{S}_i^k \mapsto \boldsymbol{S}_i^k \boldsymbol{X} \boldsymbol{S}_i^k$. As a result, we have

$$\lambda_{\max}\left(\mathbb{E}\left[\boldsymbol{S}_i^k \boldsymbol{X}_2 \boldsymbol{S}_i^k\right] - \boldsymbol{X}_2\right) \geq \lambda_{\max}\left(\mathbb{E}\left[\boldsymbol{S}_i^k \boldsymbol{X}_1 \boldsymbol{S}_i^k\right] - \boldsymbol{X}_1\right),$$

whenever $\boldsymbol{X}_2 \succeq \boldsymbol{X}_1$. Due to the fact that

$$\frac{\boldsymbol{L}^{-1}}{\lambda_{\max}\left(\boldsymbol{L}^{-1}\right)} \preceq \boldsymbol{I}_d,$$

we have

$$\lambda_{\max}\left(\mathbb{E}\left[\boldsymbol{S}_i^k \cdot \frac{\boldsymbol{L}^{-1}}{\lambda_{\max}(\boldsymbol{L}^{-1})} \cdot \boldsymbol{S}_i^k\right] - \frac{\boldsymbol{L}^{-1}}{\lambda_{\max}\left(\boldsymbol{L}^{-1}\right)}\right) \leq \lambda_{\max}\left(\mathbb{E}\left[\boldsymbol{S}_i^k \cdot \boldsymbol{I}_d \cdot \boldsymbol{S}_i^k\right] - \boldsymbol{I}_d\right) = \omega.$$

Combining (26) and (28), we have

$$\frac{\Delta_0 \det(\boldsymbol{L})^{\frac{1}{d}}}{\varepsilon^2} \cdot \left(1 + \sqrt{1 + 4\alpha\beta \cdot \Lambda_{\boldsymbol{L}^{-1},\mathcal{S}}}\right) \leq \frac{\Delta_0 L}{\varepsilon^2}\left(1 + \sqrt{\frac{(1-p)\omega}{pn}}\right),$$

which implies that the iteration complexity of det-MARINA is always better than that of MARINA.

### G.6 PROOF OF COROLLARY 6.3

The number of bits sent in expectation is $\mathcal{O}(d + K(pd + (1-p)\zeta_\mathcal{S})) = \mathcal{O}((Kp+1)d + (1-p)K\zeta_\mathcal{S})$. The special case where we choose $p = \zeta_\mathcal{S}/d$ indicates that $\alpha = \frac{1-p}{np} = \frac{1}{n}\left(\frac{d}{\zeta_\mathcal{S}} - 1\right)$. In order to reach an error of $\varepsilon^2$, we need

$$K = \mathcal{O}\left(\frac{\Delta_0 \cdot \det(\boldsymbol{L})^{\frac{1}{d}}}{\varepsilon^2} \cdot \left(1 + \sqrt{1 + \frac{4\beta}{n}\left(\frac{d}{\zeta_\mathcal{S}} - 1\right) \cdot \Lambda_{\boldsymbol{L}^{-1},\mathcal{S}}}\right)\right).$$

Applying once again (27), using the fact that $p = \zeta_\mathcal{S}/d$, the communication complexity in this case is given by

$$\mathcal{O}\left(d + \frac{\Delta_0 \cdot \det(\boldsymbol{L})^{\frac{1}{d}}}{\varepsilon^2} \cdot \left(1 + \sqrt{1 + \frac{4\beta}{n}\left(\frac{d}{\zeta_\mathcal{S}} - 1\right) \cdot \Lambda_{\boldsymbol{L}^{-1},\mathcal{S}}}\right) \cdot (pd + (1-p)\zeta_\mathcal{S})\right)$$

$$\leq \mathcal{O}\left(d + \frac{2\Delta_0 \cdot \det(\boldsymbol{L})^{\frac{1}{d}}}{\varepsilon^2} \cdot \left(1 + \sqrt{\frac{\beta}{n}\left(\frac{d}{\zeta_\mathcal{S}} - 1\right) \cdot \Lambda_{\boldsymbol{L}^{-1},\mathcal{S}}}\right) \cdot (pd + (1-p)\zeta_\mathcal{S})\right)$$

$$\leq \mathcal{O}\left(d + \frac{4\Delta_0 \cdot \det(\boldsymbol{L})^{\frac{1}{d}}}{\varepsilon^2} \cdot \left(\zeta_\mathcal{S} + \sqrt{\frac{\beta \cdot \Lambda_{\boldsymbol{L}^{-1},\mathcal{S}}}{n} \cdot \zeta_\mathcal{S}(d - \zeta_\mathcal{S})}\right)\right).$$

Ignoring the coefficient, we have

$$\mathcal{O}\left(d + \frac{\Delta_0 \cdot \det(\boldsymbol{L})^{\frac{1}{d}}}{\varepsilon^2} \cdot \left(\zeta_\mathcal{S} + \sqrt{\frac{\beta \cdot \Lambda_{\boldsymbol{L}^{-1},\mathcal{S}}}{n} \cdot \zeta_\mathcal{S}(d - \zeta_\mathcal{S})}\right)\right).$$

### G.7 PROOF OF COROLLARY G.4

Applying Corollary 4.7, notice that in this case $\lambda_{\mathrm{diag}^{-1}(\boldsymbol{L})} = \lambda_{\max}^{-1}\left(\mathrm{diag}^{-\frac{1}{2}}\left(\boldsymbol{L}\right)\boldsymbol{L}\,\mathrm{diag}^{-\frac{1}{2}}\left(\boldsymbol{L}\right)\right) = 1$, we obtain $\boldsymbol{D}_{\mathrm{diag}^{-1}(\boldsymbol{L})}^{*}$. The iteration complexity is given by

$$\mathcal{O}\left(\frac{\det\left(\mathrm{diag}(\boldsymbol{L})\right)^{\frac{1}{d}}\cdot\Delta_0}{\varepsilon^2}\cdot\left(\frac{1+\sqrt{1+4\alpha\beta\Lambda_{\mathrm{diag}^{-1}(\boldsymbol{L}),\mathcal{S}}}}{2}\right)\right).$$

We now compare it to the iteration complexity of MARINA, which is given in (25). We know that each diagonal element $\boldsymbol{L}_{jj}$ satisfies $\boldsymbol{L}_{jj} \le \lambda_{\max}\left(\boldsymbol{L}\right) = L$ for $j = 1, \ldots, d$. As a result,

$$\det\left(\mathrm{diag}(\boldsymbol{L})\right)^{\frac{1}{d}} \le L. \tag{29}$$

From (27), we deduce

$$\frac{1+\sqrt{1+4\alpha\beta\cdot\Lambda_{\mathrm{diag}^{-1}(\boldsymbol{L}),\mathcal{S}}}}{2} \le 1 + \sqrt{\alpha\beta\cdot\Lambda_{\mathrm{diag}^{-1}(\boldsymbol{L}),\mathcal{S}}}.$$

Now, let us prove the below inequality

$$1 + \sqrt{\alpha\beta\cdot\Lambda_{\mathrm{diag}^{-1}(\boldsymbol{L}),\mathcal{S}}} \le 1 + \sqrt{\frac{(1-p)}{pn}\cdot\omega}, \tag{30}$$

which is equivalent to $\beta\cdot\Lambda_{\mathrm{diag}^{-1}(\boldsymbol{L}),\mathcal{S}} \le \omega$. Plugging in the definition of $\beta$, $\omega$ and $\Lambda_{\mathrm{diag}^{-1}(\boldsymbol{L}),\mathcal{S}}$ and using Lemma 3.4, we obtain,

$$\lambda_{\max}\left(\mathbb{E}\left[\boldsymbol{S}_i^k\frac{\mathrm{diag}^{-1}\left(\boldsymbol{L}\right)}{\lambda_{\max}\left(\boldsymbol{L}^{-1}\right)}\boldsymbol{S}_i^k - \frac{\mathrm{diag}^{-1}\left(\boldsymbol{L}\right)}{\lambda_{\max}\left(\boldsymbol{L}^{-1}\right)}\right]\right) \le \lambda_{\max}\left(\mathbb{E}\left[\boldsymbol{S}_i^k\boldsymbol{I}_d\boldsymbol{S}_i^k\right] - \boldsymbol{I}_d\right).$$

It is enough to prove that $\frac{\mathrm{diag}^{-1}(\boldsymbol{L})}{\lambda_{\max}(\boldsymbol{L}^{-1})} \preceq \boldsymbol{I}_d$, which can be further simplified as $\lambda_{\min}\left(\boldsymbol{L}\right) \le \lambda_{\min}\left(\mathrm{diag}(\boldsymbol{L})\right)$. This is always true for any $\boldsymbol{L} \in \mathbb{S}_{++}^d$. Combining (29) and (30) we conclude the proof.

### G.8 PROOF OF LEMMA G.1

Let $\bar{x}^{k+1} := x^k - \boldsymbol{D}\cdot\nabla f(x^k)$. Since $f$ has matrix $\boldsymbol{L}$-Lipschitz gradient, by Lemma F.7, $f$ is also $\boldsymbol{L}$-smooth. By the $\boldsymbol{L}$-smoothness of $f$, we have

$$f(x^{k+1})$$

$$\le f(x^k) + \left\langle\nabla f(x^k), x^{k+1} - x^k\right\rangle + \frac{1}{2}\left\langle x^{k+1} - x^k, \boldsymbol{L}(x^{k+1} - x^k)\right\rangle$$

$$= f(x^k) + \left\langle\nabla f(x^k) - g^k, x^{k+1} - x^k\right\rangle + \left\langle g^k, x^{k+1} - x^k\right\rangle + \frac{1}{2}\left\langle x^{k+1} - x^k, \boldsymbol{L}(x^{k+1} - x^k)\right\rangle.$$

We can merge the last two terms and obtain,

$$f(x^{k+1}) \le f(x^k) + \left\langle\nabla f(x^k) - g^k, -\boldsymbol{D}\cdot g^k\right\rangle - \left\langle x^{k+1} - x^k, \boldsymbol{D}^{-1}(x^{k+1} - x^k)\right\rangle$$
$$+ \frac{1}{2}\left\langle x^{k+1} - x^k, \boldsymbol{L}(x^{k+1} - x^k)\right\rangle$$
$$= f(x^k) + \left\langle\nabla f(x^k) - g^k, -\boldsymbol{D}\cdot g^k\right\rangle - \left\langle x^{k+1} - x^k, \left(\boldsymbol{D}^{-1} - \frac{1}{2}\boldsymbol{L}\right)(x^{k+1} - x^k)\right\rangle.$$

We add and subtract $\left\langle\nabla f(x^k) - g^k, \boldsymbol{D}\cdot g^k\right\rangle$,

$$f(x^{k+1}) \le f(x^k) + \left\langle\nabla f(x^k) - g^k, \boldsymbol{D}\left(\nabla f(x^k) - g^k\right)\right\rangle - \left\langle\nabla f(x^k) - g^k, \boldsymbol{D}\cdot\nabla f(x^k)\right\rangle$$
$$- \left\langle x^{k+1} - x^k, \left(\boldsymbol{D}^{-1} - \frac{1}{2}\boldsymbol{L}\right)(x^{k+1} - x^k)\right\rangle$$
$$= f(x^k) + \left\|\nabla f(x^k) - g^k\right\|_{\boldsymbol{D}}^2 - \left\langle x^{k+1} - \bar{x}^{k+1}, \boldsymbol{D}^{-1}\left(x^k - \bar{x}^{k+1}\right)\right\rangle$$
$$- \left\langle x^{k+1} - x^k, \left(\boldsymbol{D}^{-1} - \frac{1}{2}\boldsymbol{L}\right)(x^{k+1} - x^k)\right\rangle.$$

Decomposing the term $\left\langle x^{k+1} - \bar{x}^{k+1}, \boldsymbol{D}^{-1}\left(x^k - \bar{x}^{k+1}\right)\right\rangle$, we obtain

$$f(x^{k+1}) \leq f(x^k) + \left\|\nabla f(x^k) - g^k\right\|_{\boldsymbol{D}}^2 - \left\langle x^{k+1} - x^k, \left(\boldsymbol{D}^{-1} - \frac{1}{2}\boldsymbol{L}\right)(x^{k+1} - x^k)\right\rangle$$

$$-\frac{1}{2}\left(\left\|x^{k+1} - \bar{x}^{k+1}\right\|_{\boldsymbol{D}^{-1}}^2 + \left\|x^k - \bar{x}^{k+1}\right\|_{\boldsymbol{D}^{-1}}^2 - \left\|x^{k+1} - x^k\right\|_{\boldsymbol{D}^{-1}}^2\right).$$

Plugging in the definition of $x^{k+1}, \bar{x}^{k+1}$, we get

$$f(x^{k+1}) \leq f(x^k) + \left\|\nabla f(x^k) - g^k\right\|_{\boldsymbol{D}}^2 - \left\|x^{k+1} - x^k\right\|_{\boldsymbol{D}^{-1} - \frac{1}{2}\boldsymbol{L}}^2$$

$$-\frac{1}{2}\left(\left\|\boldsymbol{D}(\nabla f(x^k) - g^k)\right\|_{\boldsymbol{D}^{-1}}^2 + \left\|\boldsymbol{D} \cdot \nabla f(x^k)\right\|_{\boldsymbol{D}^{-1}}^2 - \left\|x^{k+1} - x^k\right\|_{\boldsymbol{D}^{-1}}^2\right)$$

$$= f(x^k) + \left\|\nabla f(x^k) - g^k\right\|_{\boldsymbol{D}}^2 - \left\|x^{k+1} - x^k\right\|_{\boldsymbol{D}^{-1} - \frac{1}{2}\boldsymbol{L}}^2$$

$$-\frac{1}{2}\left(\left\|\nabla f(x^k) - g^k\right\|_{\boldsymbol{D}}^2 + \left\|\nabla f(x^k)\right\|_{\boldsymbol{D}}^2 - \left\|x^{k+1} - x^k\right\|_{\boldsymbol{D}^{-1}}^2\right).$$

Rearranging terms we obtain,

$$f(x^{k+1}) \leq f(x^k) - \frac{1}{2}\left\|\nabla f(x^k)\right\|_{\boldsymbol{D}}^2 + \frac{1}{2}\left\|g^k - \nabla f(x^k)\right\|_{\boldsymbol{D}}^2 - \left\|x^{k+1} - x^k\right\|_{\boldsymbol{D}^{-1} - \frac{1}{2}\boldsymbol{L}}^2$$

$$+ \frac{1}{2}\left\|x^{k+1} - x^k\right\|_{\boldsymbol{D}^{-1}}^2$$

$$= f(x^k) - \frac{1}{2}\left\|\nabla f(x^k)\right\|_{\boldsymbol{D}}^2 + \frac{1}{2}\left\|g^k - \nabla f(x^k)\right\|_{\boldsymbol{D}}^2 - \frac{1}{2}\left\|x^{k+1} - x^k\right\|_{\boldsymbol{D}^{-1} - \boldsymbol{L}}.$$

### G.9 Proof of Lemma G.2

The definition of the weighted norm yields

$$\mathbb{E}\left[\left\|\boldsymbol{S}t - t\right\|_{\boldsymbol{D}}^2\right] = \mathbb{E}[\langle t, (\boldsymbol{S} - \boldsymbol{I}_d)\,\boldsymbol{D}\,(\boldsymbol{S} - \boldsymbol{I}_d)\,t\rangle]$$

$$= \langle t, \mathbb{E}[(\boldsymbol{S} - \boldsymbol{I}_d)\boldsymbol{D}(\boldsymbol{S} - \boldsymbol{I}_d)]\,t\rangle$$

$$= \left\langle t, \boldsymbol{L}^{-\frac{1}{2}} \cdot \mathbb{E}\left[\boldsymbol{L}^{\frac{1}{2}}(\boldsymbol{S} - \boldsymbol{I}_d)\boldsymbol{D}(\boldsymbol{S} - \boldsymbol{I}_d)\boldsymbol{L}^{\frac{1}{2}}\right] \cdot \boldsymbol{L}^{-\frac{1}{2}}t\right\rangle$$

$$= \left\langle \boldsymbol{L}^{-\frac{1}{2}}t, \mathbb{E}\left[\boldsymbol{L}^{\frac{1}{2}}(\boldsymbol{S} - \boldsymbol{I}_d)\boldsymbol{D}(\boldsymbol{S} - \boldsymbol{I}_d)\boldsymbol{L}^{\frac{1}{2}}\right] \cdot \boldsymbol{L}^{-\frac{1}{2}}t\right\rangle$$

$$\leq \lambda_{\max}\left(\mathbb{E}\left[\boldsymbol{L}^{\frac{1}{2}}(\boldsymbol{S} - \boldsymbol{I}_d)\boldsymbol{D}(\boldsymbol{S} - \boldsymbol{I}_d)\boldsymbol{L}^{\frac{1}{2}}\right]\right)\left\|\boldsymbol{L}^{-\frac{1}{2}}t\right\|^2$$

$$= \lambda_{\max}\left(\boldsymbol{L}^{\frac{1}{2}}\left(\mathbb{E}[\boldsymbol{S}\boldsymbol{D}\boldsymbol{S}] - \boldsymbol{D}\right)\boldsymbol{L}^{\frac{1}{2}}\right) \cdot \|t\|_{\boldsymbol{L}^{-1}}^2.$$

## H Analysis of det-DASHA

We first present some technical lemmas essential for the proof.

**Lemma H.1.** *Assume that Definition 3.2 holds and $h_i^0 = \nabla f_i(x^0)$, then for $h_i^{k+1}$ from Algorithm 2, we have for any $\boldsymbol{D} \in \mathbb{S}_{++}^d$*

$$\left\|h^{k+1} - \nabla f(x^{k+1})\right\|_{\boldsymbol{D}}^2 = \left\|h_i^{k+1} - \nabla f_i(x^{k+1})\right\|_{\boldsymbol{D}}^2 = 0. \quad \left\|h_i^{k+1} - h_i^k\right\|_{\boldsymbol{L}_i^{-1}}^2 \leq \left\|x^{k+1} - x^k\right\|_{\boldsymbol{L}_i}^2.$$

**Lemma H.2.** *Suppose $h^{k+1}$ and $g^{k+1}$ are from Algorithm 2, then the following recurrence holds,*

$$\mathbb{E}\left[\left\|g^{k+1} - h^{k+1}\right\|_{\boldsymbol{D}}^2\right]$$

$$\leq \frac{2\Lambda_{\boldsymbol{D},\mathcal{S}} \cdot \lambda_{\max}\left(\boldsymbol{D}^{-1}\right) \cdot \lambda_{\max}\left(\boldsymbol{D}\right)}{n^2}\sum_{i=1}^n \lambda_{\max}\left(\boldsymbol{L}_i\right)\mathbb{E}\left[\left\|h_i^{k+1} - h_i^k\right\|_{\boldsymbol{L}_i^{-1}}^2\right]$$

$$+ \frac{2a^2\Lambda_{\boldsymbol{D},\mathcal{S}} \cdot \lambda_{\max}\left(\boldsymbol{D}^{-1}\right)}{n^2}\sum_{i=1}^n \mathbb{E}\left[\left\|g_i^k - h_i^k\right\|_{\boldsymbol{D}}^2\right] + (1-a)^2\mathbb{E}\left[\left\|g^k - h^k\right\|_{\boldsymbol{D}}^2\right], \quad (31)$$

*where $\Lambda_{\boldsymbol{D},\mathcal{S}} = \lambda_{\max}\left(\mathbb{E}\left[\boldsymbol{S}_i^k\boldsymbol{D}\boldsymbol{S}_i^k\right] - \boldsymbol{D}\right)$ for $\boldsymbol{D} \in \mathbb{S}_{++}^d$ and $\boldsymbol{S}_i^k \sim \mathcal{S}$.*

**Lemma H.3.** *Suppose $h_i^{k+1}$ and $g_i^{k+1}$ for $i \in [n]$ are from Algorithm 2, then the following recurrence holds,*

$$
\mathbb{E}\left[\left\|g_i^{k+1} - h_i^{k+1}\right\|_{\boldsymbol{D}}^2\right]
$$
$$
\leq \left(2a^2\lambda_{\max}\left(\boldsymbol{D}^{-1}\right) \cdot \Lambda_{\boldsymbol{D},\mathcal{S}} + (1-a)^2\right) \cdot \mathbb{E}\left[\left\|g_i^k - h_i^k\right\|_{\boldsymbol{D}}^2\right]
$$
$$
+ 2\lambda_{\max}\left(\boldsymbol{D}^{-1}\right) \cdot \lambda_{\max}\left(\boldsymbol{D}\right) \cdot \Lambda_{\boldsymbol{D},\mathcal{S}} \cdot \lambda_{\max}\left(\boldsymbol{L}_i\right) \cdot \mathbb{E}\left[\left\|h_i^{k+1} - h_i^k\right\|_{\boldsymbol{L}_i^{-1}}^2\right].
$$

## H.1 PROOF OF THEOREM 5.1

Using Lemma G.1 and taking expectations, we obtain

$$
\mathbb{E}\left[f(x^{k+1})\right]
$$
$$
\leq \mathbb{E}\left[f(x^k)\right] - \frac{1}{2}\mathbb{E}\left[\left\|\nabla f(x^k)\right\|_{\boldsymbol{D}}^2\right] - \frac{1}{2}\mathbb{E}\left[\left\|x^{k+1} - x^k\right\|_{\boldsymbol{D}^{-1}-\boldsymbol{L}}^2\right] + \frac{1}{2}\mathbb{E}\left[\left\|g^k - \nabla f(x^k)\right\|_{\boldsymbol{D}}^2\right]
$$
$$
\leq \mathbb{E}\left[f(x^k)\right] - \frac{1}{2}\mathbb{E}\left[\left\|\nabla f(x^k)\right\|_{\boldsymbol{D}}^2\right] - \frac{1}{2}\mathbb{E}\left[\left\|x^{k+1} - x^k\right\|_{\boldsymbol{D}^{-1}-\boldsymbol{L}}^2\right]
$$
$$
+ \mathbb{E}\left[\frac{1}{2}\left\|g^k - h^k + h^k - \nabla f(x^k)\right\|_{\boldsymbol{D}}^2\right]
$$
$$
\leq \mathbb{E}\left[f(x^k)\right] - \frac{1}{2}\mathbb{E}\left[\left\|\nabla f(x^k)\right\|_{\boldsymbol{D}}^2\right] - \frac{1}{2}\mathbb{E}\left[\left\|x^{k+1} - x^k\right\|_{\boldsymbol{D}^{-1}-\boldsymbol{L}}^2\right]
$$
$$
+ \mathbb{E}\left[\left\|g^k - h^k\right\|_{\boldsymbol{D}}^2 + \left\|h^k - \nabla f(x^k)\right\|_{\boldsymbol{D}}^2\right], \tag{32}
$$

where the last step is due to the convexity of the norm. Using Lemma H.2, we obtain

$$
\mathbb{E}\left[\left\|g^{k+1} - h^{k+1}\right\|_{\boldsymbol{D}}^2\right] \leq \frac{2\omega_{\boldsymbol{D}} \cdot \lambda_{\max}\left(\boldsymbol{D}\right)}{n^2} \sum_{i=1}^n \lambda_{\max}\left(\boldsymbol{L}_i\right) \mathbb{E}\left[\left\|h_i^{k+1} - h_i^k\right\|_{\boldsymbol{L}_i^{-1}}^2\right]
$$
$$
+ \frac{2a^2\omega_{\boldsymbol{D}}}{n^2} \sum_{i=1}^n \mathbb{E}\left[\left\|g_i^k - h_i^k\right\|_{\boldsymbol{D}}^2\right] + (1-a)^2 \mathbb{E}\left[\left\|g^k - h^k\right\|_{\boldsymbol{D}}^2\right]. \tag{33}
$$

Using Lemma H.3, we get

$$
\mathbb{E}\left[\left\|g_i^{k+1} - h_i^{k+1}\right\|_{\boldsymbol{D}}^2\right]
$$
$$
\leq \left(2a^2\omega_{\boldsymbol{D}} + (1-a)^2\right) \mathbb{E}\left[\left\|g_i^k - h_i^k\right\|_{\boldsymbol{D}}^2\right] + 2\omega_{\boldsymbol{D}}\lambda_{\max}\left(\boldsymbol{D}\right)\lambda_{\max}\left(\boldsymbol{L}_i\right) \mathbb{E}\left[\left\|h_i^{k+1} - h_i^k\right\|_{\boldsymbol{L}_i^{-1}}^2\right]. \tag{34}
$$

Now let us fix $\kappa \in [0, +\infty)$, $\eta \in [0, +\infty)$ which we will determine later, and construct the following Lyapunov function $\Phi_k$

$$
\Phi_k = \mathbb{E}\left[f(x^k) - f^\star\right] + \kappa \cdot \mathbb{E}\left[\left\|g^k - h^k\right\|_{\boldsymbol{D}}^2\right] + \eta \cdot \mathbb{E}\left[\frac{1}{n}\sum_{i=1}^n \left\|g_i^k - h_i^k\right\|_{\boldsymbol{D}}^2\right]. \tag{35}
$$

Combining (32), (33) and (34), we get

$$
\Phi_{k+1}
$$

$$
\leq \mathbb{E}\left[f(x^k) - f^\star - \frac{1}{2}\left\|\nabla f(x^k)\right\|_{\boldsymbol{D}}^2\right]
$$

$$
+ \mathbb{E}\left[-\frac{1}{2}\left\|x^{k+1} - x^k\right\|_{\boldsymbol{D}^{-1}-\boldsymbol{L}}^2 + \left\|g^k - h^k\right\|_{\boldsymbol{D}}^2 + \left\|h^k - \nabla f(x^k)\right\|_{\boldsymbol{D}}^2\right]
$$

$$
+ \kappa(1-a)^2\mathbb{E}\left[\left\|g^k - h^k\right\|_{\boldsymbol{D}}^2\right] + \frac{2\kappa \cdot \omega_{\boldsymbol{D}}\lambda_{\max}(\boldsymbol{D})}{n} \cdot \frac{1}{n}\sum_{i=1}^n \lambda_{\max}(\boldsymbol{L}_i)\,\mathbb{E}\left[\left\|h_i^{k+1} - h_i^k\right\|_{\boldsymbol{L}_i^{-1}}^2\right]
$$

$$
+ \frac{2a^2\omega_{\boldsymbol{D}}\cdot\kappa}{n}\cdot\frac{1}{n}\sum_{i=1}^n\mathbb{E}\left[\left\|g_i^k - h_i^k\right\|_{\boldsymbol{D}}^2\right] + \eta\left(2a^2\omega_{\boldsymbol{D}} + (1-a)^2\right)\cdot\frac{1}{n}\sum_{i=1}^n\mathbb{E}\left[\left\|g_i^k - h_i^k\right\|_{\boldsymbol{D}}^2\right]
$$

$$
+ 2\eta\cdot\omega_{\boldsymbol{D}}\cdot\lambda_{\max}(\boldsymbol{D})\cdot\frac{1}{n}\sum_{i=1}^n\lambda_{\max}(\boldsymbol{L}_i)\cdot\mathbb{E}\left[\left\|h_i^{k+1} - h_i^k\right\|_{\boldsymbol{L}_i^{-1}}^2\right].
$$

Rearranging terms, and notice that $\left\|h^k - \nabla f(x^k)\right\|_{\boldsymbol{D}}^2 = 0$,

$$
\Phi_{k+1}
$$

$$
\leq \mathbb{E}\left[f(x^k) - f^\star\right] - \frac{1}{2}\mathbb{E}\left[\left\|\nabla f(x^k)\right\|_{\boldsymbol{D}}^2\right]
$$

$$
- \frac{1}{2}\mathbb{E}\left[\left\|x^{k+1} - x^k\right\|_{\boldsymbol{D}^{-1}-\boldsymbol{L}}^2\right] + \left(1 + \kappa(1-a)^2\right)\mathbb{E}\left[\left\|g^k - h^k\right\|_{\boldsymbol{D}}^2\right]
$$

$$
+ \left(\frac{2a^2\omega_{\boldsymbol{D}}\cdot\kappa}{n} + \eta\left(2a^2\omega_{\boldsymbol{D}} + (1-a)^2\right)\right)\cdot\frac{1}{n}\sum_{i=1}^n\mathbb{E}\left[\left\|g_i^k - h_i^k\right\|_{\boldsymbol{D}}^2\right]
$$

$$
+ \left(\frac{2\kappa\cdot\omega_{\boldsymbol{D}}\lambda_{\max}(\boldsymbol{D})}{n} + 2\eta\cdot\omega_{\boldsymbol{D}}\cdot\lambda_{\max}(\boldsymbol{D})\right)\cdot\frac{1}{n}\sum_{i=1}^n\lambda_{\max}(\boldsymbol{L}_i)\cdot\mathbb{E}\left[\left\|h_i^{k+1} - h_i^k\right\|_{\boldsymbol{L}_i^{-1}}^2\right].
$$

In order to proceed, we consider the choice of $\kappa$ and $\eta$, for $\kappa$,

$$
1 + \kappa(1-a)^2 \leq \kappa. \tag{36}
$$

It is then clear that the choice of $\kappa = \frac{1}{a}$ satisfies the condition. On the other hand, we look at the terms involving $\mathbb{E}\left[\left\|g_i^k - h_i^k\right\|_{\boldsymbol{D}}^2\right]$, which we denote as $T_1$:

$$
T_1 := \left(\frac{2a^2\omega_{\boldsymbol{D}}\cdot\kappa}{n} + \eta\left(2a^2\omega_{\boldsymbol{D}} + (1-a)^2\right)\right)\cdot\frac{1}{n}\sum_{i=1}^n\mathbb{E}\left[\left\|g_i^k - h_i^k\right\|_{\boldsymbol{D}}^2\right].
$$

Picking $\kappa = \frac{1}{a}$ and $a = \frac{1}{2\omega_{\boldsymbol{D}}+1}$,

$$
T_1 = \left(\frac{2\omega_{\boldsymbol{D}}}{n\cdot(2\omega_{\boldsymbol{D}}+1)} + \eta\cdot\frac{4\omega_{\boldsymbol{D}}^2 + 2\omega_{\boldsymbol{D}}}{(2\omega_{\boldsymbol{D}}+1)^2}\right)\cdot\frac{1}{n}\sum_{i=1}^n\mathbb{E}\left[\left\|g_i^k - h_i^k\right\|_{\boldsymbol{D}}^2\right].
$$

We pick $\eta$ so that it satisfies

$$
\left(\frac{2\omega_{\boldsymbol{D}}}{n\cdot(2\omega_{\boldsymbol{D}}+1)} + \eta\cdot\frac{4\omega_{\boldsymbol{D}}^2 + 2\omega_{\boldsymbol{D}}}{(2\omega_{\boldsymbol{D}}+1)^2}\right) \leq \eta. \tag{37}
$$

Taking $\eta = \frac{2\omega_{\boldsymbol{D}}}{n}$, which is the minimum value satisfying (37), we conclude that

$$
T_1 \leq \eta\cdot\frac{1}{n}\sum_{i=1}^n\mathbb{E}\left[\left\|g_i^k - h_i^k\right\|_{\boldsymbol{D}}^2\right]. \tag{38}
$$

Combining (36) and (38), we are able to conclude that

$$\Phi_{k+1}$$

$$\leq \mathbb{E}\big[f(x^k) - f^\star\big] + \kappa \cdot \mathbb{E}\Big[\big\|g^k - h^k\big\|_{\boldsymbol{D}}^2\Big] + \eta \cdot \frac{1}{n} \sum_{i=1}^{n} \mathbb{E}\Big[\big\|g_i^k - h_i^k\big\|_{\boldsymbol{D}}^2\Big]$$

$$- \frac{1}{2}\mathbb{E}\Big[\big\|\nabla f(x^k)\big\|_{\boldsymbol{D}}^2\Big] - \frac{1}{2}\mathbb{E}\Big[\big\|x^{k+1} - x^k\big\|_{\boldsymbol{D}^{-1}-\boldsymbol{L}}^2\Big]$$

$$+ \left(\frac{2\kappa \cdot \omega_{\boldsymbol{D}}\lambda_{\max}(\boldsymbol{D})}{n} + 2\eta \cdot \omega_{\boldsymbol{D}} \cdot \lambda_{\max}(\boldsymbol{D})\right) \cdot \frac{1}{n} \sum_{i=1}^{n} \lambda_{\max}(\boldsymbol{L}_i) \cdot \mathbb{E}\Big[\big\|h_i^{k+1} - h_i^k\big\|_{\boldsymbol{L}_i^{-1}}^2\Big].$$

Using the definition of $\Phi_k$ and Lemma H.1, we obtain

$$\Phi_{k+1} \leq \Phi_k - \frac{1}{2}\mathbb{E}\Big[\big\|\nabla f(x^k)\big\|_{\boldsymbol{D}}^2\Big] - \frac{1}{2}\mathbb{E}\Big[\big\|x^{k+1} - x^k\big\|_{\boldsymbol{D}^{-1}-\boldsymbol{L}}^2\Big]$$

$$\left(\frac{2\kappa \cdot \omega_{\boldsymbol{D}}\lambda_{\max}(\boldsymbol{D})}{n} + 2\eta \cdot \omega_{\boldsymbol{D}} \cdot \lambda_{\max}(\boldsymbol{D})\right) \cdot \frac{1}{n} \sum_{i=1}^{n} \lambda_{\max}(\boldsymbol{L}_i) \cdot \mathbb{E}\Big[\big\|x^{k+1} - x^k\big\|_{\boldsymbol{L}_i}^2\Big]$$

$$= \Phi_k - \frac{1}{2}\mathbb{E}\Big[\big\|\nabla f(x^k)\big\|_{\boldsymbol{D}}^2\Big] + \mathbb{E}\Big[\big\|x^{k+1} - x^k\big\|_{\boldsymbol{N}}^2\Big],$$

where $\boldsymbol{N} \in \mathbb{S}^d$ is defined as

$$\boldsymbol{N} := \left(\frac{2\kappa \cdot \omega_{\boldsymbol{D}}\lambda_{\max}(\boldsymbol{D})}{n} + 2\eta \cdot \omega_{\boldsymbol{D}} \cdot \lambda_{\max}(\boldsymbol{D})\right) \cdot \frac{1}{n} \sum_{i=1}^{n} \lambda_{\max}(\boldsymbol{L}_i) \cdot \boldsymbol{L}_i - \frac{1}{2}\boldsymbol{D}^{-1} + \frac{1}{2}\boldsymbol{L}.$$

We require $\boldsymbol{N} \preceq \boldsymbol{O}_d$, which leads to the following condition on $\boldsymbol{D}$:

$$\boldsymbol{D}^{-1} - \boldsymbol{L} - \frac{4\lambda_{\max}(\boldsymbol{D}) \cdot \omega_{\boldsymbol{D}} \cdot (4\omega_{\boldsymbol{D}} + 1)}{n} \cdot \frac{1}{n} \sum_{i=1}^{n} \lambda_{\max}(\boldsymbol{L}_i) \cdot \boldsymbol{L}_i \succeq \boldsymbol{O}_d.$$

Given the above condition is satisfied, we have the recurrence

$$\frac{1}{2}\mathbb{E}\Big[\big\|\nabla f(x^k)\big\|_{\boldsymbol{D}}^2\Big] \leq \Phi_k - \Phi_{k+1}$$

Summing up for $k = 0 \ldots K - 1$, we obtain

$$\sum_{k=0}^{K-1} \mathbb{E}\Big[\big\|\nabla f(x^k)\big\|_{\boldsymbol{D}}^2\Big] \leq 2(\Phi_0 - \Phi_k). \tag{39}$$

Notice that we also have

$$\Phi_0 = f(x^0) - f^\star + (2\omega_{\boldsymbol{D}} + 1)\big\|g^0 - h^0\big\|_{\boldsymbol{D}}^2 + \frac{2\omega_{\boldsymbol{D}}}{n} \cdot \frac{1}{n} \sum_{i=1}^{n} \big\|g_i^0 - h_i^0\big\|^2 = f(x^0) - f^\star,$$

We divide both sides of (39) by $K$, and perform determinant normalization,

$$\frac{1}{K} \sum_{k=0}^{K-1} \mathbb{E}\Big[\big\|\nabla f(x^k)\big\|_{\frac{\boldsymbol{D}}{\det(\boldsymbol{D})^{1/d}}}^2\Big] \leq \frac{2(f(x^0) - f^\star)}{\det(\boldsymbol{D})^{1/d} \cdot K}.$$

This is to say

$$\mathbb{E}\Big[\big\|\nabla f(\tilde{x}^K)\big\|_{\frac{\boldsymbol{D}}{\det(\boldsymbol{D})^{1/d}}}^2\Big] \leq \frac{2(f(x^0) - f^\star)}{\det(\boldsymbol{D})^{1/d} \cdot K},$$

where $\tilde{x}^K$ is chosen uniformly randomly from the first K iterates of the algorithm.

### H.2 Proof of Corollary 5.3

Plug $D = \gamma_{\boldsymbol{W}} \cdot \boldsymbol{W}$ into the stepsize condition in Theorem 5.1, we obtain

$$\frac{\boldsymbol{W}^{-1}}{\gamma_{\boldsymbol{W}}} - \boldsymbol{L} - \frac{4\gamma_{\boldsymbol{W}} \cdot \lambda_{\max}(\boldsymbol{W}) \cdot \omega_{\boldsymbol{W}}(4\omega_{\boldsymbol{W}}+1)}{n} \cdot \frac{1}{n}\sum_{i=1}^{n}\lambda_{\max}(\boldsymbol{L}_i) \cdot \boldsymbol{L}_i \succeq \boldsymbol{O}_d.$$

We then simplify the above condition as

$$\frac{\boldsymbol{L}^{-\frac{1}{2}}\boldsymbol{W}^{-1}\boldsymbol{L}^{-\frac{1}{2}}}{\gamma_{\boldsymbol{W}}}$$

$$\succeq \boldsymbol{I}_d + \frac{4\gamma_{\boldsymbol{W}} \cdot \lambda_{\max}(\boldsymbol{W}) \cdot \omega_{\boldsymbol{W}}(4\omega_{\boldsymbol{W}}+1)}{n} \cdot \boldsymbol{L}^{-\frac{1}{2}}\left(\frac{1}{n}\sum_{i=1}^{n}\lambda_{\max}(\boldsymbol{L}_i)\cdot\boldsymbol{L}_i\right)\boldsymbol{L}^{-\frac{1}{2}}.$$

Using Lemma F.2, we have

$$\frac{\boldsymbol{L}^{-\frac{1}{2}}\boldsymbol{W}^{-1}\boldsymbol{L}^{-\frac{1}{2}}}{\gamma_{\boldsymbol{W}}} - \frac{4\gamma_{\boldsymbol{W}} \cdot \lambda_{\max}(\boldsymbol{W}) \cdot \omega_{\boldsymbol{W}}(4\omega_{\boldsymbol{W}}+1)}{n} \cdot \lambda_{\min}(\boldsymbol{L}) \cdot \boldsymbol{I}_d \succeq \boldsymbol{I}_d.$$

Taking the minimum eigenvalue of both sides, we obtain that,

$$\frac{\lambda_{\min}\left(\boldsymbol{L}^{-\frac{1}{2}}\boldsymbol{W}^{-1}\boldsymbol{L}^{-\frac{1}{2}}\right)}{\gamma_{\boldsymbol{W}}} - \frac{4\gamma_{\boldsymbol{W}} \cdot \lambda_{\max}(\boldsymbol{W}) \cdot \omega_{\boldsymbol{W}}(4\omega_{\boldsymbol{W}}+1)}{n} \cdot \lambda_{\min}(\boldsymbol{L}) \geq 1,$$

If we denote $C_{\boldsymbol{W}} := \frac{\lambda_{\max}(\boldsymbol{W})\cdot\omega_{\boldsymbol{W}}(4\omega_{\boldsymbol{W}}+1)}{n} > 0$, and $\lambda_{\boldsymbol{W}} := \lambda_{\max}^{-1}\left(\boldsymbol{L}^{\frac{1}{2}}\boldsymbol{W}\boldsymbol{L}^{\frac{1}{2}}\right)$, we have $4 \cdot C_{\boldsymbol{W}} \cdot \lambda_{\min}(\boldsymbol{L}) \cdot \gamma_{\boldsymbol{W}}^2 + \gamma_{\boldsymbol{W}} - \lambda_{\boldsymbol{W}} \leq 0$, which gives

$$\gamma_{\boldsymbol{W}} \leq \frac{2\lambda_{\boldsymbol{W}}}{1 + \sqrt{1 + 16C_{\boldsymbol{W}}\lambda_{\min}(\boldsymbol{L}) \cdot \lambda_{\boldsymbol{W}}}}.$$

### H.3 Proof of Corollary 6.4

The best scaling factor for $\boldsymbol{L}^{-1}$, in this case, is given as, according to Corollary 5.3, $\gamma_{\boldsymbol{L}^{-1}} = \frac{2}{1+\sqrt{1+16C_{\boldsymbol{L}^{-1}}\cdot\lambda_{\min}(\boldsymbol{L})}}$. In order to reach a $\varepsilon^2$ stationary point, we need

$$K \geq \frac{\det(\boldsymbol{L})^{\frac{1}{d}}\left(f(x^0) - f^\star\right)}{\varepsilon^2} \cdot \left(1 + \sqrt{1 + 16C_{\boldsymbol{L}^{-1}} \cdot \lambda_{\min}(\boldsymbol{L})}\right).$$

### H.4 Proof of Corollary 6.5

The iteration complexity of det-DASHA is given by, according to, Corollary 6.4,

$$\mathcal{O}\left(\frac{f(x^0) - f^\star}{\epsilon^2} \cdot \left(1 + \sqrt{1 + 16C_{\boldsymbol{L}^{-1}} \cdot \lambda_{\min}(\boldsymbol{L})}\right) \cdot \det(\boldsymbol{L})^{\frac{1}{d}}\right).$$

Using the inequality $\sqrt{1+t} \leq 1 + \sqrt{t}$ for $t > 0$ and leaving out the coefficients, we obtain

$$\mathcal{O}\left(\frac{f(x^0) - f^\star}{\epsilon^2} \cdot \left(1 + \sqrt{C_{\boldsymbol{L}^{-1}} \cdot \lambda_{\min}(\boldsymbol{L})}\right) \cdot \det(\boldsymbol{L})^{\frac{1}{d}}\right).$$

Notice that

$$C_{\boldsymbol{L}^{-1}} \cdot \lambda_{\min}(\boldsymbol{L}) = \lambda_{\max}\left(\boldsymbol{L}^{-1}\right) \cdot \frac{\omega_{\boldsymbol{L}^{-1}}(4\omega_{\boldsymbol{L}^{-1}}+1)}{n} \cdot \lambda_{\min}(\boldsymbol{L}) = \frac{\omega_{\boldsymbol{L}^{-1}}(4\omega_{\boldsymbol{L}^{-1}}+1)}{n}.$$

As a result, the iteration complexity can be further simplified as

$$\mathcal{O}\left(\frac{f(x^0) - f^*}{\epsilon^2} \cdot \left(1 + \frac{\omega_{\boldsymbol{L}^{-1}}}{\sqrt{n}}\right) \cdot \det(\boldsymbol{L})^{\frac{1}{d}}\right).$$

The iteration complexity of DASHA is, according to (Tyurin & Richtárik, 2024, Corollary 6.2)

$$\mathcal{O}\left(\frac{1}{\epsilon^2} \cdot \left(f(x^0) - f^\star\right)\left(L + \frac{\omega}{\sqrt{n}}\widehat{L}\right)\right),$$

where $\widehat{L} = \sqrt{\frac{1}{n}\sum_{i=1}^{n}L_i^2}$. Since $\det(\boldsymbol{L})^{\frac{1}{d}} \leq \lambda_{\max}(\boldsymbol{L}) = L$, and $L \leq \widehat{L}$, we see that compared to DASHA, det-DASHA has a better iteration complexity when the momentum is the same.

### H.5 Proof of Corollary 6.6

The iteration complexity of det-MARINA is given by

$$\mathcal{O}\left(\frac{f(x^0) - f^\star}{\epsilon^2} \cdot \det(\boldsymbol{L})^{\frac{1}{d}} \cdot \left(1 + \sqrt{\alpha\beta\Lambda_{\boldsymbol{L}^{-1}, \mathcal{S}}}\right)\right),$$

after removing logarithmic factors. We obtain in the case of $\omega_{\boldsymbol{L}^{-1}} + 1 = \frac{1}{p}$ that

$$\mathcal{O}\left(\frac{f(x^0) - f^\star}{\epsilon^2} \cdot \det(\boldsymbol{L})^{\frac{1}{d}} \cdot \left(1 + \frac{\omega_{\boldsymbol{L}^{-1}}}{n}\right)\right).$$

From the proof of Corollary 6.5, we know that the iteration complexity of det-DASHA is

$$\mathcal{O}\left(\frac{1}{\epsilon^2} \cdot \left(f(x^0) - f^\star\right)\left(L + \frac{\omega}{\sqrt{n}}\widehat{L}\right)\right).$$

We see that in this case the two algorithms have the same iteration complexity asymptotically. Notice that the communication complexity is the product of bytes sent per iteration and the number of iterations. det-DASHA clearly sends less bytes per iteration since it always sends the compressed gradient differences, which leads to a better communication complexity than det-MARINA.

### H.6 Proof of Lemma H.2

Throughout the following proof, we denote $\mathbb{E}_{\boldsymbol{S}}[\cdot]$ as taking expectation with respect to the randomness contained within the sketch sampled from distribution $\mathcal{S}$. For $\mathbb{E}_{\boldsymbol{S}}\left[\left\|g^{k+1} - h^{k+1}\right\|_{\boldsymbol{D}}^2\right]$, we have

$$\mathbb{E}_{\boldsymbol{S}}\left[\left\|g^{k+1} - h^{k+1}\right\|_{\boldsymbol{D}}^2\right] = \mathbb{E}_{\boldsymbol{S}}\left[\left\|g^k + \frac{1}{n}\sum_{i=1}^n m_i^{k+1} - h^{k+1}\right\|_{\boldsymbol{D}}^2\right]$$

$$= \mathbb{E}_{\boldsymbol{S}}\left[\left\|g^k + \frac{1}{n}\sum_{i=1}^n \boldsymbol{S}_i^k\left(h_i^{k+1} - h_i^k - a(g_i^k - h_i^k)\right) - h^{k+1}\right\|_{\boldsymbol{D}}^2\right]$$

Using Fact E.3, we obtain

$$\mathbb{E}_{\boldsymbol{S}}\left[\left\|g^{k+1} - h^{k+1}\right\|_{\boldsymbol{D}}^2\right]$$

$$= \mathbb{E}_{\boldsymbol{S}}\left[\left\|\frac{1}{n}\sum_{i=1}^n \boldsymbol{S}_i^k\left(h_i^{k+1} - h_i^k - a(g_i^k - h_i^k)\right) - \left(h^{k+1} - h^k - a(g^k - h^k)\right)\right\|_{\boldsymbol{D}}^2\right]$$

$$+ (1-a)^2\left\|h^k - g^k\right\|_{\boldsymbol{D}}^2$$

$$= \mathbb{E}_{\boldsymbol{S}}\left[\left\|\frac{1}{n}\sum_{i=1}^n \boldsymbol{S}_i^k\left(h_i^{k+1} - h_i^k - a(g_i^k - h_i^k)\right) - \frac{1}{n}\sum_{i=1}^n\left(h_i^{k+1} - h_i^k - a(g_i^k - h_i^k)\right)\right\|_{\boldsymbol{D}}^2\right]$$

$$+ (1-a)^2\left\|h^k - g^k\right\|_{\boldsymbol{D}}^2$$

$$= \frac{1}{n^2}\sum_{i=1}^n \mathbb{E}_{\boldsymbol{S}}\left[\left\|\boldsymbol{S}_i^k\left(h_i^{k+1} - h_i^k - a(g_i^k - h_i^k)\right) - \left(h_i^{k+1} - h_i^k - a(g_i^k - h_i^k)\right)\right\|_{\boldsymbol{D}}^2\right]$$

$$+ (1-a)^2\left\|h^k - g^k\right\|_{\boldsymbol{D}}^2.$$

Here, the last identity is obtained from the unbiasedness of the sketches:

$$\mathbb{E}_{\boldsymbol{S}}\left[\boldsymbol{S}_i^k\left(h_i^{k+1} - h_i^k - a(g_i^k - h_i^k)\right)\right] = h_i^{k+1} - h_i^k - a(g_i^k - h_i^k).$$

We further use Lemma G.2, and obtain

$$
\mathbb{E}_{\boldsymbol{S}}\left[\left\|g^{k+1}-h^{k+1}\right\|_{\boldsymbol{D}}^2\right]
$$

$$
\leq \frac{1}{n^2}\sum_{i=1}^n \lambda_{\max}\left(\boldsymbol{D}^{-\frac12}\left(\mathbb{E}\left[\boldsymbol{S}_i^k\boldsymbol{D}\boldsymbol{S}_i^k\right]-\boldsymbol{D}\right)\boldsymbol{D}^{-\frac12}\right)\left\|h_i^{k+1}-h_i-a(g_i^k-h_i^k)\right\|_{\boldsymbol{D}}^2
$$

$$
+ (1-a)^2\left\|g^k-h^k\right\|_{\boldsymbol{D}}^2
$$

$$
\leq \frac{1}{n^2}\sum_{i=1}^n \lambda_{\max}\left(\boldsymbol{D}^{-1}\right)\cdot\lambda_{\max}\left(\mathbb{E}\left[\boldsymbol{S}_i^k\boldsymbol{D}\boldsymbol{S}_i^k\right]-\boldsymbol{D}\right)\left\|h_i^{k+1}-h_i^k-a(g_i^k-h_i^k)\right\|_{\boldsymbol{D}}^2
$$

$$
+ (1-a)^2\left\|g^k-h^k\right\|_{\boldsymbol{D}}^2.
$$

Applying Jensen's inequality as

$$
\mathbb{E}_{\boldsymbol{S}}\left[\left\|g^{k+1}-h^{k+1}\right\|_{\boldsymbol{D}}^2\right]
$$

$$
\leq \frac{2\Lambda_{\boldsymbol{D},\mathcal{S}}\cdot\lambda_{\max}\left(\boldsymbol{D}^{-1}\right)}{n^2}\sum_{i=1}^n\left\|h_i^{k+1}-h_i^k\right\|_{\boldsymbol{D}}^2 + \frac{2a^2\Lambda_{\boldsymbol{D},\mathcal{S}}\cdot\lambda_{\max}\left(\boldsymbol{D}^{-1}\right)}{n^2}\sum_{i=1}^n\left\|g_i^k-h_i^k\right\|_{\boldsymbol{D}}^2
$$

$$
+ (1-a)^2\left\|g^k-h^k\right\|_{\boldsymbol{D}}^2.
$$

Notice that we have

$$
\left\|h_i^{k+1}-h_i^k\right\|_{\boldsymbol{D}}^2 \leq \lambda_{\max}\left(\boldsymbol{D}\right)\cdot\lambda_{\max}\left(\boldsymbol{L}_i\right)\cdot\left\|h_i^{k+1}-h_i^k\right\|_{\boldsymbol{L}_i^{-1}}^2.
$$

We see that,

$$
\mathbb{E}_{\boldsymbol{S}}\left[\left\|g^{k+1}-h^{k+1}\right\|_{\boldsymbol{D}}^2\right]
$$

$$
\leq \frac{2\Lambda_{\boldsymbol{D},\mathcal{S}}\cdot\lambda_{\max}\left(\boldsymbol{D}^{-1}\right)\cdot\lambda_{\max}\left(\boldsymbol{D}\right)}{n^2}\sum_{i=1}^n \lambda_{\max}\left(\boldsymbol{L}_i\right)\left\|h_i^{k+1}-h_i^k\right\|_{\boldsymbol{L}_i^{-1}}^2
$$

$$
+ \frac{2a^2\Lambda_{\boldsymbol{D},\mathcal{S}}\cdot\lambda_{\max}\left(\boldsymbol{D}^{-1}\right)}{n^2}\sum_{i=1}^n\left\|g_i^k-h_i^k\right\|_{\boldsymbol{D}}^2 + (1-a)^2\left\|g^k-h^k\right\|_{\boldsymbol{D}}^2.
$$

We obtain the inequality in the lemma after taking expectation again and applying tower property.

### H.7   PROOF OF LEMMA H.3

We start with

$$
\mathbb{E}_{\boldsymbol{S}}\left[\left\|g_i^{k+1}-h_i^{k+1}\right\|_{\boldsymbol{D}}^2\right]
$$

$$
= \mathbb{E}_{\boldsymbol{S}}\left[\left\|g_i^k+\boldsymbol{S}_i^k\left(h_i^{k+1}-h_i^k-a(g_i^k-h_i^k)\right)-h_i^{k+1}\right\|_{\boldsymbol{D}}^2\right]
$$

$$
= \mathbb{E}_{\boldsymbol{S}}\left[\left\|\boldsymbol{S}_i^k\left(h_i^{k+1}-h_i^k-a(g_i^k-h_i^k)\right)-\left(h_i^{k+1}-h_i^k-a(g_i^k-h_i^k)\right)+(1-a)(h_i^k-g_i^k)\right\|_{\boldsymbol{D}}^2\right].
$$

Using Fact E.3,

$$
\mathbb{E}_{\boldsymbol{S}}\left[\left\|g_i^{k+1}-h_i^{k+1}\right\|_{\boldsymbol{D}}^2\right]
$$

$$
= \mathbb{E}_{\boldsymbol{S}}\left[\left\|\boldsymbol{S}_i^k\left(h_i^{k+1}-h_i^k-a(g_i^k-h_i^k)\right)-\left(h_i^{k+1}-h_i^k-a(g_i^k-h_i^k)\right)\right\|_{\boldsymbol{D}}^2\right]
$$

$$
+ (1-a)^2\left\|h_i^k-g_i^k\right\|_{\boldsymbol{D}}^2.
$$

Using Lemma G.2

$$\mathbb{E}_{\boldsymbol{S}}\Big[\big\|g_i^{k+1} - h_i^{k+1}\big\|_{\boldsymbol{D}}^2\Big]$$

$$\overset{(14)}{\leq} \lambda_{\max}\Big(\boldsymbol{D}^{-\frac{1}{2}}\big(\mathbb{E}\big[\boldsymbol{S}_i^k \boldsymbol{D}\boldsymbol{S}_i^k\big] - \boldsymbol{D}\big)\boldsymbol{D}^{-\frac{1}{2}}\Big)\big\|h_i^{k+1} - h_i^k - a(g_i^k - h_i^k)\big\|_{\boldsymbol{D}}^2$$

$$+ (1-a)^2\big\|g_i^k - h_i^k\big\|_{\boldsymbol{D}}^2$$

$$\leq \lambda_{\max}\big(\boldsymbol{D}^{-1}\big)\cdot \Lambda_{\boldsymbol{D},\mathcal{S}}\big\|h_i^{k+1} - h_i^k - a(g_i^k - h_i^k)\big\|_{\boldsymbol{D}}^2 + (1-a)^2\big\|g_i^k - h_i^k\big\|_{\boldsymbol{D}}^2$$

$$\leq 2\lambda_{\max}\big(\boldsymbol{D}^{-1}\big)\cdot \Lambda_{\boldsymbol{D},\mathcal{S}}\big\|h_i^{k+1} - h_i^k\big\|_{\boldsymbol{D}}^2 + 2a^2\lambda_{\max}\big(\boldsymbol{D}^{-1}\big)\cdot \Lambda_{\boldsymbol{D},\mathcal{S}}\big\|g_i^k - h_i^k\big\|_{\boldsymbol{D}}^2$$

$$+ (1-a)^2\big\|g_i^k - h_i^k\big\|_{\boldsymbol{D}}^2$$

$$\leq 2\lambda_{\max}\big(\boldsymbol{D}^{-1}\big)\cdot \lambda_{\max}\big(\boldsymbol{D}\big)\cdot \Lambda_{\boldsymbol{D},\mathcal{S}}\cdot \lambda_{\max}\big(\boldsymbol{L}_i\big)\cdot \big\|h_i^{k+1} - h_i^k\big\|_{\boldsymbol{L}_i^{-1}}^2$$

$$+ 2a^2\lambda_{\max}\big(\boldsymbol{D}^{-1}\big)\cdot \Lambda_{\boldsymbol{D},\mathcal{S}}\big\|g_i^k - h_i^k\big\|_{\boldsymbol{D}}^2 + (1-a)^2\big\|g_i^k - h_i^k\big\|_{\boldsymbol{D}}^2$$

$$= \big(2a^2\lambda_{\max}\big(\boldsymbol{D}^{-1}\big)\cdot \Lambda_{\boldsymbol{D},\mathcal{S}} + (1-a)^2\big)\big\|g_i^k - h_i^k\big\|_{\boldsymbol{D}}^2$$

$$+ 2\lambda_{\max}\big(\boldsymbol{D}^{-1}\big)\cdot \lambda_{\max}\big(\boldsymbol{D}\big)\cdot \Lambda_{\boldsymbol{D},\mathcal{S}}\cdot \lambda_{\max}\big(\boldsymbol{L}_i\big)\cdot \big\|h_i^{k+1} - h_i^k\big\|_{\boldsymbol{L}_i^{-1}}^2.$$

Taking expectation again, and using tower property, we obtain,

$$\mathbb{E}\Big[\big\|g_i^{k+1} - h_i^{k+1}\big\|_{\boldsymbol{D}}^2\Big]$$

$$\leq \big(2a^2\lambda_{\max}\big(\boldsymbol{D}^{-1}\big)\cdot \Lambda_{\boldsymbol{D},\mathcal{S}} + (1-a)^2\big)\mathbb{E}\Big[\big\|g_i^k - h_i^k\big\|_{\boldsymbol{D}}^2\Big]$$

$$+ 2\lambda_{\max}\big(\boldsymbol{D}^{-1}\big)\cdot \lambda_{\max}\big(\boldsymbol{D}\big)\cdot \Lambda_{\boldsymbol{D},\mathcal{S}}\cdot \lambda_{\max}\big(\boldsymbol{L}_i\big)\cdot \mathbb{E}\Big[\big\|h_i^{k+1} - h_i^k\big\|_{\boldsymbol{L}_i^{-1}}^2\Big].$$

# I   DISTRIBUTED DET-CGD

This section is a brief summary of the distributed det-CGD algorithm and its theoretical analysis. The details can be found in (Li et al., 2024). The algorithm follows the standard FL paradigm. See the pseudocode in Algorithm 3.

---
**Algorithm 3** Distributed det-CGD

---
1: **Input:** Starting point $x^0$, stepsize matrix $\boldsymbol{D}$, number of iterations $K$
2: **for** $k = 0, 1, 2, \ldots, K-1$ **do**
3:     The devices in parallel:
4:     sample $\boldsymbol{S}_i^k \sim \mathcal{S}$;
5:     compute $\boldsymbol{S}_i^k \nabla f_i(x^k)$;
6:     broadcast $\boldsymbol{S}_i^k \nabla f_i(x^k)$.
7:     The server:
8:     combines $g^k = \frac{1}{n}\sum_{i=1}^n \boldsymbol{S}_i^k \nabla f_i(x^k)$;
9:     computes $x^{k+1} = x^k - \boldsymbol{D}g^k$;
10:     broadcasts $x^{k+1}$.
11: **end for**
12: **Return:** $x^K$

---

**Theorem I.1.** *Suppose that $f$ is $\boldsymbol{L}$-smooth. Under the Assumptions 3.1, 3.3, if the stepsize satisfies*

$$\boldsymbol{DLD} \preceq \boldsymbol{D}, \tag{40}$$

*then the following convergence bound is true for the iteration of Algorithm 3:*

$$\min_{0 \leq k \leq K-1} \mathbb{E}\Big[\big\|\nabla f(x^k)\big\|_{\frac{\boldsymbol{D}}{\det(\boldsymbol{D})^{1/d}}}^2\Big] \leq \frac{2(1 + \frac{\lambda_{\boldsymbol{D}}}{n})^K\big(f(x^0) - f^\star\big)}{\det(\boldsymbol{D})^{1/d} K} + \frac{2\lambda_{\boldsymbol{D}}\Delta^\star}{\det(\boldsymbol{D})^{1/d} n}, \tag{41}$$

*where $\Delta^\star := f^\star - \frac{1}{n}\sum_{i=1}^n f_i^\star$ and*

$$\lambda_{\boldsymbol{D}} := \max_i\Big\{\lambda_{\max}\Big(\mathbb{E}\Big[\boldsymbol{L}_i^{\frac{1}{2}}\big(\boldsymbol{S}_i^k - \boldsymbol{I}_d\big)\boldsymbol{DLD}\big(\boldsymbol{S}_i^k - \boldsymbol{I}_d\big)\boldsymbol{L}_i^{\frac{1}{2}}\Big]\Big)\Big\}.$$

*Remark* I.2. On the right hand side of (41) we observe that increasing $K$ will only reduce the first term, that corresponds to the convergence error. Whereas, the second term, which does not depend on $K$, will remain constant, if the other parameters of the algorithm are fixed. This testifies to the neighborhood phenomenon which we discussed in Section 2.

*Remark* I.3. If the stepsize satisfies the below conditions,

$$\boldsymbol{DLD} \preceq \boldsymbol{D}, \quad \lambda_{\boldsymbol{D}} \leq \min\left\{\frac{n}{K}, \frac{n\varepsilon^2}{4\Delta^\star} \det(\boldsymbol{D})^{1/d}\right\}, \quad K \geq \frac{12(f(x^0) - f^\star)}{\det(\boldsymbol{D})^{1/d}\,\varepsilon^2}, \qquad (42)$$

then we obtain $\varepsilon$-stationary point.

One can see that in the convergence guarantee of det-CGD in the distributed case, the result (41) is not variance-reduced. Because of this, in order to reach a $\varepsilon$ stationary point, the stepsize condition in (42) is restrictive.

## J EXTENSION OF DET-CGD2 IN MARINA FORM

In this section we want to extend det-CGD2 into its variance reduced counterpart in MARINA form.

### J.1 EXTENSION OF DET-CGD2 TO ITS VARIANCE REDUCED COUNTERPART

---

**Algorithm 4** det-CGD2-VR

1: **Input:** starting point $x^0$, stepsize matrix $\boldsymbol{D}$, probability $p \in (0, 1]$, number of iterations $K$
2: Initialize $g^0 = \boldsymbol{D} \cdot \nabla f(x^0)$
3: **for** $k = 0, 1, \ldots, K - 1$ **do**
4:     Sample $c_k \sim \text{Be}(p)$
5:     Broadcast $g^k$ to all workers
6:     **for** $i = 1, 2, \ldots$ in parallel **do**
7:         $x^{k+1} = x^k - g^k$
8:         Set $g_i^{k+1} = \begin{cases} \boldsymbol{D} \cdot \nabla f_i(x^{k+1}) & \text{if } c_k = 1 \\ g^k + \boldsymbol{T}_i^k \boldsymbol{D}\left(\nabla f_i(x^{k+1}) - \nabla f_i(x^k)\right) & \text{if } c_k = 0 \end{cases}$
9:     **end for**
10:    $g^{k+1} = \frac{1}{n}\sum_{i=1}^n g_i^{k+1}$
11: **end for**
12: **Return:** $\tilde{x}^K$ chosen uniformly at random from $\{x^k\}_{k=0}^{K-1}$

---

We call det-MARINA as the extension of det-CGD1, and Algorithm 4 as the extension of det-CGD2 due to the difference in the order of applying sketches and stepsize matrices. The key difference between det-CGD1 and det-CGD2 is that in det-CGD1 the gradient is sketched first and then multiplied by the stepsize, while for det-CGD2, the gradient is multiplied by the stepsize first after which the product is sketched. The convergence for Algorithm 4 can be obtained in a similar manner as Theorem 4.1.

**Theorem J.1.** *Let Assumptions 3.1 and 3.3 hold, with the gradient of $f$ being $\boldsymbol{L}$-Lipschitz. If the stepsize matrix $\boldsymbol{D} \in \mathbb{S}_{++}^d$ satisfies*

$$\boldsymbol{D}^{-1} \succeq \left(\frac{(1-p)\cdot R'(\boldsymbol{D}, \mathcal{S})}{np} + 1\right)\boldsymbol{L},$$

*where*

$$R'(\boldsymbol{D}, \mathcal{S}) = \frac{1}{n}\sum_{i=1}^n \lambda_{\max}\left(\boldsymbol{D}\mathbb{E}\left[\boldsymbol{T}_i^k \boldsymbol{D}^{-1}\boldsymbol{T}_i^k\right]\boldsymbol{D}\boldsymbol{L}_i^{\frac{1}{2}} - \boldsymbol{L}_i^{\frac{1}{2}}\boldsymbol{D}\right)\cdot\lambda_{\max}\left(\boldsymbol{L}_i\right)\cdot\lambda_{\max}\left(\boldsymbol{L}^{-\frac{1}{2}}\boldsymbol{L}_i\boldsymbol{L}^{-\frac{1}{2}}\right).$$

*Then after $K$ iterations of Algorithm 4, we have*

$$\mathbb{E}\left[\left\|\nabla f(\tilde{x}^K)\right\|_{\frac{\boldsymbol{D}}{\det(\boldsymbol{D})^{1/d}}}^2\right] \leq \frac{2\left(f(x^0) - f^\star\right)}{\det(\boldsymbol{D})^{1/d}\cdot K}.$$

*This is to say that in order to reach a $\varepsilon$-stationary point, we require $K \geq \frac{2(f(x^0) - f^\star)}{\det(\boldsymbol{D})^{1/d}\cdot\varepsilon^2}$.*

If we look at the scalar case where $\boldsymbol{D} = \gamma \cdot \boldsymbol{I}_d$, $\boldsymbol{L}_i = L_i \cdot \boldsymbol{I}_d$ and $\boldsymbol{L} = L \cdot I_d$, then the condition in Theorem J.1 reduces to

$$\frac{(1-p)\omega L^2}{np} + L - \frac{1}{\gamma} \leq 0. \tag{43}$$

Notice that here $\omega = \lambda_{\max}\left(\mathbb{E}\left[\left(\boldsymbol{T}_i^k\right)^2\right]\right) - 1$, and we have $L^2 = \frac{1}{n}\sum_{i=1}^n L_i^2$, which is due to Lemma F.6. This condition coincides with the condition for convergence of MARINA. One may also check that, the update rule in Algorithm 4, is the same as MARINA in the scalar case. However, the condition given in Theorem J.1 is not simpler than Theorem 4.1, contrary to the single-node case. We emphasize that Algorithm 4 is not suitable for the federated learning setting where the clients have limited resources. In order to perform the update, each client is required to store the stepsize matrix $\boldsymbol{D}$ which is of size $d \times d$. In the over-parameterized regime, the dataset size is $m \times d$ where $m$ is the number of data samples, and we have $d > m$. This means that the stepsize matrix each client needs to store is even larger than the dataset itself, which is unacceptable given the limited resources each client has.

We first present two lemmas which are necessary for the proofs of Theorem J.1.

**Lemma J.2.** *Assume that function $f$ is $\boldsymbol{L}$-smooth, and $x^{k+1} = x^k - g^k$, and matrix $\boldsymbol{D} \in \mathbb{S}_{++}^d$. Then the iterates generated by Algorithm 4 satisfy the following inequality:*

$$f(x^{k+1}) \leq f(x^k) - \frac{1}{2}\left\|\nabla f(x^k)\right\|_{\boldsymbol{D}}^2 + \frac{1}{2}\left\|\boldsymbol{D} \cdot \nabla f(x^k) - g^k\right\|_{\boldsymbol{D}^{-1}}^2 - \frac{1}{2}\left\|x^{k+1} - x^k\right\|_{\boldsymbol{D}^{-1}-\boldsymbol{L}}^2.$$

**Lemma J.3.** *For any sketch matrix $\boldsymbol{T} \in \mathbb{S}_+^d$, vector $t \in \mathbb{R}^d$, matrix $\boldsymbol{D} \in \mathbb{S}_{++}^d$ and matrix $\boldsymbol{L} \in \mathbb{S}_{++}^d$, we have*

$$\mathbb{E}\left[\left\|\boldsymbol{T}\boldsymbol{D}t - \boldsymbol{D}t\right\|_{\boldsymbol{D}^{-1}}^2\right] \leq \lambda_{\max}\left(\boldsymbol{L}^{\frac{1}{2}}\boldsymbol{D}\mathbb{E}\left[\boldsymbol{T}\boldsymbol{D}^{-1}\boldsymbol{T}\right]\boldsymbol{D}\boldsymbol{L}^{\frac{1}{2}} - \boldsymbol{L}^{\frac{1}{2}}\boldsymbol{D}\boldsymbol{L}^{\frac{1}{2}}\right)\|t\|_{\boldsymbol{L}^{-1}}^2. \tag{44}$$

### J.2 PROOF OF THEOREM J.1

We start with Lemma J.2,

$$\mathbb{E}\left[f(x^{k+1})\right] \leq \mathbb{E}\left[f(x^k)\right] - \mathbb{E}\left[\frac{1}{2}\left\|\nabla f(x^k)\right\|_{\boldsymbol{D}}^2\right]$$
$$+ \mathbb{E}\left[\frac{1}{2}\left\|\boldsymbol{D} \cdot \nabla f(x^k) - g^k\right\|_{\boldsymbol{D}^{-1}}^2\right] - \mathbb{E}\left[\frac{1}{2}\left\|x^{k+1} - x^k\right\|_{\boldsymbol{D}^{-1}-\boldsymbol{L}}^2\right]. \tag{45}$$

Now we look at the term $\mathbb{E}\left[\left\|\boldsymbol{D} \cdot \nabla f(x^{k+1}) - g^{k+1}\right\|_{\boldsymbol{D}^{-1}}^2\right]$. Recall that $g^k$ here is given by

$$g^{k+1} = \begin{cases} \boldsymbol{D} \cdot \nabla f(x^{k+1}) & \text{with probability } p \\ g^k + \frac{1}{n}\sum_{i=1}^n \boldsymbol{T}_i^k \boldsymbol{D}\left(\nabla f_i(x^{k+1}) - \nabla f_i(x^k)\right) & \text{with probability } 1-p. \end{cases}$$

As a result, we have

$$\mathbb{E}\left[\left\|g^{k+1} - \boldsymbol{D}\nabla f(x^{k+1})\right\|_{\boldsymbol{D}^{-1}}^2 \mid x^{k+1}, x^k\right]$$
$$= \mathbb{E}\left[\mathbb{E}\left[\left\|g^{k+1} - \boldsymbol{D}\nabla f(x^{k+1})\right\|_{\boldsymbol{D}^{-1}}^2 \mid x^{k+1}, x^k, c_k\right]\right]$$
$$= (1-p) \cdot \mathbb{E}\left[\left\|g^k + \frac{1}{n}\sum_{i=1}^n \boldsymbol{T}_i^k \boldsymbol{D}\left(\nabla f_i(x^{k+1}) - \nabla f_i(x^k)\right) - \boldsymbol{D}\nabla f(x^{k+1})\right\|_{\boldsymbol{D}^{-1}}^2 \mid x^{k+1}, x^k\right].$$

For the sake of presentation, we use $\mathbb{E}_k[\cdot]$ to denote the conditional expectation $\mathbb{E}[\cdot \mid x_k, x_{k+1}]$ on $x_k, x_{k+1}$. Using Fact E.2 with $x = \frac{1}{n}\sum_{i=1}^n \boldsymbol{T}_i^k \boldsymbol{D}\left(\nabla f_i(x^{k+1}) - \nabla f_i(x^k)\right), c = \boldsymbol{D}\nabla f(x^{k+1}) - g^k$,

we obtain:

$$(1-p)\mathbb{E}_k\left[\left\|g^k + \frac{1}{n}\sum_{i=1}^n \boldsymbol{T}_i^k \boldsymbol{D}\left(\nabla f_i(x^{k+1}) - \nabla f_i(x^k)\right) - \boldsymbol{D}\nabla f(x^{k+1})\right\|_{\boldsymbol{D}^{-1}}^2\right]$$

$$= (1-p)\mathbb{E}_k\left[\left\|\frac{1}{n}\sum_{i=1}^n \boldsymbol{T}_i^k \boldsymbol{D}\left(\nabla f_i(x^{k+1}) - \nabla f_i(x^k)\right) - \boldsymbol{D}\left(\nabla f(x^{k+1}) - \nabla f(x^k)\right)\right\|_{\boldsymbol{D}^{-1}}^2\right]$$

$$+ (1-p)\left\|g^k - \nabla f(x^k)\right\|_{\boldsymbol{D}^{-1}}^2$$

$$= (1-p)\mathbb{E}_k\left[\left\|\frac{1}{n}\sum_{i=1}^n \left[\boldsymbol{T}_i^k \boldsymbol{D}\left(\nabla f_i(x^{k+1}) - \nabla f_i(x^k)\right) - \boldsymbol{D}\left(\nabla f_i(x^{k+1}) - \nabla f_i(x^k)\right)\right]\right\|_{\boldsymbol{D}^{-1}}^2\right]$$

$$+ (1-p)\left\|g^k - \nabla f(x^k)\right\|_{\boldsymbol{D}^{-1}}^2.$$

The following identity holds due to the unbiasedness,

$$\mathbb{E}_k\left[\boldsymbol{T}_i^k \boldsymbol{D}(\nabla f_i(x^{k+1}) - \nabla f_i(x^k))\right] = \boldsymbol{D}(\nabla f_i(x^{k+1}) - \nabla f_i(x^k)),$$

and any two random vectors in the set $\left\{\boldsymbol{T}_i^k \boldsymbol{D}(\nabla f_i(x^{k+1}) - \nabla f_i(x^k))\right\}_{i=1}^n$ are independent if $x^{k+1}, x^k$ are fixed. As a result

$$\mathbb{E}_k\left[\left\|g^{k+1} - \boldsymbol{D}\nabla f(x^{k+1})\right\|_{\boldsymbol{D}^{-1}}^2\right]$$

$$= \frac{1-p}{n^2}\sum_{i=1}^n \mathbb{E}_k\left[\left\|\boldsymbol{T}_i^k\left(\boldsymbol{D}\nabla f_i(x^{k+1}) - \boldsymbol{D}\nabla f_i(x^k)\right) - \left(\boldsymbol{D}\nabla f_i(x^{k+1}) - \boldsymbol{D}\nabla f_i(x^k)\right)\right\|_{\boldsymbol{D}^{-1}}^2\right]$$

$$+ (1-p)\cdot\left\|g^k - \boldsymbol{D}\nabla f(x^k)\right\|_{\boldsymbol{D}^{-1}}^2. \tag{46}$$

For each term within the summation, we further upper bound it using Lemma J.3

$$\mathbb{E}_k\left[\left\|\boldsymbol{T}_i^k\left(\boldsymbol{D}\nabla f_i(x^{k+1}) - \boldsymbol{D}\nabla f_i(x^k)\right) - \left(\boldsymbol{D}\nabla f_i(x^{k+1}) - \boldsymbol{D}\nabla f_i(x^k)\right)\right\|_{\boldsymbol{D}^{-1}}^2\right]$$

$$\leq \lambda_{\max}\left(\boldsymbol{L}_i^{\frac{1}{2}}\boldsymbol{D}\mathbb{E}\left[\boldsymbol{T}_i^k\boldsymbol{D}^{-1}\boldsymbol{T}_i^k\right]\boldsymbol{D}\boldsymbol{L}_i^{\frac{1}{2}} - \boldsymbol{L}_i^{\frac{1}{2}}\boldsymbol{D}\boldsymbol{L}_i^{\frac{1}{2}}\right)\left\|\nabla f_i(x^{k+1}) - \nabla f_i(x^k)\right\|_{\boldsymbol{L}_i^{-1}}^2$$

$$\leq \lambda_{\max}\left(\boldsymbol{L}_i^{\frac{1}{2}}\boldsymbol{D}\mathbb{E}\left[\boldsymbol{T}_i^k\boldsymbol{D}^{-1}\boldsymbol{T}_i^k\right]\boldsymbol{D}\boldsymbol{L}_i^{\frac{1}{2}} - \boldsymbol{L}_i^{\frac{1}{2}}\boldsymbol{D}\boldsymbol{L}_i^{\frac{1}{2}}\right)\left\|x^{k+1} - x^k\right\|_{\boldsymbol{L}_i}^2,$$

where the last inequality is due to Assumption 3.3. Plugging this back into (46), we obtain

$$\mathbb{E}_k\left[\left\|g^{k+1} - \boldsymbol{D}\nabla f(x^{k+1})\right\|_{\boldsymbol{D}^{-1}}^2\right]$$

$$\leq \frac{1-p}{n^2}\sum_{i=1}^n \lambda_{\max}\left(\boldsymbol{L}_i^{\frac{1}{2}}\boldsymbol{D}\mathbb{E}\left[\boldsymbol{T}_i^k\boldsymbol{D}^{-1}\boldsymbol{T}_i^k\right]\boldsymbol{D}\boldsymbol{L}_i^{\frac{1}{2}} - \boldsymbol{L}_i^{\frac{1}{2}}\boldsymbol{D}\boldsymbol{L}_i^{\frac{1}{2}}\right)\left\|x^{k+1} - x^k\right\|_{\boldsymbol{L}_i}^2$$

$$+ (1-p)\cdot\left\|g^k - \boldsymbol{D}\nabla f(x^k)\right\|_{\boldsymbol{D}^{-1}}^2.$$

Similarly to Theorem 4.1, we obtain

$$\mathbb{E}_k\left[\left\|g^{k+1} - \boldsymbol{D}\nabla f(x^{k+1})\right\|_{\boldsymbol{D}^{-1}}^2\right]$$

$$\leq \frac{1-p}{n^2}\sum_{i=1}^n \lambda_{\max}\left(\boldsymbol{L}_i^{\frac{1}{2}}\boldsymbol{D}\mathbb{E}\left[\boldsymbol{T}_i^k\boldsymbol{D}^{-1}\boldsymbol{T}_i^k\right]\boldsymbol{D}\boldsymbol{L}_i^{\frac{1}{2}} - \boldsymbol{L}_i^{\frac{1}{2}}\boldsymbol{D}\boldsymbol{L}_i^{\frac{1}{2}}\right)$$

$$\times \left\langle \boldsymbol{L}^{\frac{1}{2}}\left(x^{k+1} - x^k\right), \left(\boldsymbol{L}^{-\frac{1}{2}}\boldsymbol{L}_i\boldsymbol{L}^{-\frac{1}{2}}\right)\cdot\boldsymbol{L}^{\frac{1}{2}}\left(x^{k+1} - x^k\right)\right\rangle + (1-p)\cdot\left\|g^k - \boldsymbol{D}\nabla f(x^k)\right\|_{\boldsymbol{D}^{-1}}^2$$

$$\leq \frac{1-p}{n^2}\sum_{i=1}^n \lambda_{\max}\left(\boldsymbol{L}_i^{\frac{1}{2}}\left(\boldsymbol{D}\mathbb{E}\left[\boldsymbol{T}_i^k\boldsymbol{D}^{-1}\boldsymbol{T}_i^k\right]\boldsymbol{D} - \boldsymbol{D}\right)\boldsymbol{L}_i^{\frac{1}{2}}\right)\cdot\lambda_{\max}\left(\boldsymbol{L}^{-\frac{1}{2}}\boldsymbol{L}_i\boldsymbol{L}^{-\frac{1}{2}}\right)\left\|x^{k+1} - x^k\right\|_{\boldsymbol{L}}^2$$

$$+ (1-p)\cdot\left\|g^k - \boldsymbol{D}\nabla f(x^k)\right\|_{\boldsymbol{D}^{-1}}^2.$$

Applying Fact E.5, we obtain

$$\mathbb{E}_k\left[\left\|g^{k+1} - \boldsymbol{D}\nabla f(x^{k+1})\right\|_{\boldsymbol{D}^{-1}}^2\right]$$

$$\leq \frac{1-p}{n^2}\sum_{i=1}^n \lambda_{\max}\left(\boldsymbol{D}\mathbb{E}\left[\boldsymbol{T}_i^k \boldsymbol{D}^{-1}\boldsymbol{T}_i^k\right]\boldsymbol{D} - \boldsymbol{D}\right)\lambda_{\max}\left(\boldsymbol{L}_i\right)\lambda_{\max}\left(\boldsymbol{L}^{-\frac{1}{2}}\boldsymbol{L}_i\boldsymbol{L}^{-\frac{1}{2}}\right)\left\|x^{k+1} - x^k\right\|_{\boldsymbol{L}}^2$$

$$+ (1-p)\cdot\left\|g^k - \boldsymbol{D}\nabla f(x^k)\right\|_{\boldsymbol{D}^{-1}}^2.$$

Using the definition of $R'(\boldsymbol{D}, \mathcal{S})$, we further simplify it to

$$\mathbb{E}_k\left[\left\|g^{k+1} - \boldsymbol{D}\nabla f(x^{k+1})\right\|_{\boldsymbol{D}^{-1}}^2\right]$$

$$\leq \frac{(1-p)\cdot R'(\boldsymbol{D}, \mathcal{S})}{n}\left\|x^{k+1} - x^k\right\|_{\boldsymbol{L}}^2 + (1-p)\cdot\left\|g^k - \boldsymbol{D}\nabla f(x^k)\right\|_{\boldsymbol{D}^{-1}}^2.$$

Taking expectation again and using the tower property, we have

$$\mathbb{E}\left[\left\|g^{k+1} - \boldsymbol{D}\nabla f(x^{k+1})\right\|_{\boldsymbol{D}^{-1}}^2\right] \tag{47}$$

$$\leq (1-p)\left(\frac{\cdot R'(\boldsymbol{D}, \mathcal{S})}{n}\mathbb{E}\left[\left\|x^{k+1} - x^k\right\|_{\boldsymbol{L}}^2\right] + \cdot\mathbb{E}\left[\left\|g^k - \boldsymbol{D}\nabla f(x^k)\right\|_{\boldsymbol{D}^{-1}}^2\right]\right). \tag{48}$$

Consider the Lyapunov function $\Phi_k = \Phi_k = f(x^k) - f^\star + \frac{1}{2p}\left\|g^k - \boldsymbol{D}\nabla f(x^k)\right\|_{\boldsymbol{D}^{-1}}^2$. Using (45) and (47), we have

$$\mathbb{E}[\Phi_{k+1}]$$

$$\leq \mathbb{E}\left[f(x^k) - f^\star\right] - \frac{1}{2}\mathbb{E}\left[\left\|\nabla f(x^k)\right\|_{\boldsymbol{D}}^2\right] + \frac{1}{2}\mathbb{E}\left[\left\|g^k - \boldsymbol{D}\nabla f(x^k)\right\|_{\boldsymbol{D}^{-1}}^2\right]$$

$$- \frac{1}{2}\mathbb{E}\left[\left\|x^{k+1} - x^k\right\|_{\boldsymbol{D}^{-1}-L}\right] + \frac{1}{2p}\cdot\frac{(1-p)R'(\boldsymbol{D}, \mathcal{S})}{n}\mathbb{E}\left[\left\|x^{k+1} - x^k\right\|_{\boldsymbol{L}}^2\right]$$

$$+ \frac{1-p}{2p}\mathbb{E}\left[\left\|g^k - \boldsymbol{D}\nabla f(x^k)\right\|_{\boldsymbol{D}^{-1}}^2\right]$$

$$= \mathbb{E}[\Phi_k] - \frac{1}{2}\mathbb{E}\left[\left\|\nabla f(x^k)\right\|_{\boldsymbol{D}}^2\right]$$

$$+ \frac{1}{2}\left(\frac{(1-p)R'(\boldsymbol{D}, \mathcal{S})}{np}\mathbb{E}\left[\left\|x^{k+1} - x^k\right\|_{\boldsymbol{L}}^2\right] - \mathbb{E}\left[\left\|x^{k+1} - x^k\right\|_{\boldsymbol{D}^{-1}-L}^2\right]\right).$$

Now, notice that the last term in the above inequality is non-positive as guaranteed by the condition

$$\boldsymbol{D}^{-1} \succeq \left(\frac{(1-p)R'(\boldsymbol{D}, \mathcal{S})}{np} + 1\right)\boldsymbol{L}.$$

This leads to the following recurrence after ignoring the last term,

$$\mathbb{E}[\Phi_{k+1}] \leq \mathbb{E}[\Phi_k] - \frac{1}{2}\mathbb{E}\left[\left\|\nabla f(x^k)\right\|_{\boldsymbol{D}}^2\right].$$

Unrolling this recurrence, we get

$$\frac{1}{K}\sum_{k=0}^{K-1}\mathbb{E}\left[\left\|\nabla f(x^k)\right\|_{\boldsymbol{D}}^2\right] \leq \frac{2\left(\mathbb{E}[\Phi_0] - \mathbb{E}[\Phi_K]\right)}{K}.$$

The left hand side can viewed as average over $\tilde{x}^K$, which is drawn uniformly at random from $\{x_k\}_{k=0}^{K-1}$, while the right hand side can be simplified as

$$\frac{2\left(\mathbb{E}[\Phi_0] - \mathbb{E}[\Phi_K]\right)}{K} \leq \frac{2\Phi_0}{K} = \frac{2\left(f(x^0) - f^\star + \frac{1}{2p}\left\|g^0 - \nabla f(x^0)\right\|_{\boldsymbol{D}}^2\right)}{K}.$$

Recall that $g^0 = \nabla f(x^0)$, we obtain

$$\mathbb{E}\left[\left\|\nabla f(\tilde{x}^K)\right\|_{\frac{\boldsymbol{D}}{\det(\boldsymbol{D})^{1/d}}}^2\right] \leq \frac{2\left(f(x^0) - f^\star\right)}{\det(\boldsymbol{D})^{1/d}K}.$$

### J.3 PROOF OF LEMMA J.2

From Lemma F.6, we know that $f$ is $\boldsymbol{L}$-smooth. Define $\bar{x}^{k+1} := x^k - \boldsymbol{D} \cdot \nabla f(x^k)$. Using $\boldsymbol{L}$-smoothness, we have

$$f(x^{k+1}) \le f(x^k) + \langle \nabla f(x^k), x^{k+1} - x^k \rangle + \frac{1}{2} \langle x^{k+1} - x^k, \boldsymbol{L}(x^{k+1} - x^k) \rangle$$

$$= f(x^k) + \langle \nabla f(x^k) - \boldsymbol{D}^{-1} \cdot g^k, x^{k+1} - x^k \rangle + \langle \boldsymbol{D}^{-1} \cdot g^k, x^{k+1} - x^k \rangle$$

$$= + \frac{1}{2} \langle x^{k+1} - x^k, \boldsymbol{L}(x^{k+1} - x^k) \rangle$$

$$= f(x^k) + \langle \nabla f(x^k) - \boldsymbol{D}^{-1} \cdot g^k, -g^k \rangle - \langle x^{k+1} - x^k, \boldsymbol{D}^{-1}(x^{k+1} - x^k) \rangle$$

$$+ \frac{1}{2} \langle x^{k+1} - x^k, \boldsymbol{L}(x^{k+1} - x^k) \rangle.$$

Simplify the above inner-products we have,

$$f(x^{k+1})$$

$$\le f(x^k) + \langle \nabla f(x^k) - \boldsymbol{D}^{-1} \cdot g^k, -g^k \rangle - \left\langle x^{k+1} - x^k, \left( \boldsymbol{D}^{-1} - \frac{1}{2}\boldsymbol{L} \right)(x^{k+1} - x^k) \right\rangle.$$

We then add and subtract $\langle \nabla f(x^k) - \boldsymbol{D}^{-1} \cdot g^k, \boldsymbol{D} \cdot \nabla f(x^k) \rangle$,

$$f(x^{k+1})$$

$$\le f(x^k) + \langle \nabla f(x^k) - \boldsymbol{D}^{-1} \cdot g^k, \boldsymbol{D} \cdot \nabla f(x^k) - g^k \rangle$$

$$- \langle \nabla f(x^k) - \boldsymbol{D}^{-1} \cdot g^k, \boldsymbol{D} \cdot \nabla f(x^k) \rangle - \left\langle x^{k+1} - x^k, \left( \boldsymbol{D}^{-1} - \frac{1}{2}\boldsymbol{L} \right)(x^{k+1} - x^k) \right\rangle$$

$$= f(x^k) + \left\| \nabla f(x^k) - \boldsymbol{D}^{-1} \cdot g^k \right\|_{\boldsymbol{D}}^2 - \langle \boldsymbol{D}^{-1}(x^{k+1} - \bar{x}^{k+1}), x^k - \bar{x}^{k+1} \rangle$$

$$- \left\langle x^{k+1} - x^k, \left( \boldsymbol{D}^{-1} - \frac{1}{2}\boldsymbol{L} \right)(x^{k+1} - x^k) \right\rangle.$$

Decomposing the inner product term,

$$f(x^{k+1})$$

$$\le f(x^k) + \left\| \boldsymbol{D}^{-1} \left( \boldsymbol{D} \cdot \nabla f(x^k) - g^k \right) \right\|_{\boldsymbol{D}}^2 - \left\langle x^{k+1} - x^k, \left( \boldsymbol{D}^{-1} - \frac{1}{2}\boldsymbol{L} \right)(x^{k+1} - x^k) \right\rangle$$

$$- \frac{1}{2} \left( \left\| x^{k+1} - \bar{x}^{k+1} \right\|_{\boldsymbol{D}^{-1}}^2 + \left\| x^k - \bar{x}^{k+1} \right\|_{\boldsymbol{D}^{-1}}^2 - \left\| x^{k+1} - x^k \right\|_{\boldsymbol{D}^{-1}}^2 \right)$$

$$= f(x^k) + \left\| \boldsymbol{D} \cdot \nabla f(x^k) - g^k \right\|_{\boldsymbol{D}^{-1}}^2 - \left\| x^{k+1} - x^k \right\|_{\boldsymbol{D}^{-1} - \frac{1}{2}\boldsymbol{L}}^2$$

$$- \frac{1}{2} \left( \left\| \boldsymbol{D} \cdot \nabla f(x^k) - g^k \right\|_{\boldsymbol{D}^{-1}}^2 + \left\| \boldsymbol{D} \cdot \nabla f(x^k) \right\|_{\boldsymbol{D}^{-1}}^2 - \left\| x^{k+1} - x^k \right\|_{\boldsymbol{D}^{-1}}^2 \right).$$

Therefore,

$$f(x^{k+1}) \le f(x^k) + \frac{1}{2} \left\| \boldsymbol{D}\nabla f(x^k) - g^k \right\|_{\boldsymbol{D}^{-1}}^2 - \frac{1}{2} \left\| \nabla f(x^k) \right\|_{\boldsymbol{D}}^2 - \frac{1}{2} \left\| x^{k+1} - x^k \right\|_{\boldsymbol{D}^{-1} - \boldsymbol{L}}^2.$$

### J.4 PROOF OF LEMMA J.3

We start with

$$\mathbb{E}\left[ \left\| \boldsymbol{T}\boldsymbol{D}t - \boldsymbol{D}t \right\|_{\boldsymbol{D}^{-1}}^2 \right] = \mathbb{E}\left[ \left\| (\boldsymbol{T} - \boldsymbol{I}_d)\boldsymbol{D}t \right\|_{\boldsymbol{D}^{-1}}^2 \right]$$

$$= \left\langle t, \mathbb{E}\left[ \boldsymbol{D}(\boldsymbol{T} - \boldsymbol{I}_d)\boldsymbol{D}^{-1}(\boldsymbol{T} - \boldsymbol{I}_d)\boldsymbol{D} \right] \cdot t \right\rangle$$

$$= \left\langle t, \boldsymbol{D} \left( \mathbb{E}[\boldsymbol{T}\boldsymbol{D}^{-1}\boldsymbol{T}] - \boldsymbol{D}^{-1} \right) \boldsymbol{D} \cdot t \right\rangle$$

$$= \left\langle \boldsymbol{L}^{-\frac{1}{2}}t, \boldsymbol{L}^{\frac{1}{2}}\boldsymbol{D} \left( \mathbb{E}[\boldsymbol{T}\boldsymbol{D}^{-1}\boldsymbol{T}] - \boldsymbol{D}^{-1} \right) \boldsymbol{D}\boldsymbol{L}^{\frac{1}{2}} \cdot \boldsymbol{L}^{-\frac{1}{2}}t \right\rangle$$

$$\le \lambda_{\max} \left( \boldsymbol{L}^{\frac{1}{2}}\boldsymbol{D}\mathbb{E}[\boldsymbol{T}\boldsymbol{D}^{-1}\boldsymbol{T}]\boldsymbol{D}\boldsymbol{L}^{\frac{1}{2}} - \boldsymbol{L}^{\frac{1}{2}}\boldsymbol{D}\boldsymbol{L}^{\frac{1}{2}} \right) \cdot \left\| \boldsymbol{L}^{-\frac{1}{2}}t \right\|^2$$

$$= \lambda_{\max} \left( \boldsymbol{L}^{\frac{1}{2}}\boldsymbol{D}\mathbb{E}[\boldsymbol{T}\boldsymbol{D}^{-1}\boldsymbol{T}]\boldsymbol{D}\boldsymbol{L}^{\frac{1}{2}} - \boldsymbol{L}^{\frac{1}{2}}\boldsymbol{D}\boldsymbol{L}^{\frac{1}{2}} \right) \cdot \left\| t \right\|_{\boldsymbol{L}^{-1}}^2.$$

## K  EXPERIMENTS

In this section, we present numerical experiments to support the theoretical results for det-MARINA and det-DASHA. The code for the experiments is available at https://anonymous.4open. science/r/detCGD-VR-Code-865B. All the experiment code is implemented in Python 3.11, utilizing the NumPy and SciPy libraries. The experiments were conducted on a machine equipped with an AMD Ryzen 9 5900HX processor (Radeon Graphics) running at 3.3 GHz, featuring 8 cores and 16 threads. The datasets from LibSVM (Chang & Lin, 2011), which represent non-IID real-world datasets, were randomly distributed across all clients.

### K.1  EXPERIMENT SETTING

We are interested in the following logistic regression problem with a non-convex regularizer.

$$
f(x) = \frac{1}{n} \sum_{i=1}^{n} f_i(x), \qquad f_i(x) = \frac{1}{m_i} \sum_{j=1}^{m_i} \log\left(1 + e^{-b_{i,j} \cdot \langle a_{i,j}, x \rangle}\right) + \lambda \cdot \sum_{t=1}^{d} \frac{x_t^2}{1 + x_t^2},
$$

where $x \in \mathbb{R}^d$ represents the model, and $(a_{i,j}, b_{i,j}) \in \mathbb{R}^d \times \{-1, 1\}$ denotes a data point in the dataset of client $i$, which has a size of $m_i$. The constant $\lambda > 0$ serves as the coefficient of the regularization term. For each function $f_i$, its Hessian is upper bounded by

$$
\boldsymbol{L}_i = \frac{1}{m_i} \sum_{i=1}^{m_i} \frac{a_i a_i^\top}{4} + 2\lambda \cdot \boldsymbol{I}_d.
$$

Therefore, the Hessian of $f$ is bounded by

$$
\boldsymbol{L} = \frac{1}{\sum_{i=1}^{n} m_i} \sum_{i=1}^{n} \sum_{j=1}^{m_i} \frac{a_i a_i^\top}{4} + 2\lambda \cdot \boldsymbol{I}_d.
$$

Due to Lemma F.1, $f_i$ and $f$ satisfy Definition 3.2 (Matrix Lipschitz Gradient) with $\boldsymbol{L}_i \in \mathbb{S}_{++}^d$ and $\boldsymbol{L} \in \mathbb{S}_{++}^d$, respectively.

### K.2  COMPARISON OF ALL RELEVANT METHODS

In this section, we compare all relevant methods to det-MARINA and det-DASHA, which include *(i)* DCGD with scalar stepsize $\gamma_2$, *(ii)* det-CGD with matrix stepsize $\boldsymbol{D}_3^*$, *(iii)* MARINA with scalar stepsize $\gamma_1$, *(iv)* DASHA with scalar stepsize $\gamma_4$, *(v)* det-MARINA with $\boldsymbol{D}_{\boldsymbol{L}^{-1}}^*$, *(vi)* det-DASHA with $\boldsymbol{D}_{\boldsymbol{L}^{-1}}^{**}$. Throughout the experiment, we set $\varepsilon = 0.01$, $\lambda = 0.9$ and $K = 10000$, rand-$\tau$ sketch is used as an example of the compressor.

As shown in Figure 2, the performance of det-DASHA and det-MARINA in terms of communication complexity surpasses that of their scalar counterparts, DASHA and MARINA, respectively. This highlights the efficiency of employing a matrix stepsize over a scalar stepsize. Furthermore, det-DASHA and det-MARINA demonstrate superior communication complexity in this case compared to det-CGD. Additionally, we observe evidence of variance reduction.

Note that the optimal stepsizes for det-CGD and DCGD require knowledge of the function value differences at $x^\star$. Additionally, these stepsizes are constrained by the number of iterations $K$ and the error $\varepsilon^2$. In contrast, the variance-reduced methods do not rely on such considerations, making them significantly more practical in general.

### K.3  IMPROVEMENT OF det-MARINA OVER MARINA

The purpose of this experiment is to compare the iteration complexity of MARINA with that of det-MARINA using rand-$\tau$ sketches, thereby demonstrating the improvements of det-MARINA over MARINA. According to Theorem C.1 from (Gorbunov et al., 2021), the optimal stepsize for MARINA is

$$
\gamma_1 = \frac{1}{L\left(1 + \sqrt{\frac{(1-p)\omega}{pn}}\right)}, \tag{49}
$$

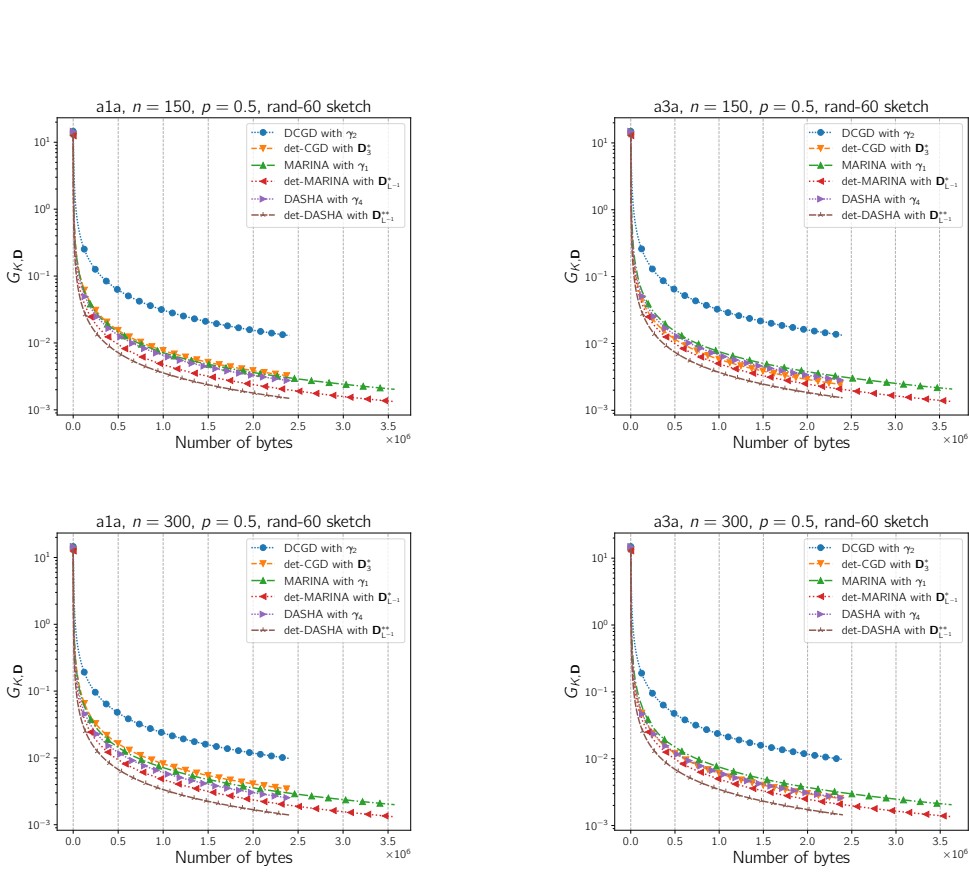

Figure 2: Comparison of DCGD with optimal scalar stepsize, det-CGD with matrix stepsize $D_3^*$, MARINA with optimal scalar stepsize, DASHA with optimal scalar stepsize, det-MARINA with optimal stepsize $D_{L^{-1}}^*$ and det-DASHA with optimal stepsize $D_{L^{-1}}^{**}$. Throughout the experiment, we are using rand-$\tau$ sketch with $\tau = 60$, and each algorithm is run for a fixed number of iterations $K = 10000$. The momentum of DASHA is set as $1/2\omega+1$ and det-DASHA is $1/2\omega_D+1$. The notation $n$ in the title stands for the number of clients in each case, and $p$ stands for the probability used by MARINA and det-MARINA.

where $\omega$ is the quantization coefficient. In particular, $\omega = \frac{d}{\tau} - 1$ for the rand-$\tau$ compressor. For further explanation, we refer the readers to Section 1.3 of (Gorbunov et al., 2021). The stepsize for det-MARINA is determined using Corollary 4.7. Below, we list some of the optimal stepsizes corresponding to different choices of $\boldsymbol{W}$, as used in the experimental section. Specifically, we have:

$$\boldsymbol{D}_{\boldsymbol{I}_d}^* = \frac{2}{1 + \sqrt{1 + 4\alpha\beta \frac{1}{\lambda_{\max}(\boldsymbol{L})} \cdot \omega}} \cdot \frac{\boldsymbol{I}_d}{\lambda_{\max}(\boldsymbol{L})},$$

$$\boldsymbol{D}_{\boldsymbol{L}^{-1}}^* = \frac{2}{1 + \sqrt{1 + 4\alpha\beta \cdot \lambda_{\max}\left(\mathbb{E}\left[\boldsymbol{S}_i^k \boldsymbol{L}^{-1} \boldsymbol{S}_i^k\right] - \boldsymbol{L}^{-1}\right)}} \cdot \boldsymbol{L}^{-1},$$

$$\boldsymbol{D}_{\operatorname{diag}^{-1}(\boldsymbol{L})}^* = \frac{2}{1 + \sqrt{1 + 4\alpha\beta \cdot \lambda_{\max}\left(\mathbb{E}\left[\boldsymbol{S}_i^k \operatorname{diag}^{-1}(\boldsymbol{L}) \boldsymbol{S}_i^k\right] - \operatorname{diag}^{-1}(\boldsymbol{L})\right)}} \cdot \operatorname{diag}^{-1}(\boldsymbol{L}) \quad (50)$$

Throughout the experiments, we set $\lambda = 0.3$. The $y$-axis in the figure represents the expectation of the corresponding matrix norm of the gradient of the function, defined as

$$G_{K,\boldsymbol{D}} = \mathbb{E}\left[\left\|\nabla f(\tilde{x}^K)\right\|_{\boldsymbol{D}/\det(\boldsymbol{D})^{1/d}}^2\right]. \quad (51)$$

Notice that for a fixed $\boldsymbol{D}$, we have

$$\lambda_{\min}\left(\frac{\boldsymbol{D}}{\det(\boldsymbol{D})^{1/d}}\right) \cdot \|\nabla f(x)\|^2 \leq \|\nabla f(x)\|_{\frac{\boldsymbol{D}}{\det(\boldsymbol{D})^{1/d}}}^2 \leq \lambda_{\max}\left(\frac{\boldsymbol{D}}{\det(\boldsymbol{D})^{1/d}}\right) \cdot \|\nabla f(x)\|^2.$$

which means that it is comparable to standard Euclidean norm once $\boldsymbol{D}$ is fixed.

As illustrated in Figure 3, det-MARINA consistently achieves a faster convergence rate compared to MARINA, provided they use the same sketch. This observation aligns with the results established in Corollary 6.1. Notably, in some cases, det-MARINA with a Rand-1 sketch even outperforms the standard MARINA with a Rand-80 sketch. This further underscores the superiority of matrix stepsizes and smoothness over the conventional scalar setting.

### K.4 IMPROVEMENT OF DET-MARINA OVER NON-VARIANCE-REDUCED METHODS

In this section, we compare two non-variance-reduced methods, distributed compressed gradient descent (DCGD) and distributed det-CGD, with two variance-reduced methods, MARINA and det-MARINA. In this experiment, Rand-1 sketch is used for all the algorithms. For the non-variance-reduced methods, $\varepsilon^2$ is fixed at 0.01 to determine the optimal stepsize. In our case, the optimal scalar stepsize for DCGD can be determined directly using Proposition 4 in (Khaled & Richtárik, 2023). To ensure that $\min_{0 \leq k \leq K-1} \mathbb{E}\left[\left\|\nabla f(x^k)\right\|^2\right] \leq \varepsilon^2$, the stepsize condition of DCGD in the non-convex case reduces to:

$$\gamma_2 \leq \min\left\{\frac{1}{L}, \sqrt{\frac{n}{\omega L L_{\max} K}}, \frac{n\varepsilon^2}{4 L L_{\max} \omega \cdot \Delta^\star}\right\},$$

where $L, L_i$ are the smoothness constants of $f, f_i$, respectively. We use $L_{\max} = \max_i L_i$, $K$ to denote the total number of iterations, and $\Delta^\star = f(x^\star) - \frac{1}{n}\sum_{i=1}^n f_i(x^\star)$. The constant $\omega$ is associated with the compressor used in the algorithm. For the rand-$\tau$ sketch, $\omega = \frac{d}{\tau} - 1$. In the case of distributed det-CGD, according to Li et al. (2024), the stepsize condition to satisfy $\min_{0 \leq k \leq K-1} \mathbb{E}\left[\left\|\nabla f(x^k)\right\|_{\boldsymbol{D}/\det(\boldsymbol{D})^{1/d}}^2\right] \leq \varepsilon^2$ is given by:

$$\boldsymbol{D}\boldsymbol{L}\boldsymbol{D} \preceq \boldsymbol{D}, \qquad \phi_{\boldsymbol{D}} \leq \min\left\{\frac{n}{K}, \frac{n\varepsilon^2}{4\Delta^\star}\det(\boldsymbol{D})^{1/d}\right\}, \quad (52)$$

where $\lambda_{\boldsymbol{D}}$ is defined as

$$\phi_{\boldsymbol{D}} = \max_i \left\{\lambda_{\max}\left(\mathbb{E}\left[\boldsymbol{L}_i^{\frac{1}{2}}\left(\boldsymbol{S}_i^k - \boldsymbol{I}_d\right)\boldsymbol{D}\boldsymbol{L}\boldsymbol{D}\left(\boldsymbol{S}_i^k - \boldsymbol{I}_d\right)\boldsymbol{L}_i^{\frac{1}{2}}\right]\right)\right\}. \quad (53)$$

In general, there is no straightforward way to determine an optimal stepsize matrix $\boldsymbol{D}$ that satisfies (52). Alternatively, we select the optimal diagonal stepsize $\boldsymbol{D}_3^*$, following a similar approach to

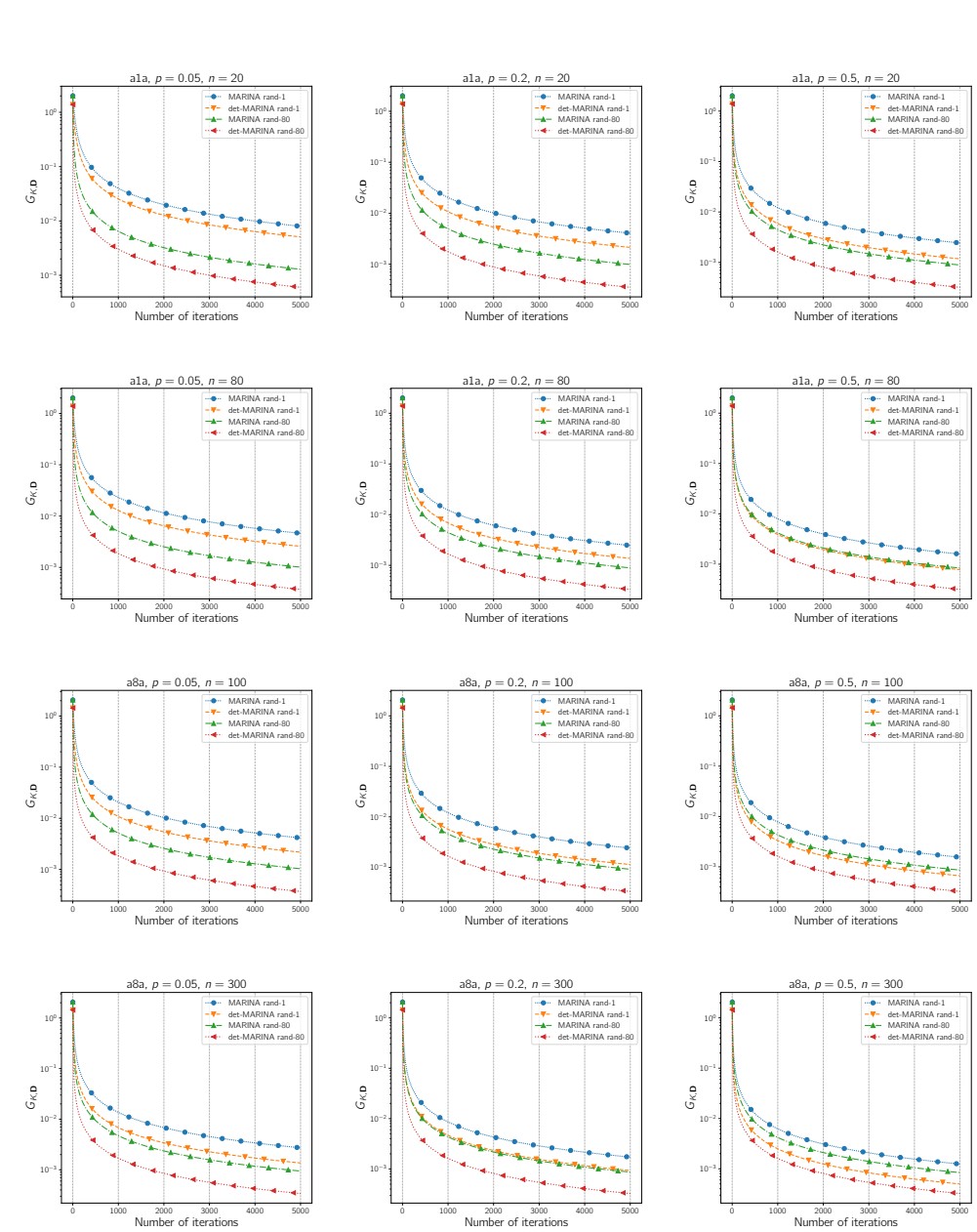

Figure 3: In this experiment, we compare det-MARINA with stepsize $\boldsymbol{D}_{\boldsymbol{L}^{-1}}^{*}$ to standard MARINA with the optimal scalar stepsize. Rand-$\tau$ compressor is used in the comparison. Throughout the experiments, $\lambda$ is fixed at $0.3$. The $x$-axis represents the number of iterations, while the $y$-axis denotes $G_{K,\boldsymbol{D}}$, as defined in (51), which is the averaged matrix norm of the gradient. The notation $p$ in the title denotes the probability used in the two algorithms, $n$ denotes the number of clients in each setting.

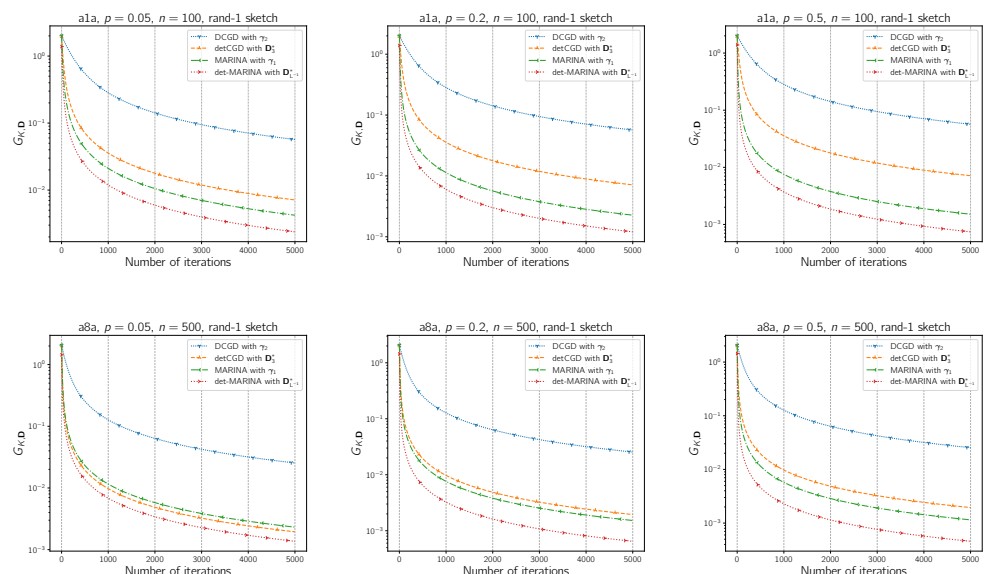

Figure 4: Comparison of DCGD with optimal scalar stepsize $\gamma_2$, det-CGD with optimal diagonal stepsize $\boldsymbol{D}_3^*$, MARINA with optimal scalar stepsize $\gamma_1$, and det-MARINA with optimal stepsize $\boldsymbol{D}_{\boldsymbol{L}^{-1}}^*$. The probability $p$ is selected from the set $\{0.05, 0.2, 0.5\}$ for MARINA and det-MARINA. $\lambda = 0.3$ is fixed throughout the experiment. The notation $n$ in the title indicates the number of clients in each case.

(Li et al., 2024). The stepsize condition for MARINA has already been described by (49). For det-MARINA, we fix $\boldsymbol{W} = \boldsymbol{L}^{-1}$ and use $\boldsymbol{D}_{\boldsymbol{L}^{-1}}^*$ as the stepsize matrix.

In Figure 4, each plot shows that det-MARINA outperforms MARINA as well as the non-variance-reduced methods. This result is anticipated, as our theoretical analysis confirms that det-MARINA achieves a better rate compared to MARINA, while the stepsizes of non-variance-reduced methods are adversely affected by the neighborhood. Furthermore, when $p$ is sufficiently large, the variance-reduced methods considered here consistently outperform the non-variance-reduced methods.

### K.5 Improvement of det-MARINA over det-CGD

In this section, we compare det-CGD in the distributed setting with det-MARINA, as both algorithms utilize matrix stepsizes and matrix smoothness. Throughout the experiment, $\lambda = 0.3$ is fixed, and for det-CGD, $\varepsilon^2 = 0.01$ is fixed to determine its stepsize. We first fix a matrix $\boldsymbol{W}$, selecting it from the set $\boldsymbol{L}^{-1}, \operatorname{diag}^{-1}(\boldsymbol{L}), \boldsymbol{I}_d$. Then, for each choice of $\boldsymbol{W}$, we determine the optimal scaling $\gamma_{\boldsymbol{W}}$ using the condition provided in (Li et al., 2024) (see (52) and (53)). The matrix stepsizes for det-CGD are defined as:

$$\boldsymbol{D}_1 = \gamma_{\boldsymbol{I}_d} \cdot \boldsymbol{I}_d, \qquad \boldsymbol{D}_2 = \gamma_{\operatorname{diag}^{-1}(\boldsymbol{L})} \cdot \operatorname{diag}^{-1}(\boldsymbol{L}), \qquad \boldsymbol{D}_3 = \gamma_{\boldsymbol{L}^{-1}} \cdot \boldsymbol{L}^{-1}. \tag{54}$$

For det-MARINA, we use the stepsize $\boldsymbol{D}_{\boldsymbol{L}^{-1}}^*$, as described in (50). In this experiment, we compare det-CGD with three stepsizes, $\boldsymbol{D}_1$, $\boldsymbol{D}_2$, and $\boldsymbol{D}_3$, against det-MARINA using the stepsize $\boldsymbol{D}_{\boldsymbol{L}^{-1}}^*$.

From Figure 5, it is evident that det-MARINA outperforms det-CGD with all matrix optimal stepsizes corresponding to the fixed choices of $\boldsymbol{W}$ considered here. This result is expected, as the convergence rate of non-variance-reduced methods is influenced by their neighborhood. This experiment highlights the advantages of det-MARINA over det-CGD and is consistent with our theoretical findings.

### K.6 det-MARINA with Different Stepsizes

As mentioned in Appendix K.3, for each choice of $\boldsymbol{W} \in \mathbb{S}_{++}^d$, an optimal stepsize $\boldsymbol{D}_{\boldsymbol{W}}^*$ can be determined. In this experiment, we compare det-MARINA with three different stepsize choices: $\boldsymbol{D}_{\boldsymbol{L}^{-1}}^*$,

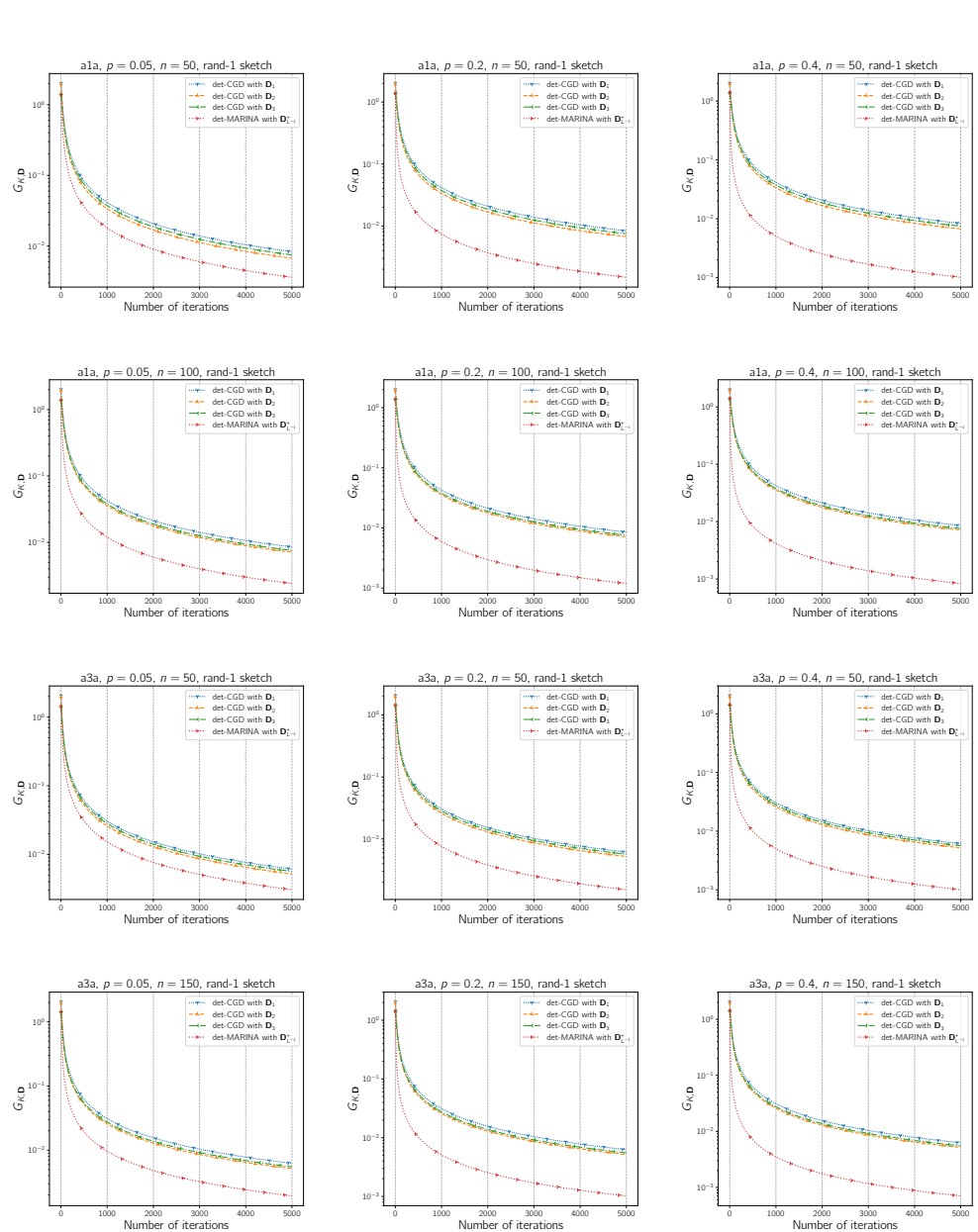

Figure 5: Comparison of det-CGD with matrix stepsize $D_1$, $D_2$ and $D_3$ and det-MARINA with optimal matrix stepsize when $W = L^{-1}$. The stepsizes $\{D_i\}_{i=1}^3$ are given in (54). Throughout the experiment $\varepsilon^2$ is fixed at 0.01. The notation $p$ in the title refers to the probability of det-MARINA, $n$ denotes the number of clients considered. Rand-1 sketch is used in all cases.

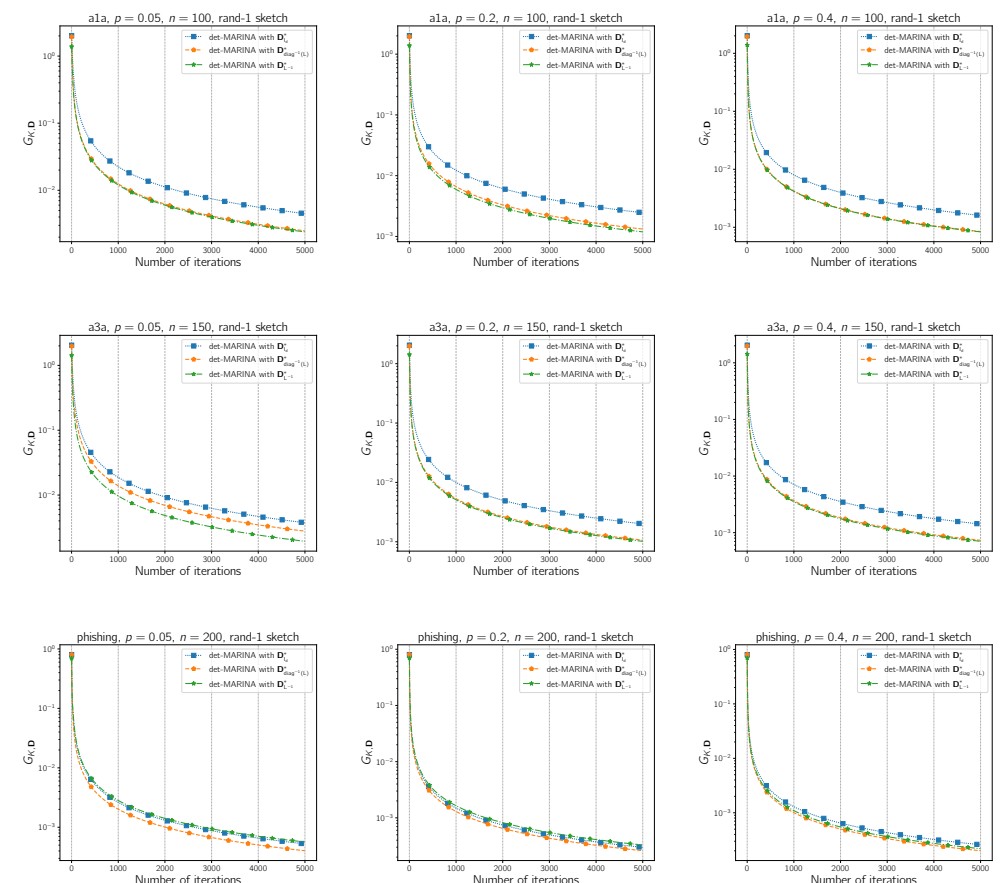

Figure 6: Comparison of det-MARINA with matrix stepsize $D_{I_d}^*$, $D_{\mathrm{diag}^{-1}(L)}^*$ and $D_{L^{-1}}^*$. The stepsizes are defined in (50). Throughout the experiment, $\lambda = 0.3$ is fixed. Rand-1 sketch is used in all cases. The notation $p$ indicates the probability of sending the true gradient in det-MARINA, $n$ denotes the number of clients considered.

$D_{\mathrm{diag}^{-1}(L)}^*$, and $D_{I_d}^*$. There stepsizes are explicitly defined in (50). Throughout the experiment, we fix $\lambda = 0.3$, and the rand-1 sketch is used in all cases.

As shown in Figure 6, in almost all cases det-MARINA with stepsize $D_{\mathrm{diag}^{-1}(L)}^*$ and $D_{L^{-1}}^*$ outperforms det-MARINA with $D_{I_d}^*$. Since det-MARINA with $D_{I_d}^*$ can be viewed as MARINA using a scalar stepsize under the matrix Lipschitz gradient assumption, this highlights the effectiveness of using a matrix stepsize over a scalar stepsize.

In Figure 6, there are cases where det-MARINA with $D_{\mathrm{diag}^{-1}(L)}^*$ outperforms $D_{L^{-1}}^*$. This suggests that these two stepsizes are perhaps incomparable in general cases. A similar observation can be made for det-CGD, where the optimal stepsizes corresponding to subspaces associated with a fixed $W$ are also incomparable.

### K.7 COMMUNICATION COMPLEXITY OF DET-MARINA

In this section, we examine how different probabilities $p$ influence the overall communication complexity of det-MARINA. We use $D_{L^{-1}}^*$ as the stepsize, determined based on the sketch employed (see (50)). Rand-$\tau$ sketches are utilized in these experiments, with the minibatch size $\tau$ varied to enable a more comprehensive comparison. For Rand-$\tau$ sketch $S$ and any matrix $A \in \mathbb{S}_{++}^d$, it can be

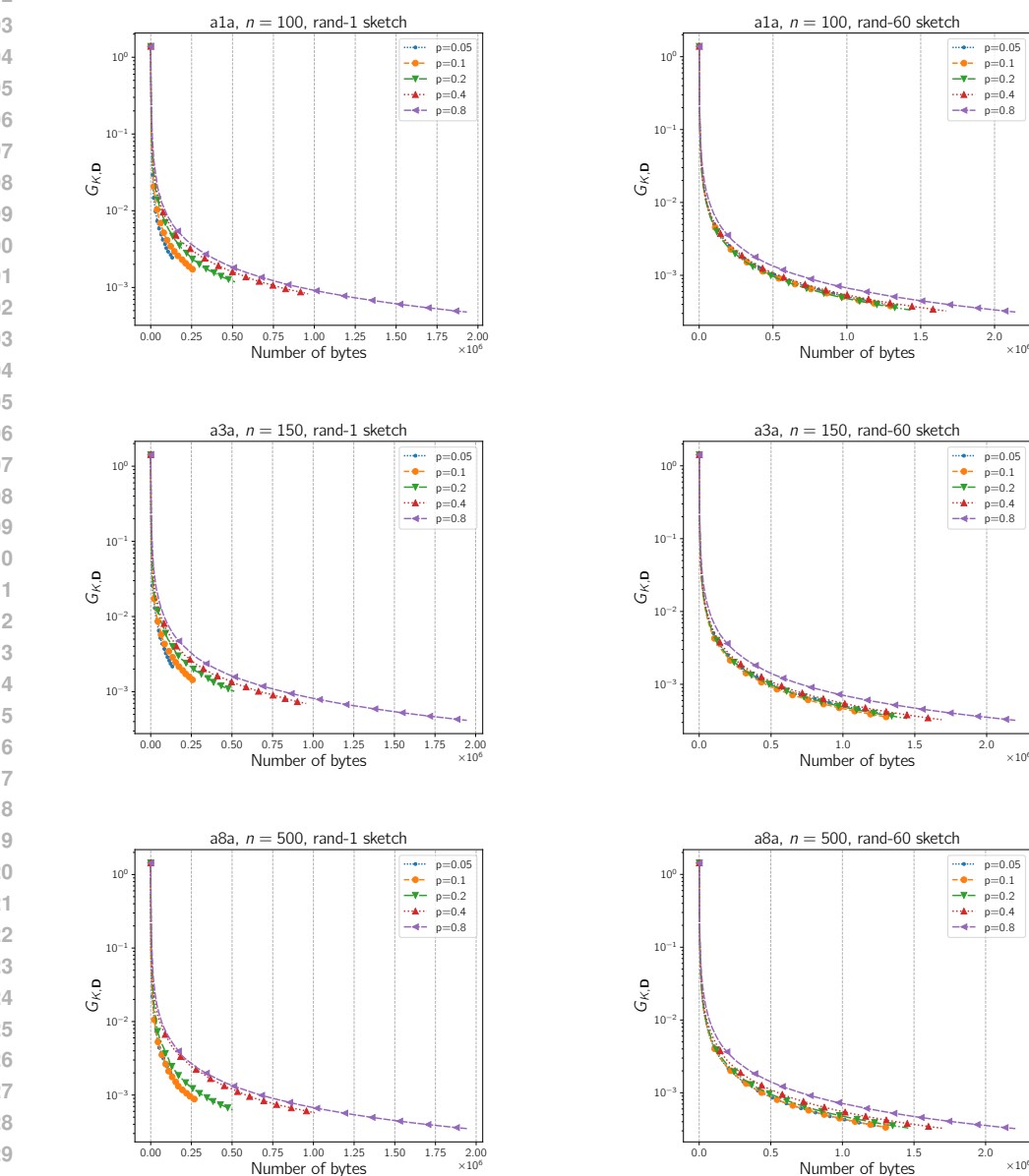

Figure 7: Comparison of det-MARINA with stepsize $\boldsymbol{D}^*_{\boldsymbol{L}^{-1}}$ using different probability $p$. The probability $p$ here is selected from the set $\{0.05, 0.1, 0.2, 0.4, 0.8\}$. The notation $n$ denotes the number of clients. The $x$-axis is the number of bytes sent from a single node to the server. In each case, we run det-MARINA for a fixed number of iterations $K = 5000$.

shown that

$$\mathbb{E}[\boldsymbol{SAS}] = \frac{d}{\tau}\left(\frac{d-\tau}{\mathrm{d}-1}\,\mathrm{diag}(\boldsymbol{A}) + \frac{\tau-1}{d-1}\boldsymbol{A}\right). \tag{55}$$

Combining (55) and (50), the corresponding matrix stepsize can be easily determined. In this experiment, we fix the total number of iterations to $K = 5000$.

As observed in Figure 7, for each dataset, the communication complexity tends to increase with a higher probability $p$. However, when the number of iterations is fixed, a larger $p$ often results in a faster convergence rate. This difference in communication complexity becomes more pronounced when using rand-1 sketch. In real-world federated learning scenarios, network bandwidth constraints

between clients and the server are common. Therefore, balancing communication complexity and iteration complexity—by carefully selecting the compression mechanism to ensure an acceptable speed that satisfies bandwidth limitations—becomes crucial.

## K.8 COMPARISON OF DASHA AND DET-DASHA

In this experiment, we compare the performance of original DASHA with det-DASHA. Throughout the experiments, $\lambda$ is fixed at $0.3$, and the same rand-$\tau$ sketch is used for both algorithms. For DASHA, setting the momentum as $a = \frac{1}{2\omega+1}$ results in the following stepsize condition:

$$
\gamma_4 \leq \left( L + \sqrt{\frac{16\omega(2\omega+1)}{n}} \widehat{L} \right)^{-1},
$$

as stated in Theorem 6.1 of Tyurin & Richtárik (2024). Here, $\widehat{L}$ satisfies $\widehat{L}^2 = \frac{1}{n}\sum_{i=1}^{n} L_i^2$, where $L_i$ is the smoothness constant of the local objective $f_i$. For simplicity, one can choose $\widehat{L} = L$. According to Corollary 5.3, the optimal stepsize matrix $\boldsymbol{D}_{\boldsymbol{L}^{-1}}^{**}$ is given by

$$
\boldsymbol{D}_{\boldsymbol{L}^{-1}}^{**} = \frac{2}{1 + \sqrt{1 + 16C_{\boldsymbol{L}^{-1}} \cdot \lambda_{\min}(\boldsymbol{L})}} \cdot \boldsymbol{L}^{-1}, \tag{56}
$$

when the momentum is set as $a = \frac{1}{2\omega_D+1}$.

As observed in Figure 8, det-DASHA with the matrix stepsize $\boldsymbol{D}_{\boldsymbol{L}^{-1}}^{**}$ outperforms DASHA with the optimal scalar stepsize using the same sketch in every setting we considered. Note that, since the same sketch is used for both algorithms, the number of bits transferred in each iteration is identical for both. This indicates that det-DASHA achieves better iteration complexity and communication complexity than DASHA.

## K.9 IMPROVEMENT OF DET-DASHA OVER NON-VARIANCE-REDUCED METHODS

In this experiment, we compare two non-variance-reduced methods, DCGD and det-CGD, with two variance-reduced methods, DASHA and det-DASHA. The stepsize choices for DCGD and det-CGD have already been discussed Appendix K.4. For DASHA and det-DASHA, we use the stepsize choices provided in Appendix K.8. We fix $\varepsilon^2$ at $0.01$, $\lambda$ at $0.9$, and use Rand-$\tau$ sketch throughout the experiment.

It is clear from Figure 9 that det-DASHA outperforms the other algorithms in each case. This is expected, as det-DASHA surpasses DASHA, a result also illustrated in Figure 8, which stems from using a matrix stepsize instead of a scalar stepsize. Additionally, we observe that det-DASHA and DASHA outperform det-CGD and DCGD, respectively, highlighting the advantages of the variance reduction technique. Note that in this case, all four algorithms use the same sketch, meaning the number of bits transferred in each iteration is identical for all algorithms. Consequently, compared to the others, det-DASHA excels in both iteration complexity and communication complexity.

## K.10 IMPROVEMENT OF DET-DASHA OVER DET-CGD

In this experiment, we compare det-DASHA and det-CGD using different matrix stepsizes. Throughout the experiment, we fix $\varepsilon^2 = 0.01$ and $\lambda = 0.9$, and the same Rand-$\tau$ sketch is used for both algorithms. For det-CGD, we use the stepsize $\boldsymbol{D}_1 = \gamma_{\boldsymbol{I}_d} \cdot \boldsymbol{I}_d, \boldsymbol{D}_2 = \gamma_{\mathrm{diag}^{-1}(\boldsymbol{L})} \cdot \mathrm{diag}^{-1}(\boldsymbol{L})$ and $\boldsymbol{D}_3 = \gamma_{\boldsymbol{L}^{-1}} \cdot \boldsymbol{L}^{-1}$, while for det-DASHA we use the stepsize $\boldsymbol{D}_{\boldsymbol{L}^{-1}}^{**}$.

It can be observed from Figure 10 that det-DASHA outperforms det-CGD with different stepsizes in all cases. This further corroborates our theory that det-DASHA is variance-reduced and, as a result, performs better in terms of both iteration complexity and communication complexity.

## K.11 DET-MARINA WITH DIFFERENT STEPSIZES

In this experiment, we compare det-DASHA using different matrix stepsizes.Specifically, we fix the matrix $\boldsymbol{W}$ to be one of three choices: $\boldsymbol{I}_d$, $\mathrm{diag}^{-1}(\boldsymbol{L})$, and $\boldsymbol{L}^{-1}$. We denote the corresponding

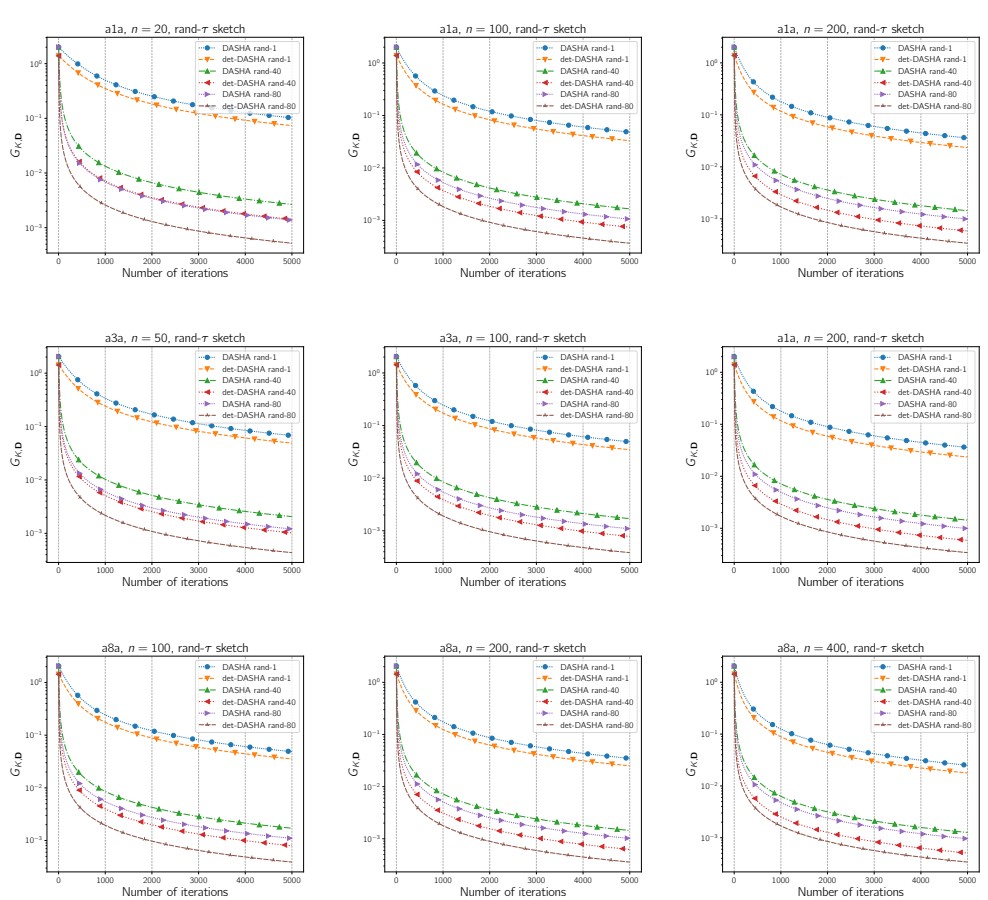

Figure 8: Comparison of det-DASHA with matrix stepsize $\boldsymbol{D}_{\boldsymbol{L}^{-1}}^{**}$ and DASHA with optimal scalar stepsize $\gamma$ using different rand-$\tau$ sketches. We fix $\lambda = 0.3$ throughout the experiments. The $x$-axis denotes the number of iterations while the notation $G_{K,\boldsymbol{D}}$ in the $y$-axis denotes the averaged matrix norm of the gradient. The notation $n$ denotes the number of clients in each setting.

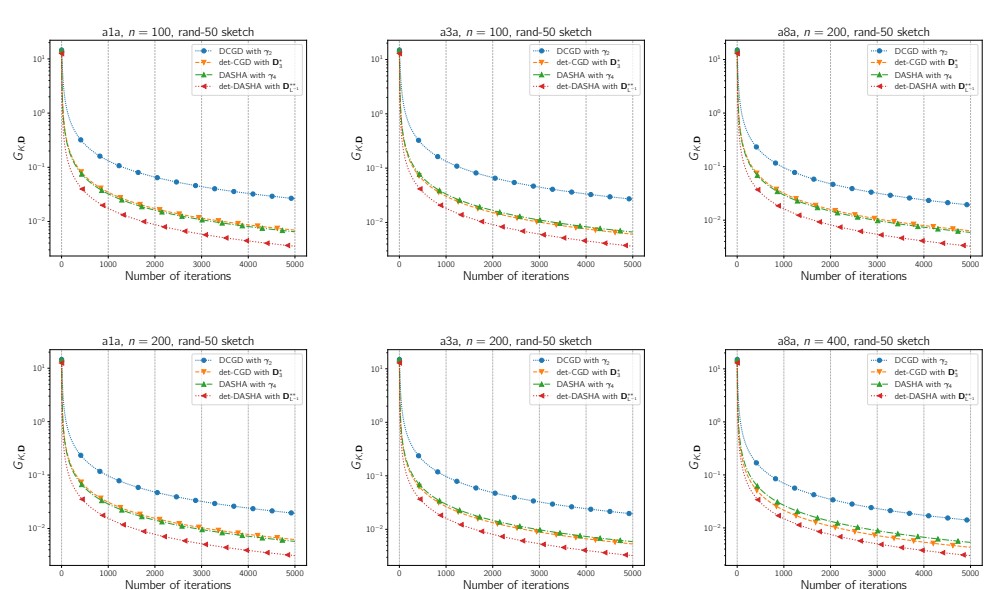

Figure 9: Comparison of DCGD with optimal scalar stepsize $\gamma_2$, det-CGD with optimal diagonal stepsize $\boldsymbol{D}_3^*$, DASHA with optimal scalar stepsize $\gamma_1$ and det-DASHA with optimal stepsize $\boldsymbol{D}_{L^{-1}}^{**}$. We fix $\lambda = 0.9$ throughout the experiment. The notation $n$ indicates the number of clients in each case. Rand-$\tau$ sketch with $\tau = 50$ are used in all cases.

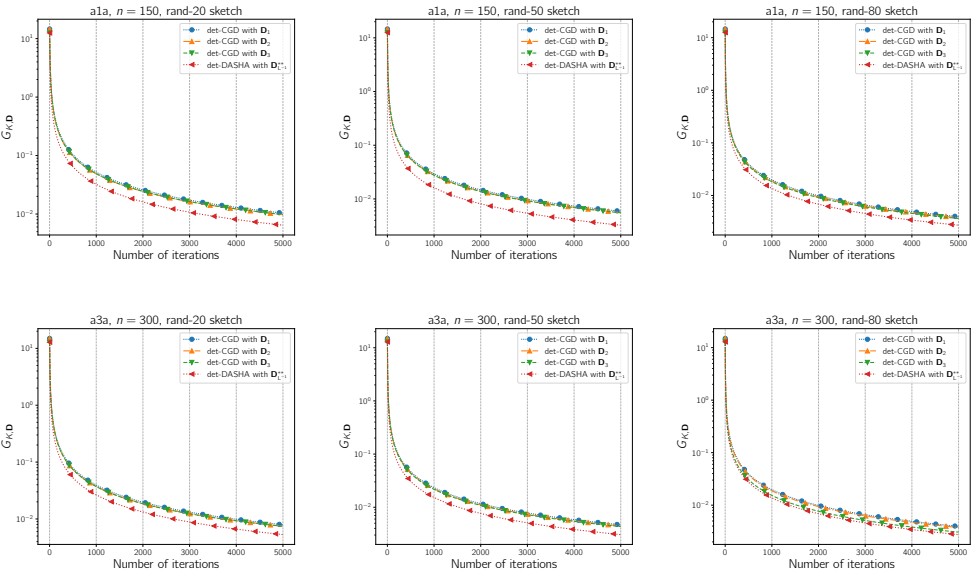

Figure 10: Comparison of det-DASHA with stepsize $\boldsymbol{D}_{L^{-1}}^{**}$ and det-CGD with three different stepsizes $\boldsymbol{D}_1$, $\boldsymbol{D}_2$ and $\boldsymbol{D}_3$. Throughout the experiment, $\lambda$ is fixed at 0.9, $\varepsilon^2$ is fixed at 0.01. Rand-$\tau$ sketch is used in all cases with $\tau$ selected from $\{20, 50, 80\}$.

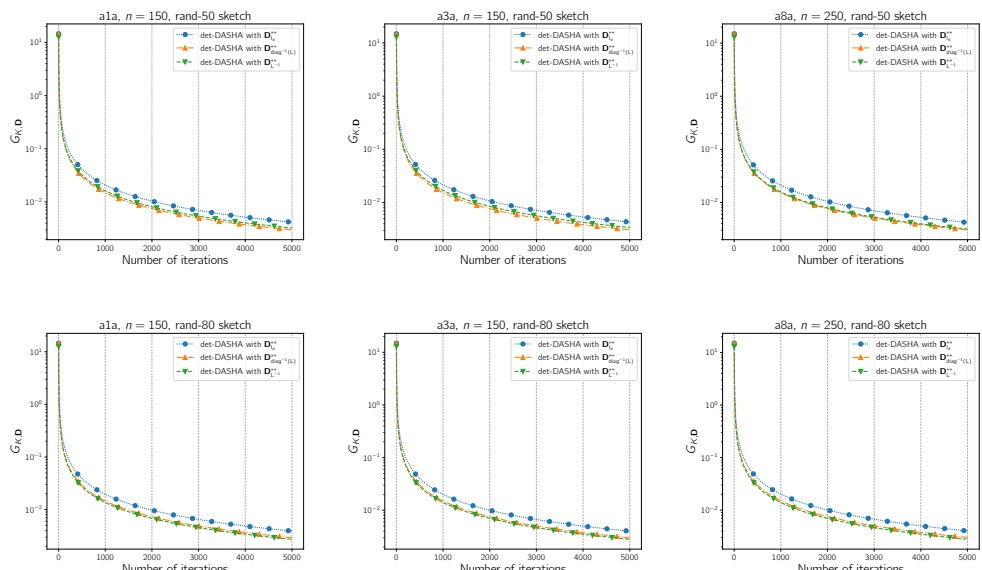

Figure 11: Comparison of det-DASHA three different stepsizes $\boldsymbol{D}_{\boldsymbol{L}^{-1}}^{**}$, $\boldsymbol{D}_{\mathrm{diag}^{-1}(\boldsymbol{L})}^{**}$ and $\boldsymbol{D}_{\boldsymbol{I}_d}^{**}$. The definition for those matrix stepsize notation are given in (56), (58) and (57) respectively. Throughout the experiment, $\lambda$ is fixed at $0.9$. Rand-$\tau$ sketch is used in all cases.

optimal stepsizes as $\boldsymbol{D}_{\boldsymbol{I}_d}^{**}$, $\boldsymbol{D}_{\mathrm{diag}^{-1}(\boldsymbol{L})}^{**}$ and $\boldsymbol{D}_{\boldsymbol{L}^{-1}}^{**}$. For $\boldsymbol{D}_{\boldsymbol{L}^{-1}}^{**}$, it is already given in (56). For $\boldsymbol{D}_{\boldsymbol{I}_d}^{**}$ and $\boldsymbol{D}_{\mathrm{diag}^{-1}(\boldsymbol{L})}^{**}$, we use Corollary 5.3 to compute them. As a result, we have

$$\boldsymbol{D}_{\boldsymbol{I}_d}^{**} = \frac{2}{1 + \sqrt{1 + 16 \cdot \frac{\omega_{\boldsymbol{I}_d}\left(4\omega_{\boldsymbol{I}_d}+1\right)}{n} \cdot \frac{\lambda_{\min}(\boldsymbol{L})}{\lambda_{\max}(\boldsymbol{L})}}} \cdot \frac{\boldsymbol{I}_d}{\lambda_{\max}(\boldsymbol{L})}, \tag{57}$$

$$\boldsymbol{D}_{\mathrm{diag}^{-1}(\boldsymbol{L})}^{**} = \frac{2}{1 + \sqrt{1 + 16 C_{\mathrm{diag}^{-1}(\boldsymbol{L})} \cdot \lambda_{\min}(\boldsymbol{L})}} \cdot \mathrm{diag}^{-1}(\boldsymbol{L}). \tag{58}$$

Throughout the experiment, $\lambda$ is fixed at $0.9$, rand-$\tau$ sketch is used for all the algorithms.

As observed in Figure 11, det-DASHA with $\boldsymbol{D}_{\boldsymbol{L}^{-1}}^{**}$ and $\boldsymbol{D}^{**}\mathrm{diag}^{-1}(\boldsymbol{L})$ both outperform det-DASHA with $\boldsymbol{D}^{**}\boldsymbol{I}_d$, demonstrating the effectiveness of using a matrix stepsize over a scalar stepsize. However, depending on the parameters of the problem, it is difficult to draw a general conclusion whether $\boldsymbol{D}_{\boldsymbol{L}^{-1}}^{**}$ is better than $\boldsymbol{D}_{\mathrm{diag}^{-1}(\boldsymbol{L})}^{**}$.

### K.12 COMPARISON OF DET-MARINA AND DET-DASHA

In this section, we provide a comparison between det-DASHA and det-MARINA. Both methods are variance-reduced versions of det-CGD, but they employ different variance reduction techniques. For det-MARINA, the method is based on MARINA and requires synchronization at intervals, depending on the probability parameter $p$. In contrast, det-DASHA utilizes the momentum variance reduction technique and does not require any synchronization at all. We primarily focus on the communication complexity, specifically the convergence with respect to the number of bits transferred. Throughout the experiment, we fix $\lambda = 0.9$. For det-DASHA we choose 3 different stepsizes: $\boldsymbol{D}_{\boldsymbol{I}_d}^{**}$, $\boldsymbol{D}_{\boldsymbol{L}^{-1}}^{**}$ and $\boldsymbol{D}_{\mathrm{diag}^{-1}(\boldsymbol{L})}^{**}$. For det-MARINA, we also select three stepsizes correspondingly: $\boldsymbol{D}_{\boldsymbol{I}_d}^{*}$, $\boldsymbol{D}_{\boldsymbol{L}^{-1}}^{*}$ and $\boldsymbol{D}_{\mathrm{diag}^{-1}(\boldsymbol{L})}^{*}$.

It is evident from Figure 12 that det-DASHA consistently exhibits better communication complexity compared to its det-MARINA. Note that since each algorithm is run for a fixed number of iterations, the $x$-axis actually records the total number of bytes transferred for each algorithm.

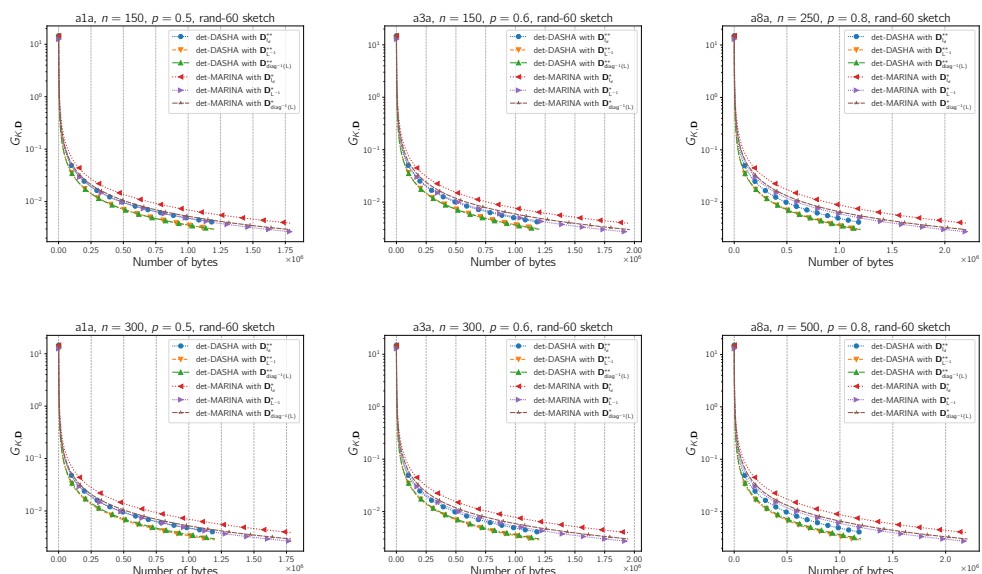

Figure 12: Comparison of det-DASHA with three different stepsizes $D_{I_d}^{**}$, $D_{L^{-1}}^{**}$ and $D_{\mathrm{diag}^{-1}(L)}^{**}$, and det-MARINA with $D_{I_d}^*$, $D_{L^{-1}}^*$ and $D_{\mathrm{diag}^{-1}(L)}^*$ in terms of communication complexity. Throughout the experiment, $\lambda$ is fixed at 0.9. Each algorithm is run for a fixed number of iteration $K = 5000$.

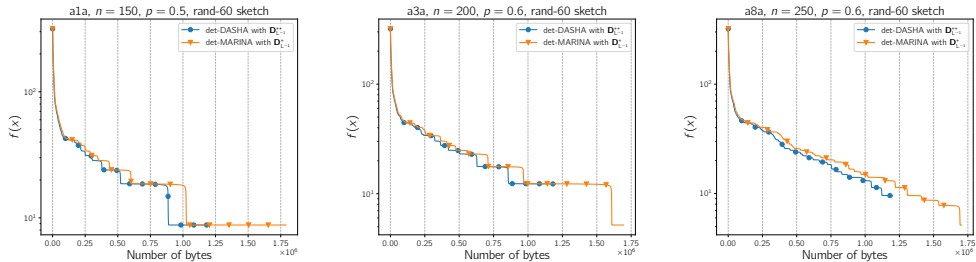

Figure 13: Comparing the performance of det-DASHA with $D^{**}L^{-1}$ and det-MARINA with $D^*L^{-1}$ in terms of the function value decreases. The function values for each algorithm represent the average of 20 runs using different random seeds. The two algorithms are initialized at the same starting point. The same rand-$\tau$ sketch is employed for both algorithms.

### K.13 COMPARISON IN TERMS OF FUNCTION VALUES

In this section, we compare det-MARINA and det-DASHA in terms of the decrease in function value. The two algorithms are initialized at the same starting point, and we run them 20 times before averaging the function values obtained in each iteration. The same sketch is used since we are interested in the performance in terms of communication complexity. We use $D_{L^{-1}}^{**}$ as the stepsize of det-DASHA and $D_{L^{-1}}^*$ as the stepsize of det-MARINA.

Observe that in Figure 13, the function values continuously decrease as the algorithms progress through more iterations. However, the stability observed here differs from that in the case of the average (matrix) norm of gradients. Our theoretical framework, as presented in this paper, primarily addresses the average norm of gradients in the non-convex case. Nonetheless, the experiment reinforces the effectiveness of our algorithms, showing consistent decreases in function values.

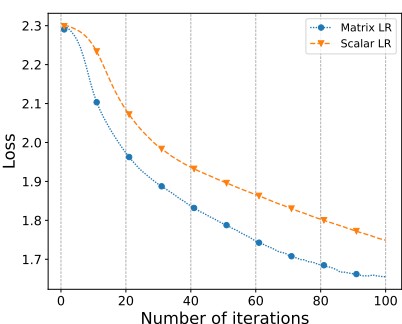 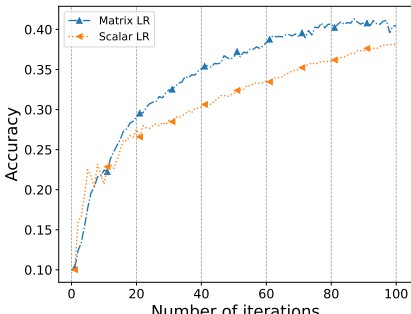

Figure 14: Deep learning experiment on CIFAR-10. We are comparing DCGD and distributed det-CGD with rand-100 sketches in this case using a simple three-layer neural network. Left: training loss curve. Right: test accuracy curve. The matrix stepsize is set as a layer-wise block-diagonal matrix. The results reported here reflect the final performance after appropriate tuning.

## K.14 DEEP LEARNING EXPERIEMNTS

In this section, we evaluate the proposed methods using a three-layer neural network on the CIFAR-10 classification task. We use the scalar stepsize variants of the algorithms as baselines and compare them against their matrix stepsize counterparts, where the stepsize matrix is chosen as a layer-wise block-diagonal matrix. As we can see from Figure 14, the matrix stepsize versions consistently outperform their scalar counterparts after proper tuning of both methods.

