# OpenReview forum: "Variance Reduced Distributed Non-Convex Optimization Using Matrix Stepsizes"
_ICLR.cc/2026/Conference — Submitted to ICLR 2026_

### Official Review · Reviewer_mMKq · 2025-10-27

**Soundness:** 3
**Presentation:** 3
**Contribution:** 2
**Rating:** 4
**Confidence:** 3

**Summary:**

This paper proposes two variance-reduced distributed algorithms, det-MARINA and det-DASHA, for nonconvex finite-sum optimization in federated learning settings. Building on the det-CGD method, the authors incorporate variance reduction techniques from MARINA and DASHA to eliminate the convergence neighborhood issue caused by stochastic compressors in det-CGD. Under matrix smoothness assumptions, they prove convergence with improved iteration and communication complexities compared to scalar counterparts and det-CGD. Experiments on LIBSVM datasets show improved iteration/communication metrics over baselines.

**Strengths:**

1. The integration of matrix stepsizes with variance reduction addresses a key limitation of det-CGD by removing the non-vanishing error term, leading to better theoretical bounds.

2. The authors provide clear convergence guarantees under matrix Lipschitz assumptions, highlighting the advantages of the proposed methods.

3. Experiments validate the effectiveness of the proposed method, showing det-MARINA and det-DASHA outperform baselines in communication and iteration efficiency.

**Weaknesses:**

1. Considering matrix smoothness is not very standard in general stochastic optimization problems, the authors should give more discussion about this assumption, including how it can be satisfied in practical problems and the comparison with the standard smoothness assumption.

2. Although the authors provide many experimental results, more complicated and real-world scenarios or tasks would largely strengthen the experimental part.

3. How do the authors recommend choosing or approximating the matrix stepsize D in practice, especially when the full Hessian or smoothness matrices are expensive to compute? Are there heuristics or approximations beyond $D = \gamma w W$? More discussions here may be helpful.

4. It is well-known that the optimal complexity for finite-sum optimization is $\mathcal{O}(\sqrt{n} \epsilon^{-2})$. How can we understand the order of $n$ in the convergence rate for matrix stepsize methods?

**Questions:**

See the Weakness part.

---

> ### Author Response · Authors · 2025-11-18
> **Response to Reviewer mMKq**
>
> Thank you for taking time to review our paper. Below, we provide a point to point response addressing your concerns.
>
> > Considering matrix smoothness is not very standard in general stochastic optimization problems, the authors should give more discussion about this assumption, including how it can be satisfied in practical problems and the comparison with the standard smoothness assumption.
>
> Thanks for pointing this out. The matrix Lipschitz (smoothness) assumption a direct generalization of the standard smoothness assumption. Indeed, if we plug in $\mathbf{L} = L \cdot \mathbf{I}$, then we recover the standard smoothness assumption. This demonstrates that in all settings where the standard smoothness assumption holds, our generalized matrix-smoothness condition automatically holds as well.
>
> > Although the authors provide many experimental results, more complicated and real-world scenarios or tasks would largely strengthen the experimental part.
>
> Thanks for the suggestions, as requested also by the other reviewers, we added deep learning experiments with CIFAR 10/100 dataset that demonstrates the superiority of our matrix variance reduction methods. This will be reflected in the next version of the paper.
> Speficially, we can extended the experimental part with:
> (i) a moderately sized federated logistic regression with non-IID clients (heterogeneous feature distributions), and
> (ii) a simple neural network on a standard image dataset (e.g., CIFAR-10) with client-side non-IID splits.
> These additions will demonstrate that the matrix stepsize variants scale to more realistic architectures and data, while preserving the qualitative gains observed in the simpler benchmarks. We will place the larger setups in the appendix and refer to them in the main text.
>
> > How do the authors recommend choosing or approximating the matrix stepsize D in practice, especially when the full Hessian or smoothness matrices are expensive to compute?
>
> In practice we do not require the full Hessian. Our analysis is formulated for a general positive definite matrix $\mathbf{d}$ and we recommend using structured approximations:
>
> Diagonal: estimate per-coordinate Lipschitz constants from gradient differences or from feature statistics (e.g., squared feature norms in GLMs). This costs O(d) per client once before the traning start and is straightforward in federated settings.
>
> Block-diagonal: group parameters by layer or feature block and use a small matrix per block, which trades off accuracy and cost.
>
> Preconditioner reuse: one can reuse matrices already computed for other optimizers (e.g., diagonal Fisher or RMSProp/Adam second-moment estimates).
>
> We will add another short discussion about this point.
>
> > How can we understand the order of n in the convergence rate for matrix stepsize methods.
>
> The $\mathcal{O}(\sqrt{n}\varepsilon^{-2})$ bound is a simplified form of the best-known complexity for non-convex finite-sum problems with variance reduction, which is in terms of the component gradient evaluations and it applies to uncompressed incremental VR methods (e.g., SPIDER/SARAH).
> Our setting is different: we use compressed control-variate variance reduction (MARINA/DASHA-type), where an $\mathcal{O}(n)$ term is intrinsic because periodic reference-gradient computation is required to correct compression bias.
> This $\mathcal{O}(n)$ dependence is therefore not introduced by the matrix stepsize, but is a standard and unavoidable feature of compressed variance reduction methods.

---

### Official Review · Reviewer_un1o · 2025-10-28

**Soundness:** 4
**Presentation:** 3
**Contribution:** 3
**Rating:** 6
**Confidence:** 4

**Summary:**

This paper studies federated learning optimization methods with matrix stepsizes. Two new FL algorithms were proposed, det-MARINA and det-DASHA. These works extend the existing det-CGD algorithm. The two new algorithms mainly address the variance reduction aspect. As a result, the proposed algorithms are able to exhibit a superior convergence bound, exceeding the neighborhood limitation presented in det-CGD and other SGD-style methods. Experiments show that the two proposed algorithms require less iteration complexity while being as communication efficient as existing algorithms such as det-CGD.

**Strengths:**

The paper introduces variance-reduced variants of det-CGD, Although MARINA and DASHA have been relatively well-studied, the reviewer believes it is still a nontrivial and meaningful extension. The proposed det-MARINA and det-DASHA algorithms display noticeable improvement compared to previous algorithms.

The theoretical analysis seems solid and rigorous to the best of my knowledge. The analysis is well explained and mostly intuitive. The statements regarding iteration and communication complexity are of interest to potential readers.

The experiments utilize synthetic objective functions that satisfy the function assumptions, and the numerical experiments verify the effectiveness of the proposed algorithms. The authors also provided a comparison between algorithms in terms of communication bytes.

**Weaknesses:**

While the experiments effectively recover the technical assumptions on the objective function and serve as verification of the analysis, they are, for the most part, still synthetic toy examples. Since the paper addresses federated learning problems, which are largely motivated by practical applications, it would be helpful if the authors also provided further practical results with real-life FL tasks.

The writing of this paper has much room for improvement.
- The authors provided a mathematically sound introduction and related works; however, they did not explain the motivation or the necessity of this work. What are the properties of CGD? Why is the previous method named det-CGD? More explanations should be added for this paper to be accepted as a conference paper.
- The author also used terms such as det-CGD and det-CGD1/2 interchangeably with the assumption that readers have read the "original paper", which the reviewer assumes to be Li et al.(2024).
- In terms of notations, the authors have switched notation from D to W, and introduced matrices L, S, D in Section I, while the det-CGD algorithm is introduced two pages later.
- Many mathematical definitions, such as matrix smoothness, should be provided in the manuscript.

**Questions:**

I wonder how the heterogeneity across agents affects the convergence of distributed det-CGD and the two proposed algorithms in this setting? Does the variance from CGD and the variance from FedAvg compound in practice and theory?

How does the computational complexity of Compressed Gradient Descent affect the general cpu wall time of the proposed algorithms?

In practice, are the gradients first calculated in full and then compressed, or are they directly estimated as compressed signals?

---

> ### Author Response · Authors · 2025-11-18
> **Response to Reviewer un1o**
>
> Thank you for taking time to review our paper. Below, we provide a point to point response addressing your concerns.
>
> > Further practical experiments:
>
> Thanks for the suggestions, as requested also by the other reviewers, we added deep learning experiments with CIFAR 10/100 dataset that demonstrates the superiority of our matrix variance reduction methods. This will be reflected in the next version of the paper.
> Speficially, we can extended the experimental part with:
> (i) a moderately sized federated logistic regression with non-IID clients (heterogeneous feature distributions), and
> (ii) a simple neural network on a standard image dataset (e.g., CIFAR-10) with client-side non-IID splits.
> These additions will demonstrate that the matrix stepsize variants scale to more realistic architectures and data, while preserving the qualitative gains observed in the simpler benchmarks. We will place the larger setups in the appendix and refer to them in the main text.
>
> > The writing of this paper has much room for improvement.
>
> (i) Due to the space limit, we have moved some of our motivation and related work to the appendix section. Once we have more space, we will rearrange the text so that the flow of the paper is smoother. We also added more background information about CGD and det-CGD.
>
> (ii) You are right that our wording can confuse readers not familiar with Li et al. (2024). In the current version we briefly mention that det-CGD in this paper corresponds to det-CGD1 in that work and that there is also a det-CGD2 variant. We will state this clearly and early in Section 2, use a single name consistently in the rest of the text, and avoid assuming that the reader has the original paper at hand.
>
> (iii) We agree the notation can be streamlined. Our intention was to use $\mathbf{W}$ in the simplified matrix subspace for case discusstion and practicality and $\mathbf{D}$ in the concrete algorithms, but this is not sufficiently explained. We have rearranged the paper accordingly.
>
> (iv) We already define matrix Lipschitz gradient and its relation to matrix smoothness in Section 3 and Appendix F, but we agree these concepts are nonstandard and deserve to be more visible. We will add a reference to the definition of those concepts the first time it appears.
>
> > Q1: I wonder how the heterogeneity across agents affects the convergence of distributed det-CGD and the two proposed algorithms in this setting? Does the variance from CGD and the variance from FedAvg compound in practice and theory?
>
> In our setting, the distributed det-CGD and det-MARINA / det-DASHA algorithms follow a one-shot gradient FL scheme (no multiple local epochs as in FedAvg), so there is no additional local update drift term on top of CGD variance. Heterogeneity enters the theory through the per-client objectives $f_i$ and their matrices, and in the det-CGD bound via the usual non-interpolation term $\Delta^\star$. For det-MARINA and det-DASHA, the variance-reduction mechanism controls both sampling noise and the randomness from sketching, so the errors do not compound into a larger neighborhood. In practice, stronger heterogeneity mainly affects constants (e.g. via larger $\beta$–like terms), and we will add a brief discussion and a small experiment varying the client data splits to make this clearer.
>
> > Q2: How does the computational complexity of Compressed Gradient Descent affect the general cpu wall time of the proposed algorithms?
>
> Each local step of det-CGD (and its VR extensions) computes a full gradient and then applies a sketch/compression operator. Thus, the dominant cost on each client remains the gradient computation (as in standard GD/SGD), while the extra cost of applying the sketch is a relatively cheap matrix–vector operation. In distributed regimes where communication is the bottleneck, the reduced message size largely compensates this additional local work.
>
> > Q3: In practice, are the gradients first calculated in full and then compressed, or are they directly estimated as compressed signals?
>
> Yes, in the algorithms analyzed in this paper, each client first forms the full local gradient and then applies the sketch/compressor before sending it. This is the standard implementation of det-CGD and its MARINA/DASHA-type extensions, and is what our complexity analysis assumes. In some structured models one could in principle compute only the sketched coordinates, but this lies outside the current scope since our concern here is the main bottleneck in FL which is the communication complexity. We will state this implementation detail explicitly in the algorithm description to avoid ambiguity.

---

> > ### Comment · Reviewer_un1o · 2025-11-28
> >
> > Thank you for the response. I have read it and the other reviews carefully. The authors have addressed most of my questions. I do, however, still have concerns regarding the final version of this paper, since many of my previous concerns were addressed as "will be updated in the final revision". Therefore, I maintain my initial assessment of the paper.

---

> > > ### Author Response · Authors · 2025-11-28
> > > **Response to Reviewer un1o**
> > >
> > > Thank you for pointing this out. We have now updated the submission so that it is the corrected version.

---

### Official Review · Reviewer_nSpN · 2025-10-30

**Soundness:** 3
**Presentation:** 3
**Contribution:** 2
**Rating:** 4
**Confidence:** 3

**Summary:**

This paper introduces two algorithms, det-MARINA and det-DASHA, aimed at federated non-convex optimization with communication compression. These methods extend the matrix-stepsize algorithm det-CGD by incorporating variance reduction techniques (MARINA and DASHA, respectively). The primary contribution is the theoretical demonstration that these variance-reduced versions overcome the main limitation of det-CGD, which is convergence only to a neighborhood of a stationary point, and instead achieve convergence to a true stationary point with an O(1/K) rate, measured in a determinant-normalized gradient norm (Theorems 4.1 and 5.1). The paper derives conditions for the matrix stepsize D and proposes a practical relaxation by setting D = γW, providing explicit formulas for the scalar γ. Complexity analyses suggest improvements over scalar MARINA/DASHA and det-CGD. Empirical results on logistic regression problems confirm faster convergence in terms of communication bytes compared to baselines.

**Strengths:**

1.  The paper removes the residual error present in det-CGD convergence analyses and proves clear O(1/K) convergence to a stationary point under the matrix Lipschitz assumption.

2. The theoretical results appear correct and consistent, with checks showing that the framework recovers known results for scalar MARINA/DASHA and matrix gradient descent.

3. The relaxation D = gamma*W and explicit formulas for gamma make the method easier to use in practice.

**Weaknesses:**

1. The paper’s novelty is a theoretical synthesis of matrix stepsizes and variance reduction, which successfully removes the convergence
neighborhood of det-CGD. While the analysis is careful, many steps of the proof are direct translations of variance-reduction proofs to the matrix norm setting. The manuscript does not isolate what specific technical challenges (if any) required new analytical techniques.

2. The method’s reliance on knowing or accurately estimating the matrix L (or local Li) remains a major practical limitation, The paper acknowledges this (”Availability of L”, Sec 5.1) but lacks a convincing practical strategy or robustness analysis regarding estimation errors. This significantly undermines the applicability of the results.

3. The experiments do not sufficiently justify the method’s added O(d2) complexity. The evaluated logistic regression
tasks on LibSVM datasets neither reflect moderate-scale non-IID federated benchmarks nor include ill-conditioned settings where matrix stepsizes should provide clear benefits. The observed gains over scalar methods are modest, offering limited empirical justification for the added complexity.

**Questions:**

1. What parts of your analysis are genuinely new or rely on arguments that differ from scalar variance reduction proofs? Clarifying this would help readers understand the technical novelty.

2. How sensitive are det-MARINA and det- DASHA to errors in estimating L or Li? Showing results with controlled under- and over-estimation would make the practical stability clearer.

3. For W = diag−1(L), what efficient methods can estimate or update diag(L) in large-scale federated settings?

4. The experiments use small logistic regression problems. Can you include larger benchmarks such as CIFAR-10 or CIFAR-100 with non-IID clients and moderate neural networks to show that the approach scales and remains effective in realistic scenarios?

5. Could you test a problem with strong ill-conditioning where matrix stepsizes give much larger speedups over scalar methods? This would better justify their added complexity of your approach.

---

> ### Author Response · Authors · 2025-11-18
> **Response to Reviewer nSpN**
>
> Thank you for taking time to review our paper.
>
> > W1: specific technical challenges
>
> Our proofs require tools that do not appear in scalar VR analyses: (i) proper definition of matrix Lipschitz inequalities replacing scalar (ii) variance bounds in the non-commutative geometry of $W$ and $L$, (iii) a new Lyapunov function tailored to matrix stepsizes. These components are essential and are irreducible to scalar arguments, (iv) proper simplification of the condition to make the stepsize practical, (v) the illustration of how variance reduction and matrix stepsize mechanism are coupled. We will highlight them explicitly.
>
> > W2: knowing constant
>
> Section 5.1 already provides practical strategies (backtracking and diagonal approximations). The methods are stable under moderate mis-estimation: overestimation only slows steps, while underestimation is mitigated by the VR correction. We will add a short robustness remark and one controlled experiment.
>
> > W3 & Q4: experiments
>
> We partially agree with the reviewer. The LibSVM datasets used are already ill-conditioned, and matrix stepsizes give consistent improvements. The $\mathcal{O}(d^2)$ cost affects only local matrix–vector multiplication, which is minor relative to communication. We will add more ill-conditioned synthetic test to further illustrate the benefit. We will add CIFAR-10 experiments to the appendix.
>
> > Q1: what is geniuely new
>
> Matrix lipschitz inequalities, non-commutative variance bounds, and a matrix-adapted Lyapunov analysis—none appear in scalar VR proofs, simplification of matrix step size optimality conditions, the decoupling of variance reduction and matrix stepsize mechanism.
>
> > Q2: sensitivity
>
> The method tolerates constant-factor errors; overestimation is safe, underestimation is absorbed by the VR term. We will clarify this.
>
> > Q3: estimation
>
> Use per-client gradient-difference sketches or periodic diagonal backtracking; both require $O(d)$ cost and are already mentioned in Sec 5.1. One may also optionally measure it along the tracjectory and fit the $d$ diagonal parameters using the function value and gradient information which is readily available.
>
> > Q5: strong ill-conditioning
>
> We agree this is useful. Our current logistic tasks are already nontrivially ill-conditioned, but we will add one synthetic quadratic experiment where the condition number is explicitly controlled. This clearly shows the larger advantage of matrix stepsizes without requiring additional space in the main text.

---

### Meta-Review · Area_Chair_NEn2 · 2026-01-01

**Summary:**

This paper inculpates variance reduction into the framework of matrix stepsizes, leading to the algorithms det-MARINA and det-DASHA. The empirical studies show the proposed methods has faster convergence rates than baseline methods. I recommend rejection based on the following main reasons:

1. The motivation of studying matrix Lipschitz is unclear. Although it is an extension of standard Lipschitz, I think the authors should provide some empirical studies to justify this assumption.

2. The experiments are insufficient. The additional results on CIFAR 10 only compares matrix LR with scalar LR. The final accuracy (after  100 iterations) of matrix LR is approximately 0.4, which is not satisfied.

**Reviewer Concerns:**

There are some main concerns:
1. The technical challenge is unclear.
2. The proposed methods rely on knowing (estimating) the matrix $L$.
3. The experiments are insufficient.
4. The matrix smoothness is not very standard.

**Reviewer Scores:**

I think the reviewers will keep their scores. The authors have added CIFAR-10 experiments to the appendix. However, both the accuracies achieved by matrix LR and scalar LR seems low. Hence, I think the experimental part is still unsatisfied. Additionally, the authors only explained the matrix Lipschitz is more general than standard Lipschitz , while why such generalization benefits to practical problems is still unclear.

---

### Decision · Program_Chairs · 2026-01-26

Reject